# Revisiting Neural Processes via Fourier Transform and Volterra Series

Peiman Mohseni [1]   Nick Duffield [1]   Raymond K. W. Wong [1]

## Abstract

Modeling unknown latent functions from finite, irregularly sampled measurements is a recurring challenge across science and engineering. Neural processes (NPs), a family of probabilistic functional models, are promising solutions—especially when endowed with domain-specific symmetries like translation equivariance, which improve sample efficiency and generalization. Yet existing translation-equivariant NPs face two limitations: (i) they stack generic components with non-linearities, obscuring the induced function class and limiting interpretability; and (ii) convolutional designs are limited by local receptive fields and the need to embed inputs onto a dense uniform grid, while attention-based alternatives lift these restrictions at quadratic cost in the number of observations. We address both with two contributions. First, using the Volterra expansion, we approximate continuous translation-equivariant operators by sums of higher-order convolutions, yielding analytical transparency while admitting efficient evaluation via first-order convolutions. Second, we introduce set Fourier convolutions (SFConvs), a frequency-domain parameterization that operates directly on irregularly sampled points, achieves approximately global receptive fields, and scales linearly in the number of observations. Building on these ideas, we propose two conditional NPs (CNPs): SFConvCNPs, which stack SFConv blocks with non-linearities, and SFVConvCNPs, which integrate the Volterra formulation. Experiments on synthetic and real-world datasets demonstrate our methods' efficacy against state-of-the-art baselines.

[1]Texas A&M University, College Station, Texas, USA. Correspondence to: Peiman Mohseni <peiman.mohseni@tamu.edu>. Code available at https://github.com/peiman-m/fourier-volterra-nps.

*Proceedings of the $43^{rd}$ International Conference on Machine Learning*, Seoul, South Korea. PMLR 306, 2026. Copyright 2026 by the author(s).

## 1. Introduction

Many scientific and engineering domains—such as turbulence modeling and climate science (Ravuri et al., 2021; Janny et al., 2023)—require reasoning about complex dynamical systems. These systems are naturally modeled as latent functions over continuous domains, yet are observed only through finitely many noisy and irregular measurements. From a probabilistic perspective, such data are generated by an underlying stochastic process. Gaussian processes (GPs; Rasmussen et al. 2006) provide a principled Bayesian framework with closed-form inference and uncertainty estimates, but their cubic computational cost and reliance on carefully designed kernels severely limit scalability, particularly in high-dimensional settings.

The scalability and expressive power of deep neural networks (LeCun et al., 2015) have motivated hybrid approaches that combine deep learning with GP-like probabilistic structure (MacKay, 1995; Korshunova et al., 2018; Sun et al., 2019; Dupont et al., 2021; Phillips et al., 2022). Among these, neural processes (NPs; Garnelo et al. 2018a;b)—a family of models that learn distributions over functions—have emerged as a particularly influential framework and have since been extended in numerous directions (Jha et al., 2022). In this work, we focus on conditional NPs (CNPs; Garnelo et al. 2018a), with particular emphasis on translation-equivariant designs (Gordon et al., 2019; Huang et al., 2023; Ashman et al., 2024a).

Translation-equivariant NPs build on the convolutional deep set framework of Gordon et al. (2019), which represents equivariant set functions as a functional embedding followed by a continuous translation-equivariant operator. In practice, this operator is realized by stacking convolution (LeCun et al., 1998; 1989) or attention (Vaswani et al., 2017) modules interleaved with pointwise non-linearities. Although such constructions are empirically effective and backed by universal approximation results (Hornik et al., 1989; Cybenko, 1989; Chen & Chen, 1995), they obscure the structure and representational capacity of the induced function class, limiting principled analysis and interpretability.

Beyond representational considerations, existing translation-equivariant NPs—notably convolutional CNPs (ConvCNPs; Gordon et al. 2019) and translation-equivariant transformer NPs (TE-TNPs; Ashman et al. 2024a)—exhibit complemen-

tary scalability limitations. ConvCNPs scale linearly with the number of observations, but to leverage convolutional networks they must embed irregularly sampled inputs onto dense uniform grids, with the dominant cost determined by the grid resolution. This step becomes impractical over large domains and in spatio-temporal settings, where grid size grows rapidly. Moreover, the fixed, local receptive fields of discrete convolution kernels can impede the capture of long-range dependencies. Transformer-based approaches, by contrast, operate directly on unstructured inputs and model global interactions through attention, dispensing with the gridding step. Full attention, however, comes at the cost of quadratic complexity in the number of data points—the complementary scalability bottleneck.

Motivated by these representational and scalability trade-offs, we propose designs that reconcile analytical transparency with practical efficiency. Our contributions are threefold:

**(1)** We approximate the continuous translation-equivariant operator in the convolutional deep set framework by a Volterra series (Volterra, 1887; 1930; Boyd et al., 1984; Boyd & Chua, 1985), i.e., a sum of convolutions of increasing order. This yields a principled and analytically transparent construction of translation-equivariant set functions with deep neural networks, while remaining amenable to practical computation (Roheda et al., 2024).

**(2)** We introduce set Fourier convolutions (SFConvs), a frequency-domain parameterization of convolutional deep sets built on a first-order convolution operator. SF-Convs act directly on irregularly sampled observations without grid construction, inducing approximately global receptive fields at linear cost in the number of observations. They are self-contained modules that compose readily into larger architectures.

**(3)** Building on SFConvs, we propose two new classes of CNPs: set Fourier ConvCNPs (SFConvCNPs), obtained by stacking SFConv blocks with pointwise non-linearities, and set Fourier Volterra ConvCNPs (SFV-ConvCNPs), constructed within the Volterra framework. We demonstrate the effectiveness of these models across multiple benchmarks, including comparisons to strong baselines and comprehensive ablation studies.

## 2. Background

### 2.1. Neural Processes

Let $\mathcal{X} = \mathbb{R}^{d_{\mathcal{X}}}$ and $\mathcal{Y} = \mathbb{R}^{d_{\mathcal{Y}}}$ denote Euclidean input and output spaces, with $d_{\mathcal{X}}, d_{\mathcal{Y}} \in \mathbb{N}$. We write $\mathcal{S}(\mathcal{X} \times \mathcal{Y})$ for the collection of all finite (possibly empty) subsets of $\mathcal{X} \times \mathcal{Y}$, and $\mathcal{P}(\mathcal{Y}^{\mathcal{X}})$ for the space of probability measures over measurable functions $f : \mathcal{X} \to \mathcal{Y}$. We assume an underlying data-generating process (prior) $\mu \in \mathcal{P}(\mathcal{Y}^{\mathcal{X}})$,

from which latent functions $f \sim \mu$ are drawn. A *task* $\mathcal{D}$ consists of a finite number of noisy observations from a realization $f$, partitioned into a context *set* and a query[1] *set* (Bruinsma, 2024; Ashman et al., 2024a;b):

$$\mathcal{D} = (\mathcal{D}_c, \mathcal{D}_q), \qquad \mathcal{D}_c, \mathcal{D}_q \in \mathcal{S}(\mathcal{X} \times \mathcal{Y}), \qquad (1)$$

with observations of the form

$$y = f(x) + \epsilon, \qquad \epsilon \sim \mathcal{N}(0, \sigma_0^2 I),$$

where $\sigma_0 > 0$ is the noise standard deviation and $I \in \mathbb{R}^{d_{\mathcal{Y}} \times d_{\mathcal{Y}}}$ is the identity matrix. Noise variables are assumed independent across observations and independent of $f$.

Neural processes (NPs; Garnelo et al. 2018a;b) are a class of models that employ neural networks to learn a mapping $\eta : \mathcal{S}(\mathcal{X} \times \mathcal{Y}) \to \mathcal{P}(\mathcal{Y}^{\mathcal{X}})$, which assigns a context set $\mathcal{D}_c$ to a stochastic process $\eta[\mathcal{D}_c]$ intended to approximate the posterior induced by $\mu$ conditioned on $\mathcal{D}_c$. In the absence of context, the mapping should ideally recover the prior, i.e., $\eta[\varnothing] = \mu$ (Bruinsma, 2024; Ashman et al., 2024a). Since $\mathcal{D}_c$ is unordered, $\eta$ must be invariant to the permutation of its elements (Zaheer et al., 2017). In practice, most NPs do *not* explicitly construct the measure $\eta[\mathcal{D}_c]$, but instead specify it through its finite-dimensional distributions (FDDs; Klenke 2008). For a finite collection of query inputs $\mathbf{x}_q = (x_{q,k})_{k=1}^{n_q}$, we denote the associated joint FDD by $\eta[\mathcal{D}_c; \mathbf{x}_q]$. Provided these FDDs satisfy the consistency conditions of the Kolmogorov Extension Theorem (Klenke, 2008), they uniquely determine a stochastic process.

Different parameterizations of these FDDs give rise to different classes of NPs. In this work, we focus on conditional neural processes (CNPs; Garnelo et al. 2018a), which impose a mean-field Gaussian structure on the FDDs. Concretely, $\eta[\mathcal{D}_c; \mathbf{x}_q] = \prod_{k=1}^{n_q} \eta[\mathcal{D}_c; x_{q,k}]$, where each marginal $\eta[\mathcal{D}_c; x_{q,k}]$ is a Gaussian. Most CNPs parameterize these FDDs using an encoder–decoder architecture of the form $\eta = \rho_d \circ \rho_e$ (Bruinsma, 2024; Ashman et al., 2025). The encoder $\rho_e : \mathcal{S}(\mathcal{X} \times \mathcal{Y}) \to \mathcal{H}$ embeds the context set $\mathcal{D}_c$ to a latent space $\mathcal{H}$, while the decoder $\rho_d : \mathcal{H} \to \Theta^{\mathcal{X}}$ maps the latent code to a function that assigns, to each query $x_q \in \mathcal{X}$, the parameters $\theta(x_q) \in \Theta$ of the predictive distribution at $x_q$. Given a distribution $q(\mathcal{D})$ over tasks of the form (1), CNPs are commonly trained by minimizing the expected negative log-likelihood

$$\mathcal{L}(\phi) = -\mathbb{E}_{(\mathcal{D}_c, \mathcal{D}_q) \sim q(\mathcal{D})} \left[ \sum_{(x_q, y_q) \in \mathcal{D}_q} \log p_{\eta[\mathcal{D}_c]}(y_q \mid x_q) \right],$$

where $p_{\eta[\mathcal{D}_c]}(\cdot \mid x_q)$ denotes the density of the FDD $\eta[\mathcal{D}_c; x_q]$, and $\phi$ collects the parameters of the model (Bruinsma et al., 2021; Ashman et al., 2024a; Bruinsma, 2024).[2]

---

[1]The NPs literature commonly uses the term "target points". To avoid ambiguity with time, we use "query points".

[2]We index by $\mathcal{D}_c$ rather than condition on it because NPs

## 2.2. Translation-equivariant Neural Processes

The ability of NPs to operate on variable-size sets of unstructured observations while producing probabilistic predictions at arbitrary query locations makes them particularly well suited to scientific applications, where data often arise from spatio-temporal fields such as climate and physical simulation systems (Vaughan et al., 2021; Scholz et al., 2023; Allen et al., 2025; Ashman et al., 2025). In many such domains, the underlying data-generating processes exhibit symmetries—most notably translation equivariance—that can be exploited to improve both data efficiency and generalization (Brown & Lunter, 2019; Finzi et al., 2020; Bulusu et al., 2021; Holderrieth et al., 2021; Gruver et al., 2022; Bruinsma, 2024; Ashman et al., 2024a).

Motivated by these considerations, Gordon et al. (2019) introduced the *convolutional deep set* representation, which characterizes a broad class of translation-equivariant prediction maps. Let $\mathbf{x}_q$ denote a finite set of query inputs, $\tau \in \mathcal{X}$ a translation, and $\mathcal{D}_c$ a context set. We define the translated context set and query inputs as $\mathcal{T}_\tau \mathcal{D}_c = \{(x_c + \tau, y_c) \mid (x_c, y_c) \in \mathcal{D}_c\}$ and $\mathcal{T}_\tau \mathbf{x}_q = (x_{q,k} - \tau)_k$, respectively. A prediction map $\eta$ is said to be *translation equivariant* if, for all $\mathbf{x}_q$, $\tau$, and $\mathcal{D}_c$, it satisfies $\eta[\mathcal{T}_\tau \mathcal{D}_c; \mathbf{x}_q] = \eta[\mathcal{D}_c; \mathcal{T}_\tau \mathbf{x}_q]$. Under mild regularity assumptions, the convolutional deep set theorem states that any such translation-equivariant map $\eta$ can be realized via a two-stage construction. First, the context set $\mathcal{D}_c$ is embedded into a function-valued representation $\rho_e[\mathcal{D}_c] \in \mathcal{H} \subseteq \mathcal{Z}^{\mathcal{X}}$ according to

$$\rho_e[\mathcal{D}_c](\cdot) = \sum_{(x_c, y_c) \in \mathcal{D}_c} \varphi(y_c)\, \psi_e(\cdot - x_c), \qquad (2)$$

where $\varphi(y) = (1, y)$ and $\mathcal{Z} = \mathbb{R}^{1+d_{\mathcal{Y}}}$.[3] Here, $\psi_e : \mathcal{X} \to \mathbb{R}$ is a continuous, strictly positive-definite kernel, typically chosen to be Gaussian. Second, this functional embedding is processed by a translation equivariant operator $\rho_d$, yielding the prediction function $\eta[\mathcal{D}_c\,;\,\cdot] = (\rho_d \circ \rho_e)[\mathcal{D}_c](\cdot)$. Translation equivariance of $\rho_d$ requires that, for all $g \in \mathcal{Z}^{\mathcal{X}}$ and $\tau \in \mathcal{X}$, $\rho_d[\mathcal{T}_\tau g] = \mathcal{T}_\tau \rho_d[g]$, where $\mathcal{T}_\tau g(\cdot) = g(\cdot - \tau)$ and $\mathcal{T}_\tau \rho_d[g](\cdot) = \rho_d[g](\cdot - \tau)$.

## 3. Method

### 3.1. Volterra representation

In contrast to the functional embedding in (2), whose structure is explicit and readily interpretable, the convolutional deep set representation leaves $\rho_d$ largely unspecified, imposing only mild regularity conditions such as continuity. In the *linear* and *continuous* regime, translation equivariance fully

are generally not Bayes-consistent under conditioning (Bruinsma, 2024; Xu et al., 2023).

[3] For scalar outputs with repeated inputs of multiplicity $M$, one may take $\varphi(y) = (1, y, \ldots, y^M)$ (Gordon et al., 2019).

characterizes $\rho_d$ as a standard convolution operator (Kondor & Trivedi, 2018). Beyond linearity, most existing deep learning constructions build $\rho_d$ by composing linear convolutional modules or non-linear softmax attention modules (Appendix B) with pointwise non-linearities (Vaswani et al., 2017; Calvello et al., 2024; Ashman et al., 2024a). While universal approximation results ensure that sufficiently expressive instantiations of these architectures can approximate broad classes of (non-linear) operators (Hornik et al., 1989; Cybenko, 1989; Chen & Chen, 1995), such guarantees provide little guidance for principled architectural design (Azeglio et al., 2025). In particular, they do not delineate which subclasses of translation-equivariant operators are representable by a given architecture, nor how individual architectural components encode specific interaction structures or inductive biases. Consequently, architectural choices are often guided by empirical heuristics rather than systematic operator-theoretic considerations.

A principled alternative is provided by the theory of Volterra series, which offers an explicit and constructive characterization of translation-equivariant operators (Boyd et al., 1984; Wray & Green, 1994; Boyd & Chua, 1985; Li et al., 2022). In analogy with the Taylor expansion of a scalar function, a Volterra series can approximate a continuous translation-equivariant operator, under suitable regularity conditions and to arbitrary accuracy, as an (infinite) sum of multilinear convolution operators of increasing order. For clarity of exposition, we present the construction for scalar-valued signals $h : \mathcal{X} \to \mathbb{R}$ (i.e., $\mathcal{Z} = \mathbb{R}$), deferring the general vector-valued case to Appendix C. Applying the Volterra expansion to the operator $\rho_d$ yields

$$\rho_d[h] = \mathcal{C}_0 + \sum_{l=1}^{\infty} \mathcal{C}_l[h], \qquad (3)$$

where $\mathcal{C}_0 = \kappa_0 \in \mathbb{R}$ is a constant bias term and, for each $l \in \mathbb{N}$, the operator $\mathcal{C}_l$ is the $l$-th order convolution

$$\mathcal{C}_l[h](x) = \int_{\mathcal{X}^l} \kappa_l(x - s_1, \ldots, x - s_l) \prod_{k=1}^{l} h(s_k)\, \mathrm{d}s_{1:l},$$

where $\mathrm{d}s_{1:l} := \mathrm{d}s_1 \cdots \mathrm{d}s_l$ and $\kappa_l : \mathcal{X}^l \to \mathbb{R}$ is absolutely integrable. When unambiguous, we use the shorthand $\kappa_l(\cdot - s_{1:l})$ to denote $\kappa_l(\cdot - s_1, \ldots, \cdot - s_l)$.

A direct evaluation of (3) is computationally infeasible due to the exponential cost of high-order convolutions. Crucially, the multilinearity of Volterra operators in $h$, together with Fubini's theorem (Folland, 1999), allows higher-order terms to be reorganized into a *nested (cascade) form* (Rauf, 1993; Osowski & Quang, 1994; Roheda et al., 2024), in which variables are integrated sequentially. Specifically, the Volterra series can be written as

$$\rho_d[h](x) = \mathcal{C}_0 + \int \mathcal{V}_1[h](x, s_1)\, h(s_1)\, \mathrm{d}s_1,$$

where the auxiliary operators $\{\mathcal{V}_l\}_{l \geq 1}$ are defined by

$$\mathcal{V}_l[h](x, s_{1:l}) = \kappa_l(x - s_{1:l})$$
$$+ \int \mathcal{V}_{l+1}[h](x, s_{1:l+1}) \, h(s_{l+1}) \, \mathrm{d}s_{l+1}. \quad (4)$$

Multiplying by $h(s_l)$ and integrating w.r.t. $s_l$ yields

$$\int \mathcal{V}_l[h](x, s_{1:l}) \, h(s_l) \, \mathrm{d}s_l = \int \kappa_l(x - s_{1:l}) \, h(s_l) \, \mathrm{d}s_l$$
$$+ \iint \mathcal{V}_{l+1}[h](x, s_{1:l+1}) \, h(s_l) \, h(s_{l+1}) \, \mathrm{d}s_{l:l+1}. \quad (5)$$

The first term is a standard first-order convolution over $s_l$. The second defines a second-order convolution over $(s_l, s_{l+1})$, hereafter denoted $\mathcal{Q}_{l+1}[h](x; s_{1:l-1})$. Crucially, every kernel entering $\mathcal{V}_{l+1}$ is evaluated only at relative displacements of the form $x - s_k$, preserving equivariance.

Repeated application of this decomposition reduces the evaluation of a truncated Volterra expansion to first- and second-order convolutions (Roheda et al., 2024). While first-order convolutions are ubiquitous and highly optimized in modern deep learning frameworks (Zoumpourlis et al., 2017; Paszke et al., 2019), second-order convolutions remain a bottleneck, due to (i) the quadratic cost of explicitly forming $h \otimes h$, and (ii) the limited library support for $2d_{\mathcal{X}}$-dimensional convolutions when $d_{\mathcal{X}} > 1$.

To address this, following Roheda et al. (2024), we impose a low-rank structure on the quadratic kernel via a Hilbert–Schmidt decomposition. Specifically, fixing all variables in $\mathcal{V}_{l+1}$ except $(s_l, s_{l+1})$ and assuming $\mathcal{V}_{l+1}[h](x, s_{1:l+1}) \in L^2(\mathcal{X} \times \mathcal{X})$ in $(s_l, s_{l+1})$, we obtain

$$\mathcal{V}_{l+1}[h](x, s_{1:l+1}) = \sum_{r=1}^{\infty} \Big( \lambda_{l+1,r}[h](x, s_{1:l-1})$$
$$\times \Upsilon^{(1)}_{l+1,r}[h](x, s_{1:l-1}, s_l) \, \Upsilon^{(2)}_{l+1,r}[h](x, s_{1:l-1}, s_{l+1}) \Big), \quad (6)$$

where $\{\Upsilon^{(1)}_{l+1,r}\}_r$ and $\{\Upsilon^{(2)}_{l+1,r}\}_r$ are orthonormal families in $L^2(\mathcal{X})$, and the singular values satisfy $\lambda_{l+1,r} \geq 0$ and $\sum_{r \geq 1} \lambda^2_{l+1,r} < \infty$. Expanding $\mathcal{Q}_{l+1}$ using (6) yields

$$\mathcal{Q}_{l+1}[h](x; s_{1:l-1}) = \sum_{r=1}^{\infty} \Big( \lambda_{l+1,r}[h](x, s_{1:l-1})$$
$$\times \Big( \int \Upsilon^{(1)}_{l+1,r}[h](x, s_{1:l-1}, s_l) h(s_l) \mathrm{d}s_l \Big)$$
$$\times \Big( \int \Upsilon^{(2)}_{l+1,r}[h](x, s_{1:l-1}, s_{l+1}) h(s_{l+1}) \mathrm{d}s_{l+1} \Big) \Big).$$

Truncating the series after $R$ terms yields an approximation of $\mathcal{Q}_{l+1}[h](x; s_{1:l-1})$ in which each summand is the product of two first-order convolutions, resulting in a total of $2R + 1$ first-order convolutions per recursion step in (5).

In practice, following Roheda et al. (2024), we directly parameterize and learn the corresponding kernels and mixing coefficients, without explicitly enforcing the orthonormality

or nonnegativity constraints implied by the spectral decomposition. To evaluate the cascade, we fix a finite depth $L$ and set $\mathcal{V}_L[h](x, s_{1:L}) = \kappa_L(x - s_{1:L})$, i.e., we truncate the recursion by setting the integral term in (4) to zero at level $l = L$. This terminal condition is then propagated recursively upward from depth $L - 1$ to $l = 1$, yielding an efficient approximation of the truncated Volterra series.

### 3.2. Set Fourier Convolution

Whether arising from the Volterra construction introduced in the previous section or instantiated via standard convolutional neural networks (CNNs; Fukushima 1980; LeCun et al. 1989; 1998), first-order convolution operators constitute the fundamental building blocks for modeling dependencies among data points. Unlike the higher-order terms of Section 3.1, which require multilinear, tensor-valued kernels to handle vector-valued signals (Appendix C), first-order operators do so at no notational cost—the kernel is simply matrix-valued—and we therefore work directly in the multi-channel form in which they are implemented. Such operators take the form

$$\mathcal{K}[h](x) = \int_{\mathcal{X}} \kappa(x - s) \, h(s) \, \mathrm{d}s, \quad (7)$$

where $h \colon \mathcal{X} \to \mathbb{R}^{c_{\mathrm{in}}}$ denotes the input function with $c_{\mathrm{in}}$ channels ($c_{\mathrm{in}} = \dim \mathcal{Z}$ for $h = \rho_e[\mathcal{D}_c]$), $\kappa \colon \mathcal{X} \to \mathbb{R}^{c_{\mathrm{out}} \times c_{\mathrm{in}}}$ is a learnable matrix-valued convolution kernel, and $\kappa(x - s) \, h(s)$ denotes a matrix–vector product.

In practice, implementations of (7) almost universally assume that inputs are observed on a *uniform grid* with *fixed resolution*. Consequently, ConvCNP-style models in which $\rho_d$ is instantiated using CNN architectures such as ResNet (He et al., 2016) or U-Net (Ronneberger et al., 2015) require discretizing $\rho_e[\mathcal{D}_c]$ over a uniform grid $\mathcal{G} \subset \mathcal{X}$ whose spatial extent covers both context and query locations (Section 2.2).[4] This discretization entails a fundamental trade-off: coarse grids lead to loss of information, while fine grids incur rapidly increasing computational and memory costs that scale exponentially with $d_{\mathcal{X}}$. As a result, ConvCNPs scale poorly to large spatial domains and to many higher-dimensional spatio-temporal settings of practical interest.

Beyond computational considerations, discrete convolution kernels exhibit *fixed, localized receptive fields* (Luo et al., 2016), limiting their ability to capture long-range dependencies—especially for sparse or irregularly sampled data (Peng et al., 2017; Wang et al., 2018; Huang et al., 2019; Ramachandran et al., 2019; Wang et al., 2020a; Romero et al., 2021; Ding et al., 2022; Knigge et al., 2023; Mohseni & Duffield, 2025). Transformer-based architectures partially alleviate these issues by enabling global interactions with-

---

[4]The context and query locations themselves need not coincide with grid points.

out explicit gridding; however, their quadratic complexity in the number of context points renders them impractical for large context sets unless aggressive approximations or structural constraints are imposed, often weakening the very properties that make them appealing (Nguyen & Grover, 2022; Feng et al., 2022; Ashman et al., 2024a; 2025).

Taken together, these limitations motivate the search for other representations. Since our operator of interest is convolution (7), the Fourier basis (Fourier, 1888) is a natural choice. Throughout, we adopt the convention $i^2 = -1$ and define the $d_\mathcal{X}$-dimensional Fourier transform as

$$\mathcal{F}_{d_\mathcal{X}}[g](\xi) := \int_\mathcal{X} g(x)\, e^{-i2\pi\langle x, \xi\rangle}\, \mathrm{d}x, \qquad g \in L^2(\mathcal{X}),$$

where $\xi \in \mathbb{R}^{d_\mathcal{X}}$ is a frequency vector and $\langle \cdot, \cdot \rangle$ denotes the Euclidean inner product; the transform is applied entrywise to vector- and matrix-valued functions, and we write $\widehat{g} := \mathcal{F}_{d_\mathcal{X}}[g]$. In this basis, the convolution theorem (Borel, 1899; Weil, 1951) states that the operator $\mathcal{K}$ acts by pointwise multiplication in frequency:

$$(\mathcal{F}_{d_\mathcal{X}} \circ \mathcal{K})[h](\xi) = \widehat{\kappa}(\xi)\,\widehat{h}(\xi), \qquad (8)$$

where $\widehat{\kappa}(\xi) \in \mathbb{C}^{c_{\mathrm{out}} \times c_{\mathrm{in}}}$ acts on $\widehat{h}(\xi) \in \mathbb{C}^{c_{\mathrm{in}}}$ by matrix–vector multiplication.

This viewpoint is attractive for two reasons. First, pointwise multiplication in the frequency domain is dual to convolution with generally non-local kernels in the spatial domain, thereby inducing global receptive fields and overcoming the locality limitations of discrete convolutions (Chi et al., 2020). Second, natural signals are empirically known to concentrate most of their energy in low-frequency bands (Field, 1987; Ruderman & Bialek, 1993; Wainwright & Simoncelli, 1999). As a result, convolutions can often be well approximated in the Fourier domain using a finite set of frequencies $\Xi \subset \mathbb{R}^{d_\mathcal{X}}$ that capture most of the signal structure.

Motivated by these observations, we parameterize $\widehat{\kappa}$ directly (Rippel et al., 2015). Specifically, for each $\xi \in \Xi$, we associate a complex-valued weight matrix $W_\xi \in \mathbb{C}^{c_{\mathrm{out}} \times c_{\mathrm{in}}}$, consistent with the matrix-valued kernel in (7), and set $\widehat{\kappa}(\xi) = W_\xi$. The matrix-valued parameterization enables cross-channel mixing. This formulation mirrors the spectral parameterization adopted by Fourier neural operators (FNOs; Li et al. 2020). With the kernel thus specified, the remaining challenge is to compute $\widehat{h}$.

In standard FNOs, $\widehat{h}$ is obtained by discretizing $h$ on a regular grid and applying the discrete Fourier transform (DFT), computed via the fast Fourier transform (FFT; Cooley & Tukey 1965; Frigo & Johnson 2005). In our setting, where $h := \rho_e[\mathcal{D}_c]$, this discretization reintroduces the very grid-based limitations we seek to avoid. We circumvent it by exploiting the form of the functional embedding $\rho_e[\mathcal{D}_c]$

in (2), which admits a closed-form Fourier transform,

$$(\mathcal{F}_{d_\mathcal{X}} \circ \rho_e)[\mathcal{D}_c](\xi) =$$
$$\mathcal{F}_{d_\mathcal{X}}[\psi_e](\xi) \sum_{(x_c, y_c) \in \mathcal{D}_c} \varphi(y_c)\, e^{-i2\pi\langle x_c, \xi\rangle}. \quad (9)$$

This formulation is especially convenient when the kernel is Gaussian, $\psi_e(x) = \exp\left(-\frac{1}{2} x^\top \Sigma^{-1} x\right)$ with diagonal covariance $\Sigma = \mathrm{diag}(\varrho_1^2, \ldots, \varrho_{d_\mathcal{X}}^2)$ and $\varrho_d > 0$ for all $d \in \{1, \ldots, d_\mathcal{X}\}$, since its Fourier transform is also available in closed form:

$$\mathcal{F}_{d_\mathcal{X}}[\psi_e](\xi) = (2\pi)^{d_\mathcal{X}/2} \left(\prod_{d=1}^{d_\mathcal{X}} \varrho_d\right) e^{-2\pi^2 \xi^\top \Sigma \xi}. \quad (10)$$

Together, these expressions enable exact evaluation of $\widehat{h}$ at arbitrary frequencies, including all $\xi \in \Xi$, without recourse to spatial discretization. As a result, the spectral coefficients are indexed directly by physical frequencies.

Finally, after multiplying in the frequency domain, we recover $\mathcal{K}[h]$ via the inverse Fourier transform. Throughout this work, we take $\Xi$ to be fixed, uniform, and symmetric about the origin, so that the inverse transform admits a simple approximation over the bounded region $\Omega \subset \mathbb{R}^{d_\mathcal{X}}$ formed by axis-aligned bins centered at the points of $\Xi$, each of volume $\mathrm{vol}(\Omega)/|\Xi|$:

$$\mathcal{K}[h](x) = \mathcal{F}_{d_\mathcal{X}}^{-1}\left[\widehat{\kappa}(\xi)\,\widehat{h}(\xi)\right](x)$$
$$= \int \widehat{\kappa}(\xi)\,\widehat{h}(\xi)\, e^{i2\pi\langle x, \xi\rangle}\, \mathrm{d}\xi$$
$$\approx \frac{\mathrm{vol}(\Omega)}{|\Xi|} \sum_{\xi \in \Xi} W_\xi\, \widehat{h}(\xi)\, e^{i2\pi\langle x, \xi\rangle}.$$

Since $\mathcal{K}[h]$ must be real-valued, we impose the Hermitian symmetry $W_{-\xi} = \overline{W_\xi}$, which guarantees a real sum and halves the frequencies to be parameterized (Appendix E.1.1). We call these parameterizations *set Fourier convolutions* (SFConvs). Operating directly on unstructured input locations, SFConvs avoid spatial gridding altogether, and their frequency-domain parameterization affords approximately global receptive fields at the linear cost of spatial convolutions—well suited to large context sets.

### 3.3. SFConv-Based Neural Processes

Building on SFConvs, we introduce two new classes of CNPs: **(i)** set Fourier ConvCNPs (SFConvCNPs), which are constructed by stacking multiple SFConv blocks interleaved with pointwise non-linearities, and **(ii)** set Fourier Volterra ConvCNPs (SFVConvCNPs), which follow the Volterra construction described in Section 3.1 and employ an SFConv-style parameterization for the first-order convolutional terms.

*Table 1.* Comparison of various classes of NPs across key design axes.

| Model Class | Computational Complexity | Translation Equivariance | Global Interaction | Discretization Domain | Interpretable Design |
|---|---|---|---|---|---|
| CNPs | $\mathcal{O}(|\mathcal{D}_c| + |\mathcal{D}_q|)$ | ✗ | ✗ | None | ✗ |
| AttnCNPs | $\mathcal{O}(|\mathcal{D}_c|^2 + |\mathcal{D}_c||\mathcal{D}_q|)$ | ✗ | ✓ | None | ✗ |
| TNPs | $\mathcal{O}(|\mathcal{D}_c|^2 + |\mathcal{D}_c||\mathcal{D}_q|)$ | ✗ | ✓ | None | ✗ |
| PT-TNPs | $\mathcal{O}(M(|\mathcal{D}_c| + |\mathcal{D}_q|))$ | ✗ | ✓ | None | ✗ |
| TE-TNPs | $\mathcal{O}(|\mathcal{D}_c|^2 + |\mathcal{D}_c||\mathcal{D}_q|)$ | ✓ | ✓ | None | ✗ |
| TE-PT-TNPs | $\mathcal{O}(M(|\mathcal{D}_c| + |\mathcal{D}_q|))$ | ⚠ | ✓ | None | ✗ |
| ConvCNPs | $\mathcal{O}(|\mathcal{G}|(|\mathcal{D}_c| + |\mathcal{D}_q|))$ | ✓ | ✗ | Spatial | ✗ |
| SConvCNPs | $\mathcal{O}(|\mathcal{G}|(|\mathcal{D}_c| + \log|\mathcal{G}| + |\mathcal{D}_q|))$ | ✓ | ✓ | Spatial | ✗ |
| SFConvCNPs (this work) | $\mathcal{O}(|\Xi|(|\mathcal{D}_c| + |\mathcal{D}_q|))$ | ✓ | ✓ | Frequency | ✗ |
| SFVConvCNPs (this work) | $\mathcal{O}(|\Xi|(|\mathcal{D}_c| + |\mathcal{D}_q|))$ | ✓ | ✓ | Frequency | ✓ |

Notes: ✓ = Exact, ⚠ = Heuristic (no guarantees), ✗ = None. $M$ = number of pseudo-tokens in PT- variants; typically $|\Xi| \ll |\mathcal{G}|$. *Global Interaction*: architectural capacity for any pair of context points to interact, irrespective of separation (the realised *effective* receptive field is studied in Appendix D.4).

## 3.4. Design Comparison and Trade-offs

Table 1 situates the proposed models within existing CNP variants across key design axes. Plain CNPs (Garnelo et al., 2018a) have favorable linear scaling but lack explicit inductive biases and, by aggregating context points independently, admit no pairwise interaction between them (Xu et al., 2020). Attention-based variants—AttnCNPs (Kim et al., 2019), TNPs (Nguyen & Grover, 2022), and their equivariant extension TE-TNPs (Ashman et al., 2024a)—address these limitations but incur prohibitive quadratic complexity. Pseudo-token TNPs (PT-TNPs; Feng et al. 2022) recover linear scaling through latent bottlenecks; however, their translation-equivariant extension TE-PT-TNPs (Ashman et al., 2024a; 2025) relies on heuristic approximations of input translations to shift data back to the training regime. These approximations lack theoretical guarantees and are prone to degeneracies (see Section 3.1 of Ashman et al. 2024a). ConvCNPs (Gordon et al., 2019) attain exact equivariance and linear complexity, but their $\mathcal{O}(|\mathcal{G}|(|\mathcal{D}_c| + |\mathcal{D}_q|))$ cost scales poorly with grid size $|\mathcal{G}|$ in large or high-dimensional domains, and their local kernels preclude global context interaction.

SFConvCNPs and SFVConvCNPs resolve these trade-offs along three axes: guaranteed translation equivariance, $\mathcal{O}(|\Xi|(|\mathcal{D}_c| + |\mathcal{D}_q|))$ complexity, and freedom from spatial discretization. By replacing dense physical grids with a compact frequency grid—where $|\Xi| \ll |\mathcal{G}|$ in practice (Field, 1987; Ruderman & Bialek, 1993; Wainwright & Simoncelli, 1999)—frequency-domain parameterization yields substantial computational savings and scales more readily to higher dimensions, particularly in spatiotemporal settings. It also induces approximately global receptive fields, enabling efficient long-range dependency modeling. SFVConvCNPs go further by providing an explicit Volterra series construction,

which characterizes the parameterized operators analytically as sums of higher-order convolutions. This interpretability is absent from most existing architectures, which rely on opaque stacked non-linear layers that are difficult to analyze.

## 4. Experiments

We evaluate our framework on five regression benchmarks with input dimensionalities $d_{\mathcal{X}} \in \{1, 2, 3\}$, comparing against various members of the CNPs family: the original CNP (Garnelo et al., 2018a), Attentive CNP (AttnCNP; Kim et al. 2019), Convolutional CNP (ConvCNP; Gordon et al. 2019), Spectral ConvCNP (SConvCNP; Mohseni & Duffield 2025), Transformer Neural Process (TNP; Nguyen & Grover 2022), and both the Translation-Equivariant TNP (TE-TNP) and its pseudo-token variant (TE-PT-TNP) (Ashman et al., 2024a). Performance is assessed with two complementary metrics: the predictive log-likelihood, which measures how well the predictive distribution concentrates mass on observed values, and the continuous ranked probability score (CRPS; Matheson & Winkler 1976; Gneiting & Raftery 2007), which evaluates the full distribution via the integrated squared difference between the predicted and empirical CDFs and, for the Gaussian predictives used in this work, admits a closed form (Appendix E). Unless stated otherwise, metrics are computed under each model's Gaussian predictive distribution at query points and reported as mean $\pm$ standard deviation over 5 independent training runs on a common test set. For each metric, the top two results, and any results tied with them, are shown in bold. Appendix E provides full details of datasets, architectures, and training/evaluation protocols.

As discussed in Section 3.2, ConvCNPs become expensive in higher dimensions due to discretizing functional embeddings over dense grids; since our 2D and 3D data already lie

*Table 2.* Predictive log-likelihoods and CRPS on synthetic 1D regression tasks.

| Model | RBF | | Matérn5/2 | | Periodic | | Sawtooth | | Square | |
|---|---|---|---|---|---|---|---|---|---|---|
| | Log-lik.↑ | CRPS↓ | Log-lik.↑ | CRPS↓ | Log-lik.↑ | CRPS↓ | Log-lik.↑ | CRPS↓ | Log-lik.↑ | CRPS↓ |
| CNP | 0.04 ±0.01 | 0.16 ±0.00 | −0.18 ±0.01 | 0.19 ±0.00 | −1.17 ±0.00 | 0.47 ±0.00 | −0.87 ±0.00 | 0.34 ±0.00 | −1.39 ±0.00 | 0.58 ±0.00 |
| AttnCNP | 0.09 ±0.12 | 0.15 ±0.01 | −0.14 ±0.11 | 0.18 ±0.01 | −0.84 ±0.07 | 0.34 ±0.01 | −0.87 ±0.00 | 0.34 ±0.00 | −1.25 ±0.09 | 0.52 ±0.04 |
| TNP | 0.25 ±0.01 | **0.13** ±0.00 | 0.04 ±0.01 | 0.17 ±0.00 | −0.47 ±0.02 | 0.27 ±0.00 | −0.87 ±0.00 | 0.34 ±0.00 | −1.37 ±0.01 | 0.57 ±0.01 |
| TE-TNP | 0.25 ±0.04 | **0.13** ±0.00 | 0.04 ±0.02 | **0.16** ±0.00 | −0.46 ±0.12 | 0.26 ±0.03 | −0.87 ±0.00 | 0.34 ±0.00 | −0.92 ±0.28 | 0.44 ±0.08 |
| TE-PT-TNP | 0.22 ±0.01 | 0.14 ±0.00 | 0.01 ±0.02 | 0.17 ±0.00 | −0.79 ±0.15 | 0.34 ±0.04 | −0.87 ±0.00 | 0.34 ±0.00 | −1.38 ±0.02 | 0.57 ±0.01 |
| ConvCNP | 0.27 ±0.02 | **0.13** ±0.00 | 0.03 ±0.02 | 0.17 ±0.00 | −0.65 ±0.16 | 0.32 ±0.05 | 0.60 ±0.23 | 0.13 ±0.03 | −0.81 ±0.22 | 0.40 ±0.06 |
| SConvCNP | **0.28** ±0.00 | **0.13** ±0.00 | **0.07** ±0.00 | **0.16** ±0.00 | −0.12 ±0.01 | 0.18 ±0.00 | 1.00 ±0.48 | 0.08 ±0.05 | **−0.30** ±0.03 | **0.29** ±0.01 |
| SFConvCNP | **0.29** ±0.00 | **0.13** ±0.00 | **0.07** ±0.00 | **0.16** ±0.00 | −0.04 ±0.01 | **0.17** ±0.00 | 1.19 ±0.01 | 0.06 ±0.00 | −0.36 ±0.08 | 0.30 ±0.01 |
| SFVConvCNP | **0.28** ±0.00 | **0.13** ±0.00 | **0.07** ±0.00 | **0.16** ±0.00 | −0.05 ±0.00 | **0.17** ±0.00 | 1.18 ±0.01 | 0.06 ±0.00 | −0.10 ±0.01 | 0.26 ±0.00 |

on uniform grids, we bypass discretization and include grid-based variants (Grid-ConvCNP and Grid-SConvCNP) that would otherwise be impractical. TE-TNP is excluded from the 2D and 3D benchmarks due to the prohibitive cost of large context and query sets, whereas the non-equivariant AttnCNP and TNP remain tractable through highly optimized scaled dot-product attention kernels (e.g., FlashAttention; Dao et al. 2022; Dao 2023; Shah et al. 2024), for which no equivariant counterparts currently exist. In contrast, SFConvCNP and SFVConvCNP use only standard PyTorch operators, so their scalability reflects the structure of our formulations rather than hardware performance engineering.

### 4.1. One-Dimensional Regression

#### 4.1.1. SYNTHETIC REGRESSION

We evaluate all models on a suite of synthetic regression benchmarks. Each benchmark consists of a finite collection of tasks of the form (1), with input and output spaces $\mathcal{X} = \mathbb{R}$ and $\mathcal{Y} = \mathbb{R}$, respectively. Task-generating latent functions are sampled from five processes: GPs with radial basis function (RBF), Matérn–5/2, and periodic kernels, as well as sawtooth and square-wave generators. Full experimental details are provided in Appendix E.1.1.

Table 2 reports metrics averaged over 1,000 test batches of 64 tasks, with input locations drawn from $\mathcal{U}[−3, 3]$, context points sampled as $|\mathcal{D}_c| \sim \mathcal{U}[5, 50]$, and a fixed query count $|\mathcal{D}_q| = 128$. Across all benchmarks, at least one of SFConvCNP and SFVConvCNP matches or outperforms every competing method, demonstrating strong empirical performance on a diverse set of synthetic processes. On the sawtooth task, all attention-based models (AttnCNP, TNP, TE-TNP, TE-PT-TNP) collapse to the trivial marginal predictor (Figure 3)—each reaching a log-likelihood of $−0.87$, identical across all runs at the reported precision—despite being competitive on GP tasks, whereas every convolution-

based model succeeds, with SFConvCNP reaching 1.19; we conjecture this is a spectral-bias effect (Appendix E.1.1; cf. Mohseni & Duffield 2025). The plain CNP underperforms more broadly, including on sawtooth (Figure 3), owing to its interaction-free mean aggregation (Section 3.4).

A potential concern is that SFVConvCNP performs well without explicit pointwise non-linearities (e.g., ReLU) in its backbone simply because the synthetic tasks are easy, rather than because the Volterra construction captures non-linear mappings. To test this, we replace the pointwise non-linearities in each baseline's backbone with identity mappings (Appendix D.1). As Table 7 shows, this degrades every baseline: TNP and TE-TNP collapse outright despite retaining non-linearity through softmax attention, and only AttnCNP—which likewise retains softmax—escapes with a mild drop. Although part of this degradation may reflect optimization difficulties (Appendix D.1), the collapse of every linearized baseline makes it implausible that the tasks are solvable without an expressive non-linear mapping, supporting the view that the multiplicative interactions of the Volterra construction supply it. The ablation also shows that the linearized SFConvCNP stays competitive with, and sometimes surpasses, the fully non-linear transformer baselines, indicating that the SFConv parameterization's inductive bias alone yields strong performance without backbone non-linearities.

Further ablations in Appendix D examine the geometry and spectral bandwidth of the frequency grid $\Xi$, the rank $R$ of the low-rank Volterra approximation, the effective receptive field (ERF; Luo et al. 2016) across all models, and robustness to translations of the input domain.

#### 4.1.2. PREDATOR–PREY REGRESSION

We assess performance on simulated trajectories from a stochastic Lotka–Volterra predator–prey system (Lotka,

*Table 3.* Predictive log-likelihoods and CRPS on predator–prey tasks. **Sim**: simulated test set; **Real**: Hudson Bay hare–lynx data.

| | Sim | | Real | |
|---|---|---|---|---|
| Model | Log-lik. ↑ | CRPS ↓ | Log-lik. ↑ | CRPS ↓ |
| CNP | −0.39 ±0.00 | 0.21 ±0.00 | −0.21 ±0.00 | 0.16 ±0.00 |
| AttnCNP | −0.06 ±0.14 | 0.16 ±0.02 | −0.06 ±0.07 | 0.14 ±0.01 |
| TNP | 0.26 ±0.22 | 0.14 ±0.02 | −0.01 ±0.06 | **0.13** ±0.01 |
| TE-TNP | 0.36 ±0.10 | 0.13 ±0.01 | **0.03** ±0.03 | **0.13** ±0.00 |
| TE-PT-TNP | 0.17 ±0.14 | 0.15 ±0.01 | −0.03 ±0.04 | 0.14 ±0.01 |
| ConvCNP | 0.34 ±0.09 | 0.13 ±0.01 | 0.02 ±0.02 | **0.13** ±0.00 |
| SConvCNP | **0.45** ±0.00 | **0.12** ±0.00 | 0.03 ±0.00 | **0.13** ±0.00 |
| SFConvCNP | **0.43** ±0.00 | **0.12** ±0.00 | 0.03 ±0.00 | **0.13** ±0.00 |
| SFVConvCNP | 0.39 ±0.00 | **0.12** ±0.00 | 0.04 ±0.00 | **0.13** ±0.00 |

*Table 4.* Predictive log-likelihoods and CRPS on image completion tasks.

| | CIFAR-10 | | SVHN | |
|---|---|---|---|---|
| Model | Log-lik. ↑ | CRPS ↓ | Log-lik. ↑ | CRPS ↓ |
| CNP | 0.96 ±0.06 | 0.06 ±0.00 | 1.72 ±0.02 | 0.03 ±0.00 |
| AttnCNP | 1.57 ±0.02 | 0.04 ±0.00 | 2.74 ±0.05 | **0.01** ±0.00 |
| TNP | 1.66 ±0.07 | 0.04 ±0.00 | 2.89 ±0.02 | **0.01** ±0.00 |
| TE-PT-TNP | 1.57 ±0.04 | 0.04 ±0.00 | 2.68 ±0.12 | **0.01** ±0.00 |
| Grid-ConvCNP | 1.67 ±0.01 | 0.04 ±0.00 | 2.86 ±0.02 | **0.01** ±0.00 |
| Grid-SConvCNP | **1.71** ±0.00 | **0.03** ±0.00 | 2.91 ±0.01 | **0.01** ±0.00 |
| SFConvCNP | **1.74** ±0.00 | **0.03** ±0.00 | **2.93** ±0.00 | **0.01** ±0.00 |
| SFVConvCNP | 1.69 ±0.00 | 0.04 ±0.00 | 2.87 ±0.00 | **0.01** ±0.00 |

1910; Volterra, 1926), following the sim-to-real setup of Bruinsma et al. (2023), modeling prey and predator populations over time ($\mathcal{X} = \mathbb{R}$, $\mathcal{Y} = \mathbb{R}^2$). Models are evaluated under two protocols: (i) a simulated test set from the training distribution, and (ii) sim-to-real transfer to the Hudson Bay hare–lynx dataset (Leigh, 1968) (91 observations, 1845–1935). Full details are provided in Appendix E.1.2, and qualitative example predictions are shown in Figure 4.

Table 3 reports results under both protocols. The simulated set comprises 64,000 test tasks, with context points sampled as $|\mathcal{D}_c| \sim \mathcal{U}[5, 50]$ and a fixed query count $|\mathcal{D}_q| = 128$. The real set instead draws context/query splits from the single Hudson Bay hare–lynx series (91 points), with $|\mathcal{D}_c| \sim \mathcal{U}[5, 50]$ context points and the remaining observations used as queries. On the simulated test set, SConvCNP and SFConvCNP attain the best predictive log-likelihoods, and the SFConv-based models match the lowest CRPS of any method. Under sim-to-real transfer to the Hudson Bay hare–lynx data, SFVConvCNP attains the highest log-likelihood, with SFConvCNP and the strongest baselines close behind and CRPS tightly clustered around 0.13. Overall, our SFConv-based models remain competitive with or ahead of the strongest baselines.

## 4.2. Two-Dimensional Image Regression

We evaluate models on an image completion benchmark. In this setting, each image is viewed as a finite set of samples from an underlying continuous latent function that maps pixel coordinates—row and column indices in $\mathcal{X} = \mathbb{R}^2$—to RGB intensity values, inducing the output space $\mathcal{Y} = \mathbb{R}^3$. We conduct experiments on the CIFAR-10 (Krizhevsky et al., 2009) and SVHN (Netzer et al., 2011) datasets, both of which consist of $32 \times 32$ images. Additional experimental details are provided in Appendix E.2, and qualitative example predictions are shown in Figure 5.

Table 4 reports performance over the test splits (10,000 CIFAR-10 and 26,032 SVHN images), with context points sampled as $|\mathcal{D}_c| \sim \mathcal{U}[5, 512]$ and the remaining pixels used as queries. Across both datasets, SFConvCNP achieves the best predictive log-likelihood, edging out the strongest baseline, Grid-SConvCNP, while SFVConvCNP remains competitive, closely matching or outperforming the transformer-based baselines; predictive CRPS is small and tightly clustered across almost all methods except the plain CNP.

*Table 5.* Predictive log-likelihoods and CRPS on Kolmogorov flow tasks.

| | Kolmogorov Flow | |
|---|---|---|
| Model | Log-lik. ↑ | CRPS ↓ |
| CNP | −1.28 ±0.19 | 0.50 ±0.08 |
| AttnCNP | 0.92 ±0.02 | 0.06 ±0.00 |
| TNP | 0.64 ±0.03 | 0.08 ±0.00 |
| TE-PT-TNP | −0.68 ±0.02 | 0.27 ±0.01 |
| Grid-ConvCNP | **1.73** ±0.06 | **0.03** ±0.00 |
| Grid-SConvCNP | 1.72 ±0.02 | **0.03** ±0.00 |
| SFConvCNP | **1.89** ±0.01 | **0.03** ±0.00 |
| SFVConvCNP | 1.48 ±0.01 | 0.04 ±0.00 |

## 4.3. Three-Dimensional Spatiotemporal Regression

### 4.3.1. KOLMOGOROV FLOW REGRESSION

We study the spatiotemporal dynamics of two-dimensional incompressible Navier–Stokes flow under spatially periodic Kolmogorov forcing (Chandler & Kerswell, 2013; Kochkov et al., 2021). The task is to model the two-component velocity field as a function of space and time ($\mathcal{X} = \mathbb{R}^3$, $\mathcal{Y} = \mathbb{R}^2$), following the experimental setup of Ashman et al. (2024a) with trajectories generated as in Rozet & Louppe (2023).

*Table 6.* Predictive log-likelihoods and CRPS on the ERA5 climate regression task, evaluated on three disjoint regions in Europe: *Center* (in-distribution, ID: the training window), and *West* and *North* (out-of-distribution, OOD).

| Model | Central Europe (ID) | | Western Europe (OOD) | | Northern Europe (OOD) | |
|---|---|---|---|---|---|---|
| | Log-lik. ↑ | CRPS ↓ | Log-lik. ↑ | CRPS ↓ | Log-lik. ↑ | CRPS ↓ |
| CNP | $-0.26$ ±0.01 | 0.20 ±0.00 | $< -10$ | 0.56 ±0.07 | $< -10$ | 0.21 ±0.04 |
| AttnCNP | $-0.20$ ±0.00 | 0.20 ±0.00 | $< -10$ | 0.41 ±0.27 | $-0.17$ ±0.19 | 0.17 ±0.02 |
| TNP | $-0.17$ ±0.01 | 0.19 ±0.00 | $< -10$ | 0.70 ±0.28 | $-1.36$ ±0.86 | 0.27 ±0.04 |
| TE-PT-TNP | $-0.25$ ±0.05 | 0.20 ±0.01 | $-0.08$ ±0.06 | 0.18 ±0.01 | 0.34 ±0.09 | 0.12 ±0.01 |
| SFConvCNP | **1.55** ±0.01 | **0.03** ±0.00 | **1.71** ±0.01 | **0.03** ±0.00 | **1.89** ±0.01 | **0.03** ±0.00 |
| SFVConvCNP | **1.09** ±0.01 | **0.05** ±0.00 | **1.18** ±0.02 | **0.05** ±0.00 | **1.34** ±0.01 | **0.04** ±0.00 |

Additional details are provided in Appendix E.3.1, and qualitative example predictions are shown in Figure 6.

Table 5 reports performance over a fixed test set of 6,592 tasks, formed by deterministically tiling the 103 test trajectories into non-overlapping $16 \times 16 \times 16$ space–time crops so that every location is evaluated exactly once, with context fraction sampled as $|\mathcal{D}_c|/|\mathcal{D}| \sim \mathcal{U}[0.05, 0.25]$. SF-ConvCNP performs best, with a predictive log-likelihood of 1.89 and the lowest CRPS (0.03), demonstrating the effectiveness of set Fourier convolutions for modeling complex spatiotemporal dynamics. The Grid-ConvCNP and Grid-SConvCNP baselines follow, with log-likelihoods of 1.73 and 1.72 and matching CRPS. SFVConvCNP reaches 1.48, comfortably exceeding every non-convolutional baseline but still trailing SFConvCNP. We attribute this gap to a parameter-budget constraint rather than the Volterra construction itself: each SFVConvCNP layer contains $2R + 1$ SFConv modules (5 at the rank $R = 2$ used here), versus one per SFConvCNP layer. At a matched per-module configuration SFVConvCNP would therefore be far larger, so to keep the total parameter count comparable across models (Table 14) we shrank each SFConv module substantially, reducing the embedding width from 176 to 52 and the feedforward width from 512 to 208. We hypothesize that this much smaller per-module capacity, rather than the Volterra parameterization, is what limits SFVConvCNP relative to SFConvCNP on this task.

### 4.3.2. ERA5 CLIMATE REGRESSION

We evaluate on a climate regression task from the ERA5 reanalysis dataset (Hersbach et al., 2020), following Ashman et al. (2024a): predicting the air temperature 2 m above the surface from time and geographic location ($\mathcal{X} = \mathbb{R}^3$, $\mathcal{Y} = \mathbb{R}$). To probe spatial generalization, we train on a central European window and evaluate on three regions: Center (the training window, in-distribution) and two windows disjoint from training, West and North, shifted in longitude and latitude respectively. Appendix E.3.2 gives full details, and Figure 7 shows example predictions.

Table 6 reports per-region performance, averaged over 16,000 test tasks formed by randomly cropping $6 \times 16 \times 16$ (time $\times$ latitude $\times$ longitude) subgrids, with context fraction $|\mathcal{D}_c|/|\mathcal{D}| \sim \mathcal{U}[0.05, 0.33]$ and the remaining points as queries. We omit Grid-ConvCNP and Grid-SConvCNP, which we could not train successfully despite trying a range of model sizes. SFConvCNP leads in every region by a wide margin, with SFVConvCNP second; as in the Kolmogorov experiment (Section 4.3.1), this gap likely reflects the smaller per-module configuration SFVConvCNP needs to match parameter counts. The key comparison, however, is between equivariant and non-equivariant models across regions. Both of our models and TE-PT-TNP generalize to the disjoint West and North regions with no loss relative to in-distribution Center. The non-equivariant baselines behave very differently: already trailing on Center ($-0.26$, $-0.20$, and $-0.17$ for CNP, AttnCNP, and TNP), their log-likelihoods collapse below $-10$ on West (and on North for CNP), with CRPS worsening in step (e.g., CNP from 0.20 to 0.56). These patterns share one explanation. The non-equivariant models rely on absolute input positions, which no longer match the training distribution once the region shifts. The equivariant models, by contrast, depend only on relative displacements, so their predictive competence transfers across regions; the residual variation in their scores tracks regional difficulty rather than positional mismatch.

## 5. Conclusion

We presented a principled, scalable framework for translation-equivariant conditional neural processes. Approximating equivariant operators through Volterra series makes the representational structure of convolutional deep sets analytically transparent. To overcome the scalability limits of grid-based methods, we introduced set Fourier convolutions, which operate directly on irregular inputs, induce approximately global receptive fields, and retain linear complexity. Building on these, we proposed SFConvCNPs and SFVConvCNPs, which achieve strong empirical performance on both synthetic and real-world experiments.

## Acknowledgements

We gratefully acknowledge the Texas A&M High Performance Research Computing facility for providing the computational resources that made this study possible. We also thank the anonymous reviewers for their suggestions, which helped improve the quality of this work.

## Impact Statement

This work introduces frequency-domain neural processes for efficient probabilistic inference on irregularly sampled spatiotemporal data, combining linear scalability with global interaction across observations and interpretable Volterra-based operators. We anticipate primarily positive societal impact through improved modeling of physical systems, such as in climate applications where existing methods face computational bottlenecks.

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

# A. Related Works

Conditional neural processes (CNPs; Garnelo et al. 2018a) recast posterior predictive inference as a variational problem: instead of specifying a prior and invoking Bayes' rule, they learn a direct mapping from a set of observations (the context set) to an approximate posterior predictive distribution (Bruinsma et al., 2021; Bruinsma, 2024). Subsequent work has refined this pipeline along several axes. Most prominently, the mean-field Gaussian predictive of vanilla CNPs ignores dependencies across query locations and is confined to Gaussian distributions, a limitation that has motivated an extensive line of research. Proposed remedies include latent-variable models (Garnelo et al., 2018b; Louizos et al., 2019; Wang & Van Hoof, 2020; Foong et al., 2020; Lee et al., 2020; Cotton et al., 2020; Wang et al., 2022; Wang & van Hoof, 2022; Kim et al., 2022; Jung & Park, 2023; Lee et al., 2023; Xu et al., 2023; Lee et al., 2025), autoregressive rollouts (Bruinsma et al., 2023; Nguyen & Grover, 2022; Hassan et al., 2026), full-covariance parameterizations (Bruinsma et al., 2021; Markou et al., 2022), implicit quantile networks for distribution-free per-query predictions (Mohseni et al., 2023), and diffusion- and flow-based variants (Dutordoir et al., 2023; Mathieu et al., 2023; Hamad & Rosenbaum, 2025).

A parallel line of work equips NPs with structural inductive biases that capture symmetries common in scientific and physical data (Gordon et al., 2019; Kawano et al., 2021; Holderrieth et al., 2021; Wang et al., 2021; Huang et al., 2023; Ashman et al., 2024a;b; Allen et al., 2025). Translation equivariance, in particular, has been pursued along two main routes: convolutional architectures, exemplified by ConvCNPs (Gordon et al., 2019), and transformer architectures, exemplified by TE-TNPs (Ashman et al., 2024a). Both, however, encounter scaling bottlenecks. ConvCNPs discretize inputs onto a dense uniform grid before applying CNNs, which scales poorly as dimensionality grows. Attention-based NPs sidestep this discretization but inherit the quadratic cost of full self-attention. For non-equivariant TNPs, this cost can be substantially alleviated by optimized implementations such as FlashAttention (Dao et al., 2022; Dao, 2023; Shah et al., 2024). The bottleneck resurfaces, however, once translation equivariance is imposed: TE-TNP (Ashman et al., 2024a) combines pairwise positional encodings with feature dot products through an MLP, an operation that no existing FlashAttention variant currently supports.

The prevailing remedy in the NP literature is the pseudo-token (inducing-point) paradigm (Feng et al., 2022), rooted in the Set Transformer (Lee et al., 2019) and the Perceiver (Jaegle et al., 2021b;a), which routes information through a fixed set of learnable tokens. Its translation-equivariant counterpart, TE-PT-TNP (Huang et al., 2023; Ashman et al., 2024a), restores linear complexity via equivariant inducing-point cross-attention, but introduces a degeneracy specific to the equivariant case (Ashman et al., 2024a, §3.1). The same degeneracy afflicts the more recent Gridded TNPs (Ashman et al., 2025), which project unstructured inputs onto a structured grid of pseudo-tokens (He et al., 2022) and process them with efficient vision transformers (Dosovitskiy et al., 2021; Liu et al., 2021). These shared limitations point to a substantial opportunity: integrating other efficient-transformer variants into TNPs (Wang et al., 2020b; Choromanski et al., 2021; Cao, 2021; He et al., 2023; Zhdanov et al., 2025). Not every such variant transfers cleanly, however; Cosformer (Qin et al., 2022), for instance, scales near-linearly on 1-D sequences but does not extend readily to the unordered, multi-dimensional point sets arising in NP problems. Identifying designs that handle variable point sets efficiently while respecting structural inductive biases therefore remains an active research direction.

# B. Softmax Attention as a Non-linear Integral Operator

The first-order convolution in (7) is linear in the input function $h$ and hence straightforward to analyze. Following the success of transformer architectures (Vaswani et al., 2017)—driven primarily by softmax attention (Bahdanau, 2014; Jean et al., 2015; Luong et al., 2015; Wu et al., 2016)—there has been growing interest in non-linear constructions, despite their being harder to analyze. Here we cast softmax attention as one such construction, writing it as a non-linear integral operator over $h$.

Softmax attention computes pairwise similarity scores, normalizes them via a softmax, and updates each query by aggregating value embeddings using these normalized scores. To formalize this construction, we define the token embedding

$$v_h : \mathcal{X} \to \mathcal{X} \times \mathcal{Z}, \qquad v_h(x) := (x, h(x)).$$

Let $\gamma : (\mathcal{X} \times \mathcal{Z})^2 \to \mathbb{R}$ be a similarity scoring function, which may be asymmetric, nonpositive, or highly non-linear. We define the induced, $h$-dependent similarity

$$\gamma_h(x, x') := \gamma(v_h(x), v_h(x')).$$

The continuous softmax attention operator can then be written as

$$\mathcal{A}[h](x) = \int_{\mathcal{X}} \alpha_h(x, x') \, h(x') \, \mathrm{d}x',\tag{11}$$

where the attention weights are defined by the normalized kernel

$$\alpha_h(x, x') = \frac{e^{\gamma_h(x, x')}}{\int_{\mathcal{X}} e^{\gamma_h(x, x'')} \, \mathrm{d}x''}.\tag{12}$$

Unlike first-order convolution, the operator $\mathcal{A}$ is non-linear in $h$, since the kernel $\alpha_h$ depends globally on $h$ through the similarity function $\gamma_h$. This global, data-dependent normalization fundamentally distinguishes softmax attention from linear integral operators with fixed kernels.

In practice, the integrals in (11) and (12) are approximated using a Monte Carlo–style estimator based on a finite set of samples from the input function $h$ (Calvello et al., 2024). Specifically, given a collection of sample locations $\{x_k\}_{k=1}^N \subset \mathcal{X}$, the integrals are replaced by empirical sums over these samples, yielding the discrete approximation

$$\mathcal{A}[h](x) \approx \frac{\sum_{k=1}^N e^{\gamma_h(x, x_k)} \, h(x_k)}{\sum_{k=1}^N e^{\gamma_h(x, x_k)}}.$$

Importantly, unlike practical implementations of convolutional operators—which typically require the input function to be discretized over a uniform grid—this approximation of softmax attention operates directly on an unordered set of samples. Softmax attention therefore extends naturally to unstructured data and irregularly sampled domains, with no underlying lattice or fixed spatial discretization. Evaluating the similarity between each query and all other points, however, incurs quadratic complexity in the number of samples—a major drawback of transformers relative to the linear complexity of convolutional operators with fixed kernels.

## C. Volterra Series for Vector-Valued Signals

Section 3.1 presents the Volterra expansion for scalar-valued signals ($\mathcal{Z} = \mathbb{R}$), whereas the functional embedding of the convolutional deep set representation is vector-valued: $\mathcal{Z} = \mathbb{R}^{1+d_{\mathcal{Y}}}$ in the convention of (2), and $\mathcal{Z} = \mathbb{R}^{2d_{\mathcal{Y}}}$ in our implementation (Appendix E). This section records the general construction and verifies that every step of Section 3.1—the nested (cascade) decomposition, the low-rank approximation, and translation equivariance—carries over with only notational changes. Throughout, $\mathcal{Z} = \mathbb{R}^{c_{\mathrm{in}}}$ denotes the input space of the operator $\rho_d$ and $\mathcal{Z}' = \mathbb{R}^{c_{\mathrm{out}}}$ its output space; we write $[d] := \{1, \ldots, d\}$, and parenthesized superscripts index channels, e.g., $h(s) = \big(h^{(1)}(s), \ldots, h^{(c_{\mathrm{in}})}(s)\big)$.

**Higher-Order Terms.** For each $l \in \mathbb{N}$, the $l$-th order kernel is a map $\kappa_l : \mathcal{X}^l \to \mathrm{L}\big(\mathcal{Z}^{\otimes l}, \mathcal{Z}'\big)$ into the space of multilinear operators from $\mathcal{Z}^{\otimes l}$ to $\mathcal{Z}'$, with every component absolutely integrable, and the $l$-th order convolution becomes

$$\mathcal{C}_l[h](x) = \int_{\mathcal{X}^l} \kappa_l(x - s_{1:l}) \big[h(s_1) \otimes \cdots \otimes h(s_l)\big] \, \mathrm{d}s_{1:l},\tag{13}$$

with the bias $\mathcal{C}_0 = \kappa_0 \in \mathcal{Z}'$ and $\rho_d[h] = \mathcal{C}_0 + \sum_{l \geq 1} \mathcal{C}_l[h]$ as in (3). In coordinates, for each output channel $j \in [c_{\mathrm{out}}]$,

$$\big(\mathcal{C}_l[h](x)\big)^{(j)} = \sum_{a_{1:l} \in [c_{\mathrm{in}}]^l} \int_{\mathcal{X}^l} \kappa_l^{(j; \, a_{1:l})}(x - s_{1:l}) \prod_{k=1}^l h^{(a_k)}(s_k) \, \mathrm{d}s_{1:l},\tag{14}$$

where $\kappa_l^{(j; \, a_{1:l})}$ denotes the corresponding component of $\kappa_l$. For $c_{\mathrm{in}} = c_{\mathrm{out}} = 1$, Equation (14) reduces to the scalar form of Section 3.1; each $\mathcal{C}_l$ remains multilinear in $h$.

**Nested (Cascade) Form.** For a multilinear operator $T \in \mathrm{L}\big(\mathcal{Z}^{\otimes(l+1)}, \mathcal{Z}'\big)$ and a vector $z \in \mathcal{Z}$, let $T \lrcorner z \in \mathrm{L}\big(\mathcal{Z}^{\otimes l}, \mathcal{Z}'\big)$ denote contraction of the last slot,

$$(T \lrcorner z)\big[z_1 \otimes \cdots \otimes z_l\big] = T\big[z_1 \otimes \cdots \otimes z_l \otimes z\big], \qquad \text{i.e., in coordinates} \qquad (T \lrcorner z)^{(j; \, a_{1:l})} = \sum_{a \in [c_{\mathrm{in}}]} T^{(j; \, a_{1:l}, \, a)} \, z^{(a)}.$$

The auxiliary operators $\mathcal{V}_l[h](x, s_{1:l}) \in \mathrm{L}(\mathcal{Z}^{\otimes l}, \mathcal{Z}')$ are then defined by the recursion

$$\mathcal{V}_l[h](x, s_{1:l}) = \kappa_l(x - s_{1:l}) + \int \mathcal{V}_{l+1}[h](x, s_{1:l+1}) \lrcorner\, h(s_{l+1})\, \mathrm{d}s_{l+1}, \tag{15}$$

the exact analogue of (4), and the expansion takes the nested form

$$\rho_d[h](x) = \mathcal{C}_0 + \int \mathcal{V}_1[h](x, s_1)\, h(s_1)\, \mathrm{d}s_1,$$

where $\mathcal{V}_1[h](x, s_1) \in \mathrm{L}(\mathcal{Z}, \mathcal{Z}') \cong \mathbb{R}^{c_{\mathrm{out}} \times c_{\mathrm{in}}}$ acts on $h(s_1)$ by matrix–vector multiplication. All integrands are multilinear in finitely many evaluations of $h$ and componentwise absolutely integrable, so Fubini's theorem applies coordinatewise exactly as in the scalar case: contracting (15) against $h(s_l)$ in its last free slot and integrating over $s_l$ yields the analogue of (5), namely the first-order term $\int \kappa_l(x - s_{1:l}) \lrcorner\, h(s_l)\, \mathrm{d}s_l$ plus a second-order term $\mathcal{Q}_{l+1}[h](x; s_{1:l-1})$ in which the pair $(s_l, s_{l+1})$ is integrated against $h(s_l) \otimes h(s_{l+1})$.

**Low-Rank Decomposition.** The Hilbert–Schmidt argument of (6) applies after bundling each integration variable with its channel index. Fix $x$, $s_{1:l-1}$, an output channel $j \in [c_{\mathrm{out}}]$, and leading channel indices $a_{1:l-1} \in [c_{\mathrm{in}}]^{l-1}$, and regard the remaining dependence of $\mathcal{V}_{l+1}$ on $(s_l, a_l)$ and $(s_{l+1}, a_{l+1})$ as a scalar kernel

$$K\big((s_l, a_l), (s_{l+1}, a_{l+1})\big) := \mathcal{V}_{l+1}^{(j;\, a_{1:l+1})}[h](x, s_{1:l+1})$$

on $\big(\mathcal{X} \times [c_{\mathrm{in}}]\big)^2$. Assuming $K$ is square-integrable with respect to the product of the Lebesgue measure on $\mathcal{X}$ and the counting measure on $[c_{\mathrm{in}}]$—the direct analogue of the $L^2(\mathcal{X} \times \mathcal{X})$ assumption of Section 3.1—the Schmidt decomposition on $L^2\big(\mathcal{X} \times [c_{\mathrm{in}}]\big) \cong L^2(\mathcal{X}; \mathcal{Z})$ yields

$$K\big((s_l, a_l), (s_{l+1}, a_{l+1})\big) = \sum_{r=1}^{\infty} \lambda_r\, \Upsilon_r^{(1)}(s_l, a_l)\, \Upsilon_r^{(2)}(s_{l+1}, a_{l+1}), \tag{16}$$

with orthonormal families $\{\Upsilon_r^{(1)}\}_r$ and $\{\Upsilon_r^{(2)}\}_r$ in $L^2(\mathcal{X}; \mathcal{Z})$ and singular values $\lambda_r \geq 0$ with $\sum_{r \geq 1} \lambda_r^2 < \infty$; as in (6), all quantities depend on $h$, $x$, and $s_{1:l-1}$—and here additionally on $(j, a_{1:l-1})$—which we suppress for readability. Substituting (16) into $\mathcal{Q}_{l+1}$ and truncating after $R$ terms expresses each summand as $\lambda_r$ times the product of two integrals of the form

$$\int_{\mathcal{X}} \sum_{a \in [c_{\mathrm{in}}]} \Upsilon_r^{(i)}(s, a)\, h^{(a)}(s)\, \mathrm{d}s, \qquad i \in \{1, 2\},$$

i.e., first-order convolutions of the vector-valued signal $h$ in which the channel sum is absorbed into a matrix–vector product; stacking over the output channels $j \in [c_{\mathrm{out}}]$ yields convolutions with matrix-valued kernels, exactly of the form (7). The operation count of Section 3.1 is therefore unchanged: a rank-$R$ truncation costs $2R + 1$ first-order (now multi-channel) convolutions per recursion step.

**Translation Equivariance.** As in the scalar case, every kernel above is evaluated only at relative displacements of the form $x - s_k$; the channel structure enters solely through the displacement-independent multilinear action on $\mathcal{Z}^{\otimes l}$. Translation equivariance of the (truncated) expansion is therefore unaffected by the passage to vector-valued signals.

**Relation to the Implementation.** The SFVConvCNP parameterization (Appendix E.1.1) instantiates a structured subclass of the rank-$R$ construction above. Each of the $2R + 1$ branches is a first-order SFConv with matrix-valued spectral weights $W_\xi$, so cross-channel mixing is absorbed into the convolutions themselves; the element-wise products $z_r^{(1)} \odot z_r^{(2)}$ realize, per output channel, the products of paired convolutions in the truncated decomposition; and the learnable coefficients $\{\alpha_r\}_{r=1}^{R}$ play the role of the singular values $\lambda_r$. As in the scalar case (Section 3.1), neither the orthonormality of $\{\Upsilon_r^{(i)}\}_r$ nor the nonnegativity of the singular values is enforced during learning, and grouped spectral weights further restrict the class of channel mixings in exchange for parameter efficiency.

# D. Ablation Studies

We present ablation studies isolating the impact of key architectural components—non-linear activation functions, frequency-grid geometry and spectral bandwidth (Section 3.2), and the rank of the low-rank Volterra approximation (Section 3.1)—together with two diagnostic analyses of the trained models: their effective receptive field (Appendix D.4) and their robustness to translations of the input domain (Appendix D.5). Due to computational constraints, the quantitative ablations report mean ± standard deviation over two independent training runs rather than the five used in Section 4. We consider this sufficient because the results in that section were empirically stable, with consistent trends across seeds.

## D.1. Effects of Removing Non-linear Activations

This ablation studies the role of non-linear activation functions in the model architectures we consider, assessing whether the regression tasks remain solvable when expressivity is deliberately constrained by removing non-linearities. For each architecture, we replace the non-linear activations with identity mappings and keep all other components of the pipeline fixed. Data generation, optimization hyperparameters, training procedures, and evaluation protocols match those in Appendix E.1.1; architectural details are given in Appendix E.1.1. The components modified in each model are:

- **CNP**: all activations in the 6-layer MLP of the Deep Set applied to the concatenated context embeddings.

- **AttnCNP**: all activations in the self-attention layers for context processing and in the cross-attention layer between contexts and queries.

- **TNP**: all activations in the transformer layers.

- **TE-TNP**: all activations in the transformer layers and in the MLP kernels within the translation-equivariant attention modules that map pairwise location differences and dot products to attention logits.

- **TE-PT-TNP**: as in TE-TNP, all activations in the transformer layers and in the MLP kernels within the translation-equivariant attention modules.

- **ConvCNP**: all activations in the U-Net backbone.

- **SConvCNP**: all activations in the FNO backbone.

- **SFConvCNP**: all activations in the SFConvBlocks.

Three caveats are worth noting. First, the ablation linearizes only the backbone: the shared token-embedding and decoder MLPs, as well as layer normalization where present, remain non-linear in every model, so the resulting models are not fully linear maps. Second, removing non-linearities from the ConvCNP U-Net and SConvCNP FNO backbones causes severe training instability and consistent optimization failure. Third, sawtooth results are meaningful only for SFConvCNP: every other model in Table 7 already collapses to the trivial marginal predictor on this task even with non-linearities retained (see Table 2 and Figure 3), and remains at that collapse point when linearized.

*Table 7.* Predictive log-likelihoods and CRPS on synthetic regression tasks for models with reduced nonlinearity. On Sawtooth, every model except SFConvCNP collapses to the trivial marginal predictor (see text); these degenerate entries are not highlighted.

| Model | Matérn5/2 | | Periodic | | Sawtooth | |
|---|---|---|---|---|---|---|
| | Log-lik. ↑ | CRPS ↓ | Log-lik. ↑ | CRPS ↓ | Log-lik. ↑ | CRPS ↓ |
| CNP | −1.29 ±0.00 | 0.51 ±0.00 | −1.17 ±0.00 | 0.47 ±0.00 | −0.87 ±0.00 | 0.34 ±0.00 |
| AttnCNP | **−0.21** ±0.00 | **0.19** ±0.00 | **−0.90** ±0.02 | **0.35** ±0.00 | −0.87 ±0.00 | 0.34 ±0.00 |
| TNP | −1.42 ±0.00 | 0.57 ±0.00 | −1.42 ±0.00 | 0.57 ±0.00 | −0.87 ±0.00 | 0.34 ±0.00 |
| TE-TNP | −1.42 ±0.00 | 0.57 ±0.00 | −1.42 ±0.00 | 0.57 ±0.00 | −0.87 ±0.00 | 0.34 ±0.00 |
| TE-PT-TNP | −1.35 ±0.08 | 0.54 ±0.03 | −1.17 ±0.01 | 0.47 ±0.00 | −0.87 ±0.00 | 0.34 ±0.00 |
| SFConvCNP | **0.06** ±0.00 | **0.16** ±0.00 | **−0.32** ±0.00 | **0.22** ±0.00 | **−0.04** ±1.18 | **0.22** ±0.17 |

As Table 7 shows, removing non-linear activations degrades performance markedly for most models: CNP and the TNP-family models (TNP, TE-TNP, TE-PT-TNP) collapse on all three tasks, whereas AttnCNP—whose softmax attention still supplies non-linearity—suffers only a mild drop. Non-linearities are thus essential not only for expressive capacity but also for stable, effective optimization, underscoring their central role in modeling complex functional relationships.

### D.2. Effects of the Frequency Grid

As described in Section 3.2, our implementation directly parameterizes the Fourier transform of the convolution kernel, denoted by $\widehat{\kappa}(\cdot)$ in (8), over a finite frequency grid $\Xi$. The choice of this grid determines which spectral components the model can represent and therefore plays a critical role in predictive performance.

Throughout this work, we employ symmetric, uniformly spaced frequency grids of the form

$$\Xi = \big\{ -n_{\mathcal{F}}\Delta_{\mathcal{F}}, \ldots, -\Delta_{\mathcal{F}}, 0, \Delta_{\mathcal{F}}, \ldots, n_{\mathcal{F}}\Delta_{\mathcal{F}} \big\},$$

where $\Delta_{\mathcal{F}}$ is the grid spacing (resolution), $\xi_{\max}$ is an upper frequency cutoff, and $n_{\mathcal{F}} = \lfloor \xi_{\max}/\Delta_{\mathcal{F}} \rfloor$ is the number of positive frequencies per axis. The grid therefore comprises $|\Xi| = 2n_{\mathcal{F}} + 1$ frequencies (the $n_{\mathcal{F}}$ positive frequencies, their negatives, and zero), with largest attained frequency $n_{\mathcal{F}}\Delta_{\mathcal{F}} = \xi_{\max}$, i.e., the cutoff itself. The grid is controlled by two knobs: its *reach* $\xi_{\max}$ and its *resolution* $\Delta_{\mathcal{F}}$. These cannot be varied independently while holding the grid size $|\Xi|$—and hence the number of learnable spectral weights—fixed, since for fixed $|\Xi|$ they are tied through $n_{\mathcal{F}} = \lfloor \xi_{\max}/\Delta_{\mathcal{F}} \rfloor$. We therefore probe the grid with a pair of complementary ablations, in each case using SFVConvCNP and taking the configuration $(\xi_{\max}, \Delta_{\mathcal{F}}) = (4.9, 0.1)$—used in the main experiments of Section 4.1.1, with $|\Xi| = 99$—as the reference point (implementation details in Appendix E.1.1). The first fixes $|\Xi|$ and trades resolution against reach; the second fixes $\Delta_{\mathcal{F}}$ and lets $|\Xi|$, and thus the parameter count, grow with $\xi_{\max}$.

**Fixed grid size: resolution versus reach.** We first vary $(\xi_{\max}, \Delta_{\mathcal{F}})$ while keeping the grid size $|\Xi|$ fixed, so that the total number of learnable parameters is constant across configurations and changes in performance cannot be attributed to model capacity. Because $|\Xi|$ is fixed, extending the reach $\xi_{\max}$ necessarily coarsens the resolution $\Delta_{\mathcal{F}}$, and vice versa.

*Table 8.* Predictive log-likelihoods and CRPS for different frequency grid truncations on synthetic regression tasks. The grid size $|\Xi|$ is kept fixed across configurations, and only the maximum frequency $\xi_{\max}$ and spacing $\Delta_{\mathcal{F}}$ are varied.

| | Matérn5/2 | | Periodic | | Sawtooth | |
|---|---|---|---|---|---|---|
| $(\xi_{max}, \Delta_{\mathcal{F}})$ | Log-lik. ↑ | CRPS ↓ | Log-lik. ↑ | CRPS ↓ | Log-lik. ↑ | CRPS ↓ |
| (0.98, 0.02) | 0.04 ±0.00 | 0.17 ±0.00 | −0.51 ±0.03 | 0.26 ±0.01 | −0.40 ±0.24 | 0.26 ±0.04 |
| (2.45, 0.05) | 0.06 ±0.00 | **0.16** ±0.00 | −0.16 ±0.00 | 0.19 ±0.00 | 0.57 ±0.12 | 0.12 ±0.01 |
| (4.9, 0.1) | **0.07** ±0.00 | 0.16 ±0.00 | **−0.05** ±0.00 | **0.17** ±0.00 | **1.18** ±0.00 | **0.06** ±0.00 |
| (7.35, 0.15) | **0.07** ±0.00 | 0.16 ±0.00 | −0.10 ±0.01 | 0.18 ±0.00 | 1.14 ±0.01 | 0.07 ±0.00 |
| (9.8, 0.2) | −0.40 ±0.00 | 0.26 ±0.00 | −0.44 ±0.00 | 0.25 ±0.00 | 0.47 ±0.01 | 0.13 ±0.00 |
| (14.7, 0.3) | −0.91 ±0.00 | 0.39 ±0.00 | −0.82 ±0.00 | 0.35 ±0.00 | −0.27 ±0.00 | 0.23 ±0.00 |
| (19.6, 0.4) | −1.07 ±0.00 | 0.44 ±0.00 | −0.93 ±0.00 | 0.39 ±0.00 | −0.45 ±0.00 | 0.26 ±0.00 |

The results are summarized in Table 8. Across all three synthetic regression tasks, predictive performance is maximized for intermediate cutoffs, with the best results around $\xi_{\max} \in [4.9, 7.35]$, and falls off at both extremes for distinct reasons. Restricting the grid to very low frequencies ($\xi_{\max} \le 2.45$) removes higher-frequency structure necessary to represent the target functions accurately. The degradation at large cutoffs ($\xi_{\max} \ge 9.8$), however, should not be read as evidence that high frequencies are intrinsically harmful: since $|\Xi|$ is held fixed, raising $\xi_{\max}$ coarsens $\Delta_{\mathcal{F}}$ (up to $\Delta_{\mathcal{F}} = 0.4$ at $\xi_{\max} = 19.6$), and it is the resulting loss of spectral resolution in the low- and mid-frequency bands—where the targets concentrate most of their energy—that drives the degradation. The fixed-resolution ablation below confirms this reading, adding the very same high frequencies without sacrificing resolution and finding no such degradation.

Notably, early spectral truncation affects the sawtooth task more severely than the GP-based targets. For small frequency cutoffs ($\xi_{\max} \le 2.45$), performance on the sawtooth data degrades substantially relative to the Matérn and Periodic cases. This behavior is consistent with classical Fourier analysis: the Fourier coefficients of a sawtooth wave decay only

algebraically (as $\mathcal{O}(1/|\xi|)$), whereas samples from smooth or periodic GPs exhibit significantly faster spectral decay. As a result, accurately modeling the sharp transitions present in the sawtooth function requires retaining higher-frequency components, making the model particularly sensitive to aggressive low-frequency truncation in this setting.

**Fixed resolution: spectral bandwidth and capacity.** The complementary question is whether adding spectral parameters helps or instead causes overfitting. We now fix the resolution $\Delta_\mathcal{F}=0.1$ and increase the cutoff $\xi_{\max}$ from 0.9 to 19.9, so that $|\Xi| = 2\lfloor \xi_{\max}/\Delta_\mathcal{F} \rfloor + 1$ grows from 19 to 399 points and the learnable spectral weights grow proportionally, with all other hyperparameters at their default SFVConvCNP values (Appendix E.1.1). This isolates spectral bandwidth, and the associated capacity, from all other architectural choices.

*Table 9.* Predictive log-likelihoods and CRPS for SFVConvCNP with fixed $\Delta_\mathcal{F} = 0.1$ and increasing $\xi_{\max}$ on synthetic regression tasks. The frequency grid $|\Xi|$ and number of parameters grow with $\xi_{\max}$.

| $\xi_{\max}$ | $|\Xi|$ | Matérn5/2 | | Periodic | | Sawtooth | |
|---|---|---|---|---|---|---|---|
| | | Log-lik. ↑ | CRPS ↓ | Log-lik. ↑ | CRPS ↓ | Log-lik. ↑ | CRPS ↓ |
| 0.9 | 19 | 0.03 ±0.00 | 0.17 ±0.00 | −0.54 ±0.02 | 0.26 ±0.00 | −0.60 ±0.00 | 0.30 ±0.00 |
| 2.4 | 49 | 0.06 ±0.00 | **0.16** ±0.00 | −0.20 ±0.01 | 0.20 ±0.00 | 0.63 ±0.09 | 0.12 ±0.01 |
| 4.9 | 99 | **0.07** ±0.00 | **0.16** ±0.00 | −0.05 ±0.00 | **0.17** ±0.00 | 1.18 ±0.00 | **0.06** ±0.00 |
| 7.4 | 149 | **0.07** ±0.00 | **0.16** ±0.00 | −0.04 ±0.01 | **0.17** ±0.00 | **1.19** ±0.00 | **0.06** ±0.00 |
| 9.9 | 199 | **0.07** ±0.00 | **0.16** ±0.00 | −0.02 ±0.00 | **0.17** ±0.00 | 1.17 ±0.00 | **0.06** ±0.00 |
| 14.9 | 299 | **0.07** ±0.00 | **0.16** ±0.00 | −0.01 ±0.00 | **0.17** ±0.00 | **1.19** ±0.02 | **0.06** ±0.00 |
| 19.9 | 399 | **0.07** ±0.00 | **0.16** ±0.00 | −0.01 ±0.00 | **0.17** ±0.00 | **1.19** ±0.00 | **0.06** ±0.00 |

Table 9 shows performance improving up to $\xi_{\max} \approx 4.9$–7.4 and then plateauing, with no degradation even at $\xi_{\max} = 19.9$. Once the energy-bearing bands are covered, the added high-frequency capacity is neither helpful nor harmful—SFVConvCNP leaves it effectively unused rather than overfitting—so the model scales gracefully with spectral bandwidth.

**Resolution, not reach, is the binding factor.** Read together, the two ablations disentangle resolution from reach. With $\Delta_\mathcal{F}$ fixed, extending $\xi_{\max}$ to add high-frequency bands leaves performance essentially flat once $\xi_{\max}$ is large enough to cover the energy-bearing range; with $|\Xi|$ fixed, coarsening $\Delta_\mathcal{F}$ to reach the same cutoff degrades it sharply. The sharp degradation in the first ablation is therefore attributable to the lost resolution, not to the high frequencies themselves. Provided $\xi_{\max}$ is large enough to span the energy-bearing bands, predictive performance is governed primarily by the spectral resolution $\Delta_\mathcal{F}$ retained in those bands rather than by the maximum frequency $\xi_{\max}$.

### D.3. Effects of the Rank of the Low-Rank Volterra Approximation

Following the ablation on the geometry of the frequency grid (Appendix D.2), we next examine a complementary source of inductive bias in our model: the expressiveness of the low-rank approximation used in the Volterra representation introduced in Section 3.1.

We briefly recall the relevant construction. Each recursion level of the Volterra expansion—corresponding to a single layer of SFVConvCNP—consists of one first-order convolution and one second-order convolution. Approximating the second-order Volterra term with a rank-$R$ decomposition expresses it as a sum of $R$ separable components, where each component is implemented as the product of two first-order convolutions. Consequently, a rank-$R$ approximation results in $2R + 1$ first-order convolutions per layer.

A natural ablation strategy would be to increase the rank $R$ while keeping all other architectural choices fixed. However, this would increase the total number of learnable parameters, making it difficult to disentangle the effect of Volterra rank from changes in model capacity. To enable a fair comparison, we instead follow the same principle as in the frequency-grid ablation and control for parameter count. Specifically, we take the SFVConvCNP configuration used in the main experiments of Section 4.1.1, with $(\xi_{\max}, \Delta_\mathcal{F}) = (4.9, 0.1)$, as a reference point, with implementation details provided in Appendix E.1.1. As the rank $R$ increases, we adjust the frequency resolution $\Delta_\mathcal{F}$ of the grid $\Xi$ so that the overall number of parameters remains approximately constant across configurations.

Table 10 summarizes the results on the synthetic regression benchmarks. Performance is stable across small to moderate ranks ($R \in [1, 6]$, with the best results around $R \in [2, 4]$) and then degrades sharply at large ranks ($R \geq 8$). This drop should not be read as evidence that a high Volterra rank overfits. Because the parameter budget is held fixed, raising $R$ forces the grid spacing $\Delta_{\mathcal{F}}$ to coarsen in step, and the collapse appears precisely once $\Delta_{\mathcal{F}}$ leaves the resolution range that the frequency-grid ablation (Appendix D.2) independently found acceptable: the strong configurations all keep $\Delta_{\mathcal{F}} \leq 0.144$ ($R \leq 6$), whereas the degraded ones coarsen it to $0.188$ ($R = 8$) and $0.233$ ($R = 10$). The effect of rank is therefore confounded with the loss of spectral resolution. Unlike the frequency-grid study, we do not run the complementary experiment that increases $R$ with the grid held fixed—which would instead let the parameter count grow—so we cannot isolate the effect of rank itself or rule out that the degradation is driven entirely by the coarser $\Delta_{\mathcal{F}}$. What the experiment does establish is the practically relevant point: under a fixed parameter budget, a small number of Volterra interaction terms ($R \in [2, 4]$) already suffices to capture the dominant non-linear structure present in the data considered here, so reallocating the budget from spectral resolution to additional rank is not beneficial in this setting.

*Table 10.* Predictive log-likelihoods and CRPS on synthetic regression tasks for varying Volterra rank $R$.

| $(R, \Delta_{\mathcal{F}})$ | Matérn5/2 | | Periodic | | Sawtooth | |
|---|---|---|---|---|---|---|
| | Log-lik. ↑ | CRPS ↓ | Log-lik. ↑ | CRPS ↓ | Log-lik. ↑ | CRPS ↓ |
| (1, 0.033) | **0.07** ±0.00 | **0.16** ±0.00 | −0.08 ±0.00 | **0.18** ±0.00 | 1.15 ±0.00 | **0.06** ±0.00 |
| (2, 0.055) | **0.07** ±0.00 | **0.16** ±0.00 | **−0.07** ±0.00 | **0.18** ±0.00 | 1.14 ±0.02 | 0.07 ±0.00 |
| (4, 0.100) | **0.07** ±0.00 | **0.16** ±0.00 | **−0.05** ±0.00 | **0.17** ±0.00 | 1.18 ±0.00 | **0.06** ±0.00 |
| (6, 0.144) | **0.07** ±0.00 | **0.16** ±0.00 | −0.09 ±0.00 | **0.18** ±0.00 | 1.15 ±0.01 | **0.06** ±0.00 |
| (8, 0.188) | −0.32 ±0.00 | 0.23 ±0.00 | −0.39 ±0.00 | 0.24 ±0.00 | 0.56 ±0.03 | 0.12 ±0.00 |
| (10, 0.233) | −0.63 ±0.00 | 0.31 ±0.00 | −0.66 ±0.00 | 0.31 ±0.00 | 0.08 ±0.00 | 0.18 ±0.00 |

## D.4. Effective Receptive Field Analysis

A global theoretical receptive field need not imply a global effective one (ERF): Luo et al. (2016) showed that for deep CNNs the ERF is Gaussian-distributed and occupies only a small central fraction of the theoretical field. We therefore measure the ERF empirically across all nine one-dimensional models of Section 4.1.1 to test whether the proposed set-Fourier models retain global support in practice while discrete-convolution baselines stay local. Adapting the ERF to neural processes, the context observations $\{(x_c, y_c)\}$ play the role of the input and the predictive mean $m(x_q)$ (of $\eta[\mathcal{D}_c; x_q]$) that of the output:

$$\text{ERF}(x_q, x_c) = \mathbb{E}_{y_c}\left[\left(\frac{\partial m(x_q)}{\partial y_c}\right)^2\right], \tag{17}$$

with the expectation over context values $y_c$ at fixed locations $x_c$. A local field makes this decay sharply with $|x_q - x_c|$; a global one keeps it appreciable at large offsets. The estimator is the gradient second moment $\mathbb{E}_{y_c}[(\partial m(x_q)/\partial y_c)^2]$ in both settings; it upper-bounds the gradient variance analysed by Luo et al. (2016) and coincides with it whenever the mean gradient is negligible, whereas after training the generally nonzero mean gradient also contributes.

**Setup.** To separate the architectural prior from learned behaviour, we evaluate every model under a $2 \times 2$ design—randomly initialized vs. trained weights, crossed with random ($y_c \sim \mathcal{N}(0, 1)$) vs. data-driven context values sampled from the task—on three benchmarks (GP-Matérn-5/2, GP-Periodic, Sawtooth), reusing the trained checkpoints of Appendix E.1.1. For each setting, we form the Monte-Carlo estimate $\widehat{\text{ERF}}(x_q, x_{c,i})$ of Eq. (17) by averaging squared gradients over $J=1024$ context draws on a dense grid of $n_c=512$ locations $\{x_{c,i}\}$ in $[-3, 3]$, evaluated at five query points $x_q \in \{-2, -1, 0, 1, 2\}$. Because the absolute gradient scale varies by orders of magnitude across models, we summarise each profile with a scale-invariant locality measure, the near-field concentration

$$\zeta(r_0 \mid x_q) = \frac{\sum_{i:|x_{c,i}-x_q|\leq r_0} \widehat{\text{ERF}}(x_q, x_{c,i})}{\sum_i \widehat{\text{ERF}}(x_q, x_{c,i})}, \tag{18}$$

the fraction of sensitivity energy within radius $r_0=1$ of the query (averaged over queries): values near 1 indicate a local effective field, small values a global one.

ERF locality across all models × 4 scenarios   ("coll." = identically-zero trained gradient)

*Figure 1.* Effective-receptive-field locality across all models, the four scenarios, and three benchmarks. Each cell is the near-field concentration $\zeta(r_0 \mid x_q)$ (Eq. (18), $r_0{=}1$), averaged over queries: **blue** = local (energy at the query), **red** = global (spread across the domain). Columns vary the weights (random init vs. trained) and context values ($\mathcal{N}(0, 1)$ vs. data-driven). Hatched "coll." cells have an identically-zero trained gradient on Sawtooth (context-independent mean); for the non-equivariant baselines this is the data-mean collapse of Appendix D.5.

Figure 1 summarises the results. The two random-init columns are nearly identical within each benchmark, confirming that at initialisation the ERF is set by the architecture, independent of the input distribution (Luo et al., 2016). There the models split cleanly: ConvCNP and SConvCNP are strictly local ($\zeta{\approx}100\%$), whereas all others—including the proposed SFConvCNP and SFVConvCNP ($\zeta{\approx}25$–$29\%$)—spread sensitivity across the domain, so the set-Fourier parameterisation is globally supported by construction. After training, ConvCNP stays rigidly local on every task and SConvCNP predominantly so, while SFConvCNP and SFVConvCNP retain global reach where long-range structure exists (GP-Periodic, $59/44\%$; Sawtooth, $59/72\%$) but contract on the short-correlation GP-Matérn-5/2 ($70/96\%$)—i.e. their effective field adapts to the task while remaining globally capable. The mean-pooling and attention baselines are globally spread but unstructured (e.g. CNP attains the near-uniform $\zeta{\approx}33\%$ on GP-Periodic); CNP's apparent locality on GP-Matérn-5/2 ($99\%$) merely reflects a near-vanishing gradient that barely depends on the context *values* $y_c$, so $\zeta$ is computed on a negligible, noise-level signal rather than signalling a genuinely local field. On Sawtooth the five non-convolutional models collapse to the data-mean predictor and carry no gradient signal (hatched cells; Figure 3). Overall, the proposed models uniquely combine global support with task-adaptive reach, while convolutional baselines remain local and the rest provide only unstructured global spread. Note that Table 1 reports architectural capacity for global interaction, whereas the ERF measures the effective receptive field that is actually realised; the two need not coincide. SConvCNP is the clearest example: its spectral mixing is globally connected in principle (and is marked accordingly in Table 1), yet its trained ERF stays local, underscoring that global capacity does not guarantee global effective support.

### D.5. Translation Equivariance Under Domain Shift

If the data-generating process is stationary, a translation-equivariant model should generalise to input regions unseen during training. We test this by training all models on the synthetic benchmarks (Section 4.1.1) with context and query inputs from $\mathcal{U}[-3, 3]$, then evaluating on domains shifted by $\tau \in \{0, 3, 6, 9, 12, 15\}$, i.e. inputs from $\mathcal{U}[-3 + \tau, 3 + \tau]$; $\tau{=}0$ is the training domain, all shifts $\tau \geq 6$ yield domains disjoint from it.

Figure 2 confirms the expected split. The non-equivariant CNP, AttnCNP, and TNP degrade sharply as the domain shifts, revealing their reliance on absolute positions; the lone exception—CNP on Periodic—is flat only because it has collapsed to the (translation-invariant) data-mean predictor, not because it is equivariant. The equivariant ConvCNP, SConvCNP, TE-TNP, SFConvCNP, and SFVConvCNP remain stable across all shifts, depending only on relative displacements; TE-PT-TNP, whose equivariance is heuristic, also remains stable under these rigid shifts.

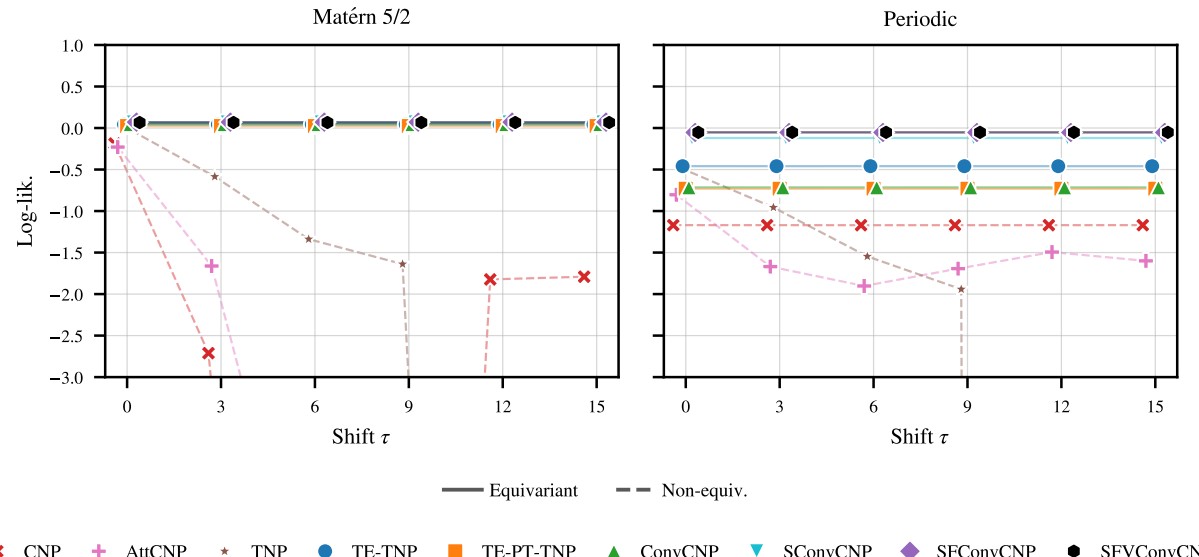

*Figure 2.* Predictive log-likelihood ($\uparrow$) vs. input-domain shift $\tau$ on the Matérn-5/2 and Periodic tasks (inputs from $\mathcal{U}[-3+\tau, 3+\tau]$; $\tau=0$ is the training domain). Solid lines: equivariant models; dashed: non-equivariant. Markers are dodged at each shift to expose the overlapping equivariant cluster. Sawtooth is omitted: there the non-equivariant models collapse to the data-mean predictor and are trivially flat under shift.

## E. Experimental Details

All experiments were implemented in PyTorch (Paszke et al., 2019). All models were trained on a single NVIDIA A100 GPU with 40 GB of memory. For testing, however, we used a mix of NVIDIA A100 and T4 (16 GB) GPUs, depending on the model size and cluster availability. Our implementation builds upon and extends the codebases of Ashman et al. (2024a) and Bruinsma et al. (2023). For a fair comparison, we tried to keep the trainable parameter count as close as possible across all models; as a result, some configurations may be less conventional than is common.

**Functional-Embedding Convention.** Section 2.2 presents the functional embedding (2) with $\varphi(y) = (1, y)$ and $\mathcal{Z} = \mathbb{R}^{1+d_{\mathcal{Y}}}$, i.e., a single density channel shared across all output dimensions. Following the codebases above, our implementations instead embed each output dimension separately: each pair $(1, y^{(j)})$, $j \in \{1, \ldots, d_{\mathcal{Y}}\}$, is smoothed by its own kernel $\psi_e^{(j)}$ with independently learnable length scales, yielding $\mathcal{Z} = \mathbb{R}^{2d_{\mathcal{Y}}}$ with one density channel per output dimension. Since the kernels are learned per channel, the density channels do not coincide in general.

**Metrics and Aggregation.** Unless stated otherwise, all reported metrics are computed by aggregating pointwise scores in three stages, consistent across every benchmark. (i) Pointwise. For each query point $(x_q, y_q)$, we evaluate a pointwise score: either the predictive log-likelihood or, since all models produce Gaussian predictives, the closed-form Gaussian CRPS

$$\text{CRPS}(\mathcal{N}(m, s^2), y) = s \left[ z\left(2\Phi(z) - 1\right) + 2\Phi'(z) - \pi^{-1/2} \right],$$

where $z = (y - m)/s$, $\Phi$ is the standard normal CDF, and $\Phi'$ its density. (ii) Per task. Pointwise scores are reduced to a single scalar per task by taking the arithmetic mean over that task's query points. (iii) Per run. The per-task scalars are then averaged across all tasks in the (validation or test) split, so that each task contributes equal weight independent of its query-set size. Finally, table entries are reported as mean $\pm$ sample standard deviation across the 5 independent training seeds, computed on the common, fixed test split.

## E.1. One-Dimensional Regression

### E.1.1. SYNTHETIC REGRESSION

**Model Architectures.** This section describes the architectures of all models used in the one-dimensional regression experiments (Section 4.1.1). Parameter counts for each model are reported in Table 11. All models predict a factorized Gaussian distribution over query points, parameterized by a mean and a scale. The scale is produced in pre-softplus form and transformed via a softplus non-linearity (Dugas et al., 2000), with an added minimum noise floor of $10^{-6}$ to ensure numerical stability. All multilayer perceptrons (MLPs) use ReLU activations (Nair & Hinton, 2010). Decoder output layers have $2d_{\mathcal{Y}}$ units, corresponding to the predictive mean and pre-softplus scale.

*Table 11.* Trainable parameter counts for models used in the one-dimensional synthetic regression experiments.

| Model | Parameters |
| --- | --- |
| CNP | 31.16M |
| AttnCNP | 30.35M |
| TNP | 38.86M |
| TE-TNP | 35.83M |
| TE-PT-TNP | 36.09M |
| ConvCNP | 32.75M |
| SConvCNP | 31.50M |
| SFConvCNP | 34.21M |
| SFVConvCNP | 38.25M |

**Conditional Neural Process (CNP).** The CNP processes each context pair $(x_{c,k}, y_{c,k}) \in \mathcal{D}_c$ using separate input and output pathways. Inputs $x_{c,k}$ and outputs $y_{c,k}$ are passed through distinct MLPs, each with two hidden layers of width 1280, yielding representations $h_{c,k}^{(x)}$ and $h_{c,k}^{(y)}$. These are concatenated and processed by an MLP with six hidden layers of width 1280, producing a 1280-dimensional embedding $h_{c,k}$ for each context point. Context embeddings are averaged to obtain a single representation $h_c$. For prediction, each query input $x_q$ is encoded by the shared input MLP, concatenated with $h_c$, and passed through a decoder MLP with six hidden layers of width 1280.

**Attentive Conditional Neural Process (AttnCNP).** The AttnCNP processes context pairs using an attention-based aggregation mechanism. Each context pair $(x_{c,k}, y_{c,k})$ is encoded by two separate MLPs with two hidden layers of width 640, concatenated, and passed through an additional MLP with two hidden layers of width 640 to produce context embeddings. These embeddings are refined using four layers of multi-head self-attention with 10 heads (head dimension 64, feedforward width 2560), incorporating residual connections and pre-layer normalization. Query representations are obtained via a single layer of multi-head cross-attention from queries to the attended context embeddings, using the same attention configuration. The resulting query embeddings are mapped to predictive distribution parameters by a decoder MLP with six hidden layers of width 640.

**Transformer Neural Process (TNP).** The TNP processes context and query points jointly using transformer layers. Token representations are constructed for both context and query points. For each context pair $(x_{c,k}, y_{c,k})$, the token is $[x_{c,k}, y_{c,k}, 0]$, where the final scalar is a missingness flag set to 0 at observed points. For each query location $x_q$, the token is $[x_q, \mathbf{0}, 1]$, with a zero-valued placeholder for the unobserved output and the flag set to 1 to indicate its absence (Nguyen & Grover, 2022; Ashman et al., 2024a). Tokens are embedded via a shared MLP with two hidden layers of width 512.

The model adopts the Efficient Query TNP architecture (Feng et al., 2022), which separates context and query processing to reduce computational complexity from $\mathcal{O}((|\mathcal{D}_c| + |\mathcal{D}_q|)^2)$ to $\mathcal{O}(|\mathcal{D}_c|^2 + |\mathcal{D}_q||\mathcal{D}_c|)$. Context tokens are first updated using self-attention, after which each query attends to the updated contexts via cross-attention. The model uses six transformer layers with 8 heads (head dimension 64, feedforward width 2048) and pre-layer normalization. Context self-attention and query–context cross-attention use distinct parameter sets at each layer. Final query embeddings are passed through a decoder MLP with two hidden layers of width 512.

**Translation-Equivariant Transformer Neural Process (TE-TNP).** The TE-TNP enforces translation equivariance by excluding absolute input locations from token representations. Context tokens take the form $[y_{c,k}, 0]$, while query tokens are

$[\mathbf{0}, 1]$; the trailing scalar is a missingness flag, set to $1$ at unobserved (query) locations and $0$ at observed (context) locations. Tokens are embedded using a shared MLP with two hidden layers of width 512.

Standard multi-head attention is replaced with a translation-equivariant attention mechanism. For each head, scaled dot products between token embeddings are computed independently of absolute locations, while pairwise differences between token locations are computed separately. These quantities are concatenated and passed through an MLP with two hidden layers of width 512, producing attention logits for each head. The model uses six transformer layers with 8 heads (head dimension 64, feedforward width 1536) and pre-layer normalization. Following the Efficient Query TNP design, context self-attention and query–context cross-attention are performed sequentially at each layer using distinct parameter sets. Final query embeddings are passed through a decoder MLP with two hidden layers of width 512.

**Translation-Equivariant Pseudo-Token Transformer Neural Process (TE-PT-TNP).**  The TE-PT-TNP introduces a set of $M = 32$ learnable pseudo-tokens that act as an information bottleneck, eliminating direct context–context and context–query attention to reduce complexity from $\mathcal{O}(|\mathcal{D}_c|^2 + |\mathcal{D}_q||\mathcal{D}_c|)$ to $\mathcal{O}(M|\mathcal{D}_c| + M|\mathcal{D}_q|)$. Each pseudo-token consists of a learnable 320-dimensional feature embedding and a learnable location. The model adopts the Induced Set Attention variant (Lee et al., 2019; Jaegle et al., 2021b; Ashman et al., 2024a). Token representations follow the TE-TNP design: context tokens are $[y_{c,k}, 0]$ and query tokens are $[\mathbf{0}, 1]$, embedded using a shared MLP with two hidden layers of width 320.

In contrast to TE-TNP, where each layer applies context self-attention followed by context–query cross-attention, each intermediate layer of TE-PT-TNP performs three attention steps: (i) context-to-pseudo attention, where pseudo-tokens attend to context tokens; (ii) pseudo-to-context attention, where context tokens attend to the updated pseudo-tokens; and (iii) pseudo-to-query attention, where query tokens attend to the updated pseudo-tokens. In the final layer, only context-to-pseudo and pseudo-to-query attention are performed, as context embeddings are no longer required thereafter. The three attention steps within each layer use distinct parameter sets. As in TE-TNP, standard multi-head attention is replaced with the translation-equivariant attention mechanism: scaled dot products between token embeddings are computed independently of absolute locations, while pairwise differences between token locations are computed separately, and these quantities are concatenated and passed through an MLP with two hidden layers of width 320 to produce attention logits for each head. The model uses six transformer layers with 8 heads (head dimension 64, feedforward width 2048) and pre-layer normalization.

To preserve translation equivariance, the pseudo-token locations are not used in their raw learnable form but are first shifted to track the context set, while the pseudo-token feature embeddings are left unchanged. This shift is produced by a learned attention-based initializer: each pseudo-token forms a query and the context tokens form keys (via separate linear projections), and multi-head scaled dot-product attention is computed using the same number of heads and head dimension as the main attention layers (8 heads, head dimension 64). Unlike those layers, the attention logits here depend only on the token embeddings and not on the input locations, so the weights are unchanged by a shift of the data. Within each head, these weights are applied to the context input locations, yielding a per-head attention-weighted average of context locations for each pseudo-token. The per-head estimates are then combined through a softmax-normalized set of learnable per-head weights, and the result is added as an offset to the corresponding learnable pseudo-token location. Because both the per-head averages and the per-head combination are convex, the offset is a convex combination of context locations, and hence tracks the context set under translation. We emphasize that this is a heuristic proxy for the translation rather than a guarantee: the offset summarizes the context locations by a weighted average, which Ashman et al. (2024a, §3.1) show can degenerate—for data split into two distant clusters, for instance, the pseudo-tokens are placed near the midpoint, far from every observation. Final query embeddings are passed through a decoder MLP with two hidden layers of width 320.

**Convolutional Conditional Neural Process (ConvCNP).**  The ConvCNP processes context observations on a discretized grid using convolutional operations. The model determines per-dimension minima and maxima over both context and query inputs, expands these bounds by 0.1, and discretizes the resulting interval at a resolution of 64 points per unit. The discretization range is further expanded as needed to satisfy CNN grid-size constraints while preserving resolution. The resulting grid $\mathcal{G}$ is formed via a Cartesian product across dimensions.

The functional embedding in (2) is evaluated on $\mathcal{G}$ using Gaussian kernels initialized with separate length scales of $2/64$ per input dimension and embedding channel. The density channel corresponding to

$$\text{Density}(x) = \sum_{(x_c, y_c) \in \mathcal{D}_c} \psi_e(x - x_c),$$

is used to normalize the functional embedding, yielding the following normalized representation

$$
\left( \text{Density}(x_g),\ \frac{\sum_{(x_c, y_c) \in \mathcal{D}_c} \varphi(y_c)\, \psi_e(x_g - x_c)}{\text{Density}(x_g)} \right)_{x_g \in \mathcal{G}}.
\tag{19}
$$

At each grid point the density channel and the density-normalized feature channels are concatenated into a single vector (the ordered pair in (19)) and processed independently by an MLP with two hidden layers of width 256; since this pointwise MLP mixes all channels jointly, the order in which the density and feature channels are concatenated is immaterial here.

The resulting grid is passed to a U-Net-style CNN, following the implementation of Bruinsma et al. (2023). The encoder applies six residual convolutional blocks (kernel size 11; output channels 288 for the first three blocks and 512 for the last three), each reducing the grid resolution by a factor of 2. Each downsampling step is realized either as a stride-2 convolution or, when the running receptive field is odd, as a stride-1 convolution followed by $2\times$ average pooling. A symmetric decoder upsamples through six transposed-convolution blocks, concatenating the corresponding encoder feature map at each level via a skip connection, with a final $1\times1$ convolution producing the output channels. Since the six encoder blocks downsample the grid by a total factor of $2^6 = 64$, the grid size is ensured to be divisible by 64 via symmetric expansion of the discretization interval.

To produce predictions at off-grid query locations, features are interpolated from the CNN output using a Gaussian kernel with separate learnable length scales per input dimension and embedding channel, without density normalization. As in the encoder, these length scales are initialized to $2/64$. These query-specific embeddings are passed through a decoder MLP with two hidden layers of width 256 to produce predictive distribution parameters.

**Spectral Convolutional Conditional Neural Process (SConvCNP).**   The SConvCNP replaces the U-Net backbone of the ConvCNP with a Fourier Neural Operator (Li et al., 2020). The grid-based encoder is identical to that of ConvCNP (Appendix E.1.1); as there, the density and feature channels are concatenated, and the order is immaterial because the pointwise MLPs and the FNO's channel-wise spectral mixing combine all channels jointly. The FNO backbone consists of six residual layers with 384 channels and retains 32 Fourier modes. A pointwise MLP first projects the gridded encoder representation to the working width of 384 channels, after which the six residual Fourier blocks are applied, and a final pointwise MLP projects back to the embedding dimension. Each residual Fourier block computes a spectral convolution on its main branch: the input is mapped to the frequency domain via an FFT, the lowest 32 modes per dimension are retained (higher frequencies are discarded), these coefficients are mixed by a learnable complex-valued linear transform, and an inverse FFT returns the result to the spatial domain. This spectral path is interleaved with pointwise (channel-wise) MLPs and GELU non-linearities (Hendrycks & Gimpel, 2016), and its output is added back to the block input through a residual connection.

**Set Fourier Convolutional Conditional Neural Process (SFConvCNP).**   SFConvCNP replaces the attention mechanism in TNPs with Set Fourier Convolution (SFConv; Section 3.2), yielding a translation-equivariant architecture. We follow the tokenization scheme of TE-TNP: each context input is represented as $[y_{c,k}, 0]$, while query inputs are represented as $[\mathbf{0}, 1]$. Both token types are embedded using a shared MLP with two hidden layers of width 288. All remaining components of a transformer layer—residual connections, layer normalization, and position-wise feedforward networks—are kept unchanged. We refer to a transformer layer in which attention is replaced by SFConv as an SFConvBlock. Following the Efficient Query design, each layer first updates the context representations with a context–context SFConvBlock and then maps to the query locations with a query–context SFConvBlock; the two streams use separate parameter sets at every layer. Within each SFConvBlock, the position-wise feedforward network uses GELU activations in place of ReLU; the shared token-embedding and decoder MLPs retain ReLU.

In the continuous setting, SFConv computes convolutions via the Fourier domain. Given a convolution kernel $\kappa$ and a functional embedding $h = \rho_e[\mathcal{D}_c]$, the convolution operator $\mathcal{K}$ can be written as

$$
\begin{aligned}
g(x) := \mathcal{K}[h](x) &= \mathcal{F}_{d_\mathcal{X}}^{-1}\!\left[ \widehat{\kappa}(\xi)\, \widehat{h}(\xi) \right](x) \\
&= \int_{\mathbb{R}^{d_\mathcal{X}}} \widehat{\kappa}(\xi)\, \widehat{h}(\xi)\, e^{i2\pi \langle x, \xi \rangle}\, \mathrm{d}\xi,
\end{aligned}
\tag{20}
$$

where $\widehat{\kappa} = \mathcal{F}_{d_\mathcal{X}}[\kappa]$ and $\widehat{h} = \mathcal{F}_{d_\mathcal{X}}[h]$ denote the Fourier transforms of the kernel and functional embedding, respectively.

In practice, the integral in (20) is approximated over a finite, symmetric frequency grid $\Xi \subset \mathbb{R}^{d_{\mathcal{X}}}$. For each input dimension $j \in \{1, \ldots, d_{\mathcal{X}}\}$, we specify a frequency resolution $\Delta_{\mathcal{F}}^{(j)}$ and a maximum frequency $\xi_{\max}^{(j)}$, and define the one-dimensional grid

$$\Xi^{(j)} = \left\{ -n_{\mathcal{F}}^{(j)} \Delta_{\mathcal{F}}^{(j)}, \ldots, 0, \ldots, n_{\mathcal{F}}^{(j)} \Delta_{\mathcal{F}}^{(j)} \right\},$$

where $n_{\mathcal{F}}^{(j)} = \lfloor \xi_{\max}^{(j)} / \Delta_{\mathcal{F}}^{(j)} \rfloor$ is the number of positive frequencies along dimension $j$; equivalently, the grid retains all integer multiples of $\Delta_{\mathcal{F}}^{(j)}$ whose magnitude is at most $\xi_{\max}^{(j)}$, so the largest retained frequency is $n_{\mathcal{F}}^{(j)} \Delta_{\mathcal{F}}^{(j)} = \xi_{\max}^{(j)}$. The full $d_{\mathcal{X}}$-dimensional grid is then given by the Cartesian product

$$\Xi = \Xi^{(1)} \times \cdots \times \Xi^{(d_{\mathcal{X}})},$$

containing $\prod_{j=1}^{d_{\mathcal{X}}} (2n_{\mathcal{F}}^{(j)} + 1)$ frequency components.

When $\rho_e$ is constructed using a Gaussian kernel, the Fourier transform $\widehat{h}$ can be computed in closed form on $\Xi$ using Equations (9) and (10). The Gaussian length scales are learnable, with separate parameters for each input dimension and embedding channel, initialized to 0.05.

Approximating the inverse Fourier transform in (20) with a Riemann sum over $\Xi$ yields

$$g(x) \approx \sum_{\xi \in \Xi} \widehat{\kappa}(\xi)\, \widehat{h}(\xi)\, e^{i2\pi\langle x, \xi \rangle}\, \mathrm{vol}(B(\xi)),$$

where $B(\xi)$ denotes the axis-aligned hypercube centered at $\xi$ with side lengths $\Delta_{\mathcal{F}}^{(j)}$. Since the grid is uniform, all bins have equal volume $\Delta_{\Xi} := \prod_{j=1}^{d_{\mathcal{X}}} \Delta_{\mathcal{F}}^{(j)}$, and the approximation simplifies to

$$g(x) \approx \Delta_{\Xi} \sum_{\xi \in \Xi} \widehat{\kappa}(\xi)\, \widehat{h}(\xi)\, e^{i2\pi\langle x, \xi \rangle}. \tag{21}$$

We require the output $g(x)$ to be real-valued, which holds precisely when its Fourier transform satisfies the Hermitian symmetry condition

$$\widehat{g}(-\xi) = \overline{\widehat{g}(\xi)}, \qquad \xi \in \mathbb{R}^{d_{\mathcal{X}}},$$

where $\overline{(\cdot)}$ denotes complex conjugation. Since $h$ is real-valued, $\widehat{h}$ is automatically Hermitian, so this amounts to a condition on the kernel spectrum $\widehat{\kappa}$. Consequently, the spectrum is fully determined by any subset of frequencies containing exactly one representative from each pair $\{\xi, -\xi\}$. We exploit this property by retaining only the closed half-grid

$$\Xi^{+} = \{\xi \in \Xi : \xi^{(d_{\mathcal{X}})} \geq 0\}.$$

For a frequency with $\xi^{(d_{\mathcal{X}})} > 0$, the partner $-\xi$ lies outside $\Xi^{+}$, and its contribution is recovered from Hermitian symmetry by doubling the real part. Frequencies on the boundary hyperplane $\{\xi^{(d_{\mathcal{X}})} = 0\}$, by contrast, appear in $\Xi^{+}$ together with their partners ($\xi = \mathbf{0}$ being self-paired), so their contributions must be counted only once. Introducing the quadrature weights $w(\xi) = \frac{1}{2}$ if $\xi^{(d_{\mathcal{X}})} = 0$ and $w(\xi) = 1$ otherwise, the sum in (21) can be rewritten exactly as

$$g(x) = 2\Delta_{\Xi} \sum_{\xi \in \Xi^{+}} w(\xi)\, \Re\left[ \widehat{\kappa}(\xi)\, \widehat{h}(\xi)\, e^{i2\pi\langle x, \xi \rangle} \right].$$

Expanding the real part yields

$$\Re\left[ \widehat{\kappa}(\xi)\, \widehat{h}(\xi)\, e^{i2\pi\langle x, \xi \rangle} \right] = \Re[\widehat{\kappa}(\xi)\widehat{h}(\xi)] \cos(2\pi\langle x, \xi \rangle)$$
$$- \Im[\widehat{\kappa}(\xi)\widehat{h}(\xi)] \sin(2\pi\langle x, \xi \rangle).$$

Thus, it suffices to parameterize $\widehat{\kappa}$ only on the half-grid $\Xi^{+}$, reducing the number of required frequency parameters by a factor of approximately two, analogous to the real-input FFT conventions employed by Fourier neural operators (Li et al., 2020; Gupta & Brandstetter, 2022). For each retained frequency $\xi$, we parameterize the kernel spectrum as

$$\widehat{\kappa}(\xi) = W_{\xi},$$

where $W_\xi \in \mathbb{C}^{c_{\text{out}} \times c_{\text{in}}}$ is a learnable complex-valued weight matrix acting on the $c_{\text{in}}$ channels of the Fourier-transformed functional embedding. One subtlety arises for $d_\mathcal{X} \geq 2$: pairs $\{\xi, -\xi\}$ on the boundary hyperplane $\{\xi^{(d_\mathcal{X})} = 0\}$ both lie in $\Xi^+$ and receive independent weights, so the Hermitian condition is not imposed there explicitly. Because the output depends on the weights only through the real part above, each such pair contributes solely through the Hermitian combination $\frac{1}{2}(W_\xi + \overline{W_{-\xi}})$, and $W_0$ only through its real part; the construction therefore remains equivalent to a convolution with a Hermitian kernel spectrum, at the cost of a mild parameter redundancy confined to this hyperplane (roughly 2–3% of the spectral weights in our 2D and 3D configurations, and only $\Im W_0$ in 1D), mirroring the duplicated conjugate modes retained by real-input FFT conventions. To further reduce the parameter count, we optionally employ grouped convolutions: input and output channels are partitioned into groups of equal size, and each group is parameterized independently, analogous to grouped convolutions in CNNs (Krizhevsky et al., 2012; Xie et al., 2017; Howard et al., 2017).

For the one-dimensional synthetic regression experiments, we set $\xi_{\max}^{(1)} = 4.9$ and $\Delta_\mathcal{F}^{(1)} = 0.1$, yielding $n_\mathcal{F}^{(1)} = 49$ positive frequencies and, including zero, 50 non-negative frequencies $\{0, 0.1, \ldots, 4.9\}$ (largest retained frequency $n_\mathcal{F}^{(1)} \Delta_\mathcal{F}^{(1)} = 4.9$) for a total of $|\Xi| = 99$ frequency points after symmetrization. Exploiting Hermitian symmetry, only these 50 frequencies with $\xi^{(1)} \geq 0$ are parameterized. Each $W_\xi$ acts on the Fourier-transformed functional embedding. The embedding stacks the 288 density channels and the 288 feature channels as two **contiguous** blocks—all 288 density channels first, then all 288 feature channels, rather than interleaving each feature channel with its density channel—for $2 \times 288 = 576$ effective input channels, mapped to 288 output channels. For the grouped convolution these 576 channels are split into 4 contiguous groups of 144 (each group's weight maps $144 \to 72$); under this blocked ordering the split falls on the block boundary, so two groups process only density channels and two only feature channels. The concatenation order thus determines which channels are mixed within each group and, as we observe empirically, materially affects predictive performance. The convolution output is then projected linearly to 288 dimensions; we refer to this cross-group linear projection as *output feature mixing* (when it is disabled, the grouped-convolution outputs are used directly).

The full model consists of six SFConvBlocks. Each position-wise feedforward network has hidden width 1152. Following the Efficient Query TNP design, context embeddings are first updated via SFConv applied to the context set, after which query embeddings are updated via a second SFConv using the updated context embeddings. This context–query separation is repeated at every layer. Final query embeddings are passed through a decoder MLP with two hidden layers of width 288.

**Set Fourier Volterra Convolutional Conditional Neural Process (SFVConvCNP).** The SFVConvCNP explicitly approximates a truncated Volterra expansion of the translation-equivariant operator (Section 3.1). Token representations follow the TE-TNP design: context tokens are $[y_{c,k}, 0]$ and query tokens are $[\mathbf{0}, 1]$, embedded using a shared MLP with two hidden layers of width 128. Whereas SFConvCNP applies a single SFConv per block, SFVConvCNP augments each block with additional SFConvs whose outputs are combined multiplicatively to approximate higher-order (quadratic) Volterra terms. Unlike SFConvCNP, whose position-wise feedforward networks use GELU activations, SFVConvBlocks use an identity activation, since the required non-linearity is supplied by the multiplicative combination of SFConv outputs. All other transformer-layer components—residual connections and layer normalization—are retained as in SFConvCNP; layer normalization is a pointwise, translation-equivariant operation that lies outside the Volterra formalism, so the SFVConvBlock backbone realizes the truncated Volterra cascade up to these normalization layers.

Each SFVConvBlock effectively computes $2R + 1$ SFConvs in parallel, where $R$ is the Volterra rank truncation parameter (we use $R = 4$). To implement this, the input feature tensor is broadcast $2R + 1$ times along the channel dimension. These replicas correspond to the $2R + 1$ Volterra branches and are processed jointly using a single SFConv module with an increased number of effective groups. Specifically, within each of the $2R + 1$ Volterra branches, the 128 density channels and 128 feature channels are stacked **blockwise** (as in SFConvCNP) into 256 channels and partitioned into 4 *contiguous* groups of 64, and convolution is performed independently within each group. As a result, the SFConv module uses grouped convolution both across Volterra branches and within each branch, substantially reducing the number of learnable parameters while preserving parallel evaluation. All convolutions share the same input locations and frequency grid (with $\xi_{\max} = 4.9$ and $\Delta_\mathcal{F} = 0.1$), but use independent Fourier-domain kernel weights across groups.

The grouped SFConv produces $2R + 1$ output feature blocks

$$\{z_0, z_1^{(1)}, z_1^{(2)}, \ldots, z_R^{(1)}, z_R^{(2)}\}.$$

The first block $z_0$ represents the standard linear convolution contribution. For each rank $r \in \{1, \ldots, R\}$, the corresponding

pair $(z_r^{(1)}, z_r^{(2)})$ is combined via element-wise multiplication,

$$q_r = z_r^{(1)} \odot z_r^{(2)},$$

yielding a quadratic interaction feature. These $R$ quadratic features are aggregated using a learnable affine combination,

$$q = \sum_{r=1}^{R} \alpha_r \, q_r + \beta,$$

with scalar trainable coefficients $\{\alpha_r\}_{r=1}^{R}$ and a scalar bias $\beta$ (shared across channels), implemented as a single linear layer over the $R$ rank dimension. The output of the SFVConvBlock is then given by

$$z_{\text{out}} = z_0 + q,$$

reflecting the additive combination of first- and second-order terms in the Volterra expansion (Equation (5)).

The model uses five SFVConvBlocks; since each block contributes the first- and second-order terms of one cascade level, the number of stacked blocks sets the truncation depth of the Volterra cascade (Section 3.1), here $L = 5$. The position-wise feedforward subnetwork within each SFVConvBlock uses a hidden width of 512. Following the Efficient Query TNP design, at each layer context embeddings are first updated using an SFVConvBlock over the context set, then query embeddings are updated via a second SFVConvBlock using the updated context embeddings. The context–context and context–query SFVConvBlocks use separate parameter sets at every layer. This context–query separation is repeated across layers. Final query embeddings are passed through a decoder MLP with two hidden layers of width 128.

**Data-Generating Process.** We evaluate all models on five families of synthetic 1D regression tasks, each defined by a distinct stochastic generative process: GPs with RBF, Matérn–5/2, and periodic kernels, sawtooth waves, and square waves. GP tasks are sampled using GPyTorch (Gardner et al., 2018). All processes include independent Gaussian observation noise with standard deviation $\sigma_0 > 0$, following the observation model of Section 2.1. For the sawtooth and square-wave families, process-specific parameters are sampled independently for each task. For the GP families, kernel hyperparameters are instead sampled once per batch and shared by all tasks in that batch (and resampled across batches), while the latent function realization is drawn independently for each task conditional on these hyperparameters; the parameter ranges below therefore describe the marginal distribution of each task's hyperparameters.

- **GP with RBF kernel.** Latent functions are sampled from $f \sim \mathcal{GP}(0, k_{\text{RBF}})$, where

  $$k_{\text{RBF}}(x, x') = \exp(-\frac{1}{2\lambda^2}(x - x')^2),$$

  and $\lambda$ is the lengthscale parameter, sampled log-uniformly on $[0.25, 1)$ (i.e., $\log_{10} \lambda \sim \mathcal{U}[-0.602, 0)$). Observations are generated as $y = f(x) + \epsilon$ with $\epsilon \sim \mathcal{N}(0, \sigma_0^2)$ and $\sigma_0 = 0.1$.

- **GP with Matérn–5/2 kernel.** Latent functions are sampled from $f \sim \mathcal{GP}(0, k_{\text{m5/2}})$, where

  $$k_{\text{m5/2}}(x, x') = \frac{2^{-1.5}}{\Gamma(2.5)} \, (\sqrt{5} \, d)^{2.5} K_{2.5}(\sqrt{5} \, d),$$

  and $K_{2.5}$ is the modified Bessel function of the second kind. The scaled distance is $d = |x - x'|/\lambda$ with lengthscale $\lambda$ sampled log-uniformly on $[0.25, 1)$ (i.e., $\log_{10} \lambda \sim \mathcal{U}[-0.602, 0)$). Observations are generated as $y = f(x) + \epsilon$ with $\epsilon \sim \mathcal{N}(0, \sigma_0^2)$ and $\sigma_0 = 0.1$.

- **GP with periodic kernel.** Latent functions are sampled from $f \sim \mathcal{GP}(0, k_{\text{p}})$, where

  $$k_{\text{p}}(x, x') = \exp\left(-\frac{2\sin^2(\pi|x - x'|/\rho)}{\lambda^2}\right),$$

  with period $\rho$ and lengthscale $\lambda$ both sampled log-uniformly, $\log_{10} \rho \sim \mathcal{U}[-0.301, 0.301)$ (i.e., $\rho \in [0.5, 2)$) and $\log_{10} \lambda \sim \mathcal{U}[-0.602, 0)$ (i.e., $\lambda \in [0.25, 1)$). Observations are generated as $y = f(x) + \epsilon$ with $\epsilon \sim \mathcal{N}(0, \sigma_0^2)$ and $\sigma_0 = 0.1$.

- **Sawtooth wave.** The latent function is deterministic with mean function

$$f(x) = m_{\text{saw}}(x) = 2\left((\omega u x - c) \bmod 1\right) - 1,$$

  with frequency $\omega \sim \mathcal{U}[0.5, 5]$, direction $u \in \{+1, -1\}$ (sampled uniformly), and phase offset $c \sim \mathcal{U}[0, 1)$. Observations are generated as $y = f(x) + \epsilon$ with $\epsilon \sim \mathcal{N}(0, \sigma_0^2)$ and $\sigma_0 = 0.05$.

- **Square wave.** The latent function is deterministic with mean function

$$f(x) = m_{\text{sq}}(x) = 2\,\mathbb{1}_{\{((\omega x - c) \bmod 1) < D\}} - 1,$$

  with frequency $\omega \sim \mathcal{U}[0.5, 5]$, duty cycle $D \sim \mathcal{U}[0.25, 0.75]$, and phase offset $c \sim \mathcal{U}[0, 1)$. Observations are generated as $y = f(x) + \epsilon$ with $\epsilon \sim \mathcal{N}(0, \sigma_0^2)$ and $\sigma_0 = 0.05$.

**Tasks and Splits.** Each task presents a set of context and query points drawn from a single sampled function, with input locations sampled uniformly and independently from $[-3, 3)$ for each task. During training, the numbers of context and query points are sampled independently per batch as $|\mathcal{D}_c|, |\mathcal{D}_q| \sim \mathcal{U}[5, 50]$ and *shared* across all tasks in the batch. Validation and test use fixed sets of 8,000 and 64,000 tasks (125 and 1,000 batches of 64, respectively), held fixed across all epochs and runs; for both, the number of query points is fixed at $|\mathcal{D}_q| = 128$ while $|\mathcal{D}_c| \sim \mathcal{U}[5, 50]$.

**Training Protocol.** All models are trained for 250 epochs using the AdamW optimizer (Loshchilov & Hutter, 2017a) with learning rate $5 \times 10^{-4}$ and gradient clipping at maximum norm 0.5 (Pascanu et al., 2013). The learning rate is annealed via a cosine schedule (Loshchilov & Hutter, 2017b) with minimum learning rate $10^{-6}$. Each epoch processes 16,000 training tasks in batches of 64. Data loading used a single worker process for both training and evaluation.

**Evaluation Protocol.** Validation is performed every epoch on the fixed validation set, and the best checkpoint is selected based on validation log-likelihood. Final results are reported on the fixed test set.

**Qualitative Predictions.** Figure 3 shows example predictions of all nine models on a single held-out task for each of four representative benchmarks (GP-Matérn-5/2, GP-Periodic, Sawtooth, and Square). Within each column (benchmark), every model is evaluated on the same context and query points so that panels are directly comparable.

**Baseline Failure Modes.** Two distinct failure modes appear among the weaker baselines. First, the plain CNP underperforms on most benchmarks (e.g., periodic and square wave) and is the weakest baseline in higher-dimensional settings as well. As discussed in Section 3.4, this reflects its mean-pooling aggregation, which embeds context points independently and allows no pairwise interaction between them (Xu et al., 2020). Second, and more strikingly, the attention-based models collapse to the trivial marginal predictor on the sawtooth task specifically, yet remain competitive on the smooth GP benchmarks (Mohseni & Duffield, 2025). We attribute this *selective* collapse to spectral bias—the tendency of neural networks to favor low-frequency structure (Rahaman et al., 2019; Ronen et al., 2019; Basri et al., 2020; Tancik et al., 2020; Fridovich-Keil et al., 2022), reportedly stronger in transformers than in convolutional networks (Vasudeva et al., 2025). The slowly decaying ($\mathcal{O}(1/\xi)$) sawtooth spectrum would make this failure especially pronounced. The square wave, however, shares similar discontinuities and high-frequency content yet does not collapse as severely, so the precise cause remains open. The CNP also reaches the degenerate predictor on sawtooth, but likely as part of its broader underfitting rather than this high-frequency-specific failure.

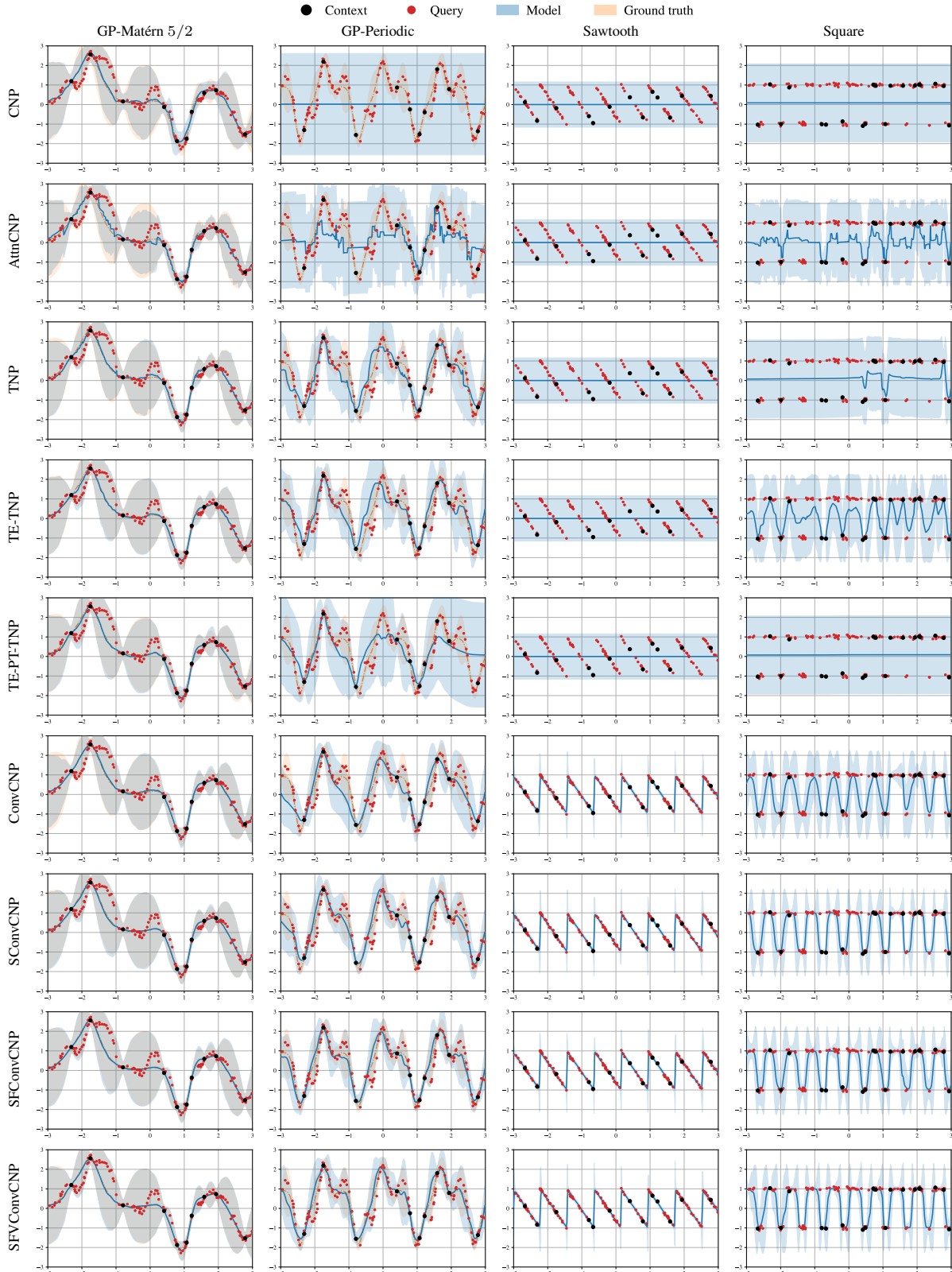

*Figure 3.* Example predictions on the synthetic 1D benchmarks. Columns: benchmarks; rows: models. All models in a column see the same task. Blue: predictive mean $\pm 2$ std. Black: context; red: query. Orange (GP rows): true posterior mean $\pm 2$ std.

### E.1.2. PREDATOR–PREY REGRESSION

**Model Architectures.** This section describes the architectures of all models used in the predator–prey experiments (Section 4.1.2). Parameter counts for each model are reported in Table 12. All models predict factorized Gaussian distributions over query points and over the two output dimensions (prey and predator populations), parameterized by a mean and a scale. The scale is produced in pre-softplus form and transformed via a softplus non-linearity, with an added minimum noise floor of $10^{-6}$. We note that, although population counts are inherently discrete, they are treated as continuous variables in our implementation, consistent with the Gaussian predictive distributions employed above.

*Table 12.* Trainable parameter counts for models used in the predator–prey experiments.

| Model | Parameters |
| --- | --- |
| CNP | 31.16M |
| AttnCNP | 30.36M |
| TNP | 38.87M |
| TE-TNP | 35.83M |
| TE-PT-TNP | 36.09M |
| ConvCNP | 32.75M |
| SConvCNP | 31.50M |
| SFConvCNP | 34.21M |
| SFVConvCNP | 38.25M |

The architectures used in the predator–prey experiments are identical to those of the one-dimensional synthetic regression experiments (Appendix E.1.1)—including all embedding dimensions, MLP and feedforward widths, attention head configurations, numbers of transformer layers and SFConv(V) blocks, pseudo-token counts, FNO channels and retained Fourier modes, and the SFConv/SFVConv frequency grids ($\xi_{\max} = 4.9$, $\Delta_{\mathcal{F}} = 0.1$), Volterra rank ($R = 4$), and grouping. Parameter counts (Table 12) are therefore matched across the two benchmarks. Only two architectural details differ. First, the output dimension is $d_{\mathcal{Y}} = 2$ (prey and predator) rather than $d_{\mathcal{Y}} = 1$, so every decoder emits $2d_{\mathcal{Y}}$ outputs and the output-pathway inputs are two-dimensional. Second, the grid-based models (ConvCNP and SConvCNP) discretize the functional embedding at 48 points per unit (rather than 64), with grid margin 0.5 (rather than 0.1) and Gaussian set-convolution length scales initialized to $2/48$ (rather than $2/64$); the grid size remains constrained to a multiple of 64. All remaining models are unaffected.

**Data-Generating Process.** We adopt the stochastic Lotka–Volterra simulation framework of Bruinsma et al. (2023). Let $U_t$ and $V_t$ denote the prey and predator populations at time $t$, respectively. Their dynamics evolve according to

$$\mathrm{d}U_t = \alpha U_t \,\mathrm{d}t - \beta U_t V_t \,\mathrm{d}t + \sigma U_t^{\nu} \,\mathrm{d}B_t^{(1)},$$
$$\mathrm{d}V_t = -\gamma V_t \,\mathrm{d}t + \delta U_t V_t \,\mathrm{d}t + \sigma V_t^{\nu} \,\mathrm{d}B_t^{(2)},$$

where $B_t^{(1)}$ and $B_t^{(2)}$ are independent Brownian motions. In the deterministic component of the dynamics, $U_t$ grows exponentially at rate $\alpha$, while $V_t$ decays at rate $\gamma$. The bilinear interaction terms $\beta U_t V_t$ and $\delta U_t V_t$ model predation and the corresponding transfer of biomass from prey to predators. To account for stochastic fluctuations commonly observed in empirical population counts, the dynamics are augmented with multiplicative noise terms $\sigma U_t^{\nu} \,\mathrm{d}B_t^{(1)}$ and $\sigma V_t^{\nu} \,\mathrm{d}B_t^{(2)}$, where $\sigma$ controls the noise magnitude and $\nu$ determines how the variability scales with population size.

The parameters are sampled independently for each trajectory from the following ranges: $\alpha \sim \mathcal{U}[0.2, 0.8)$, $\beta \sim \mathcal{U}[0.04, 0.08)$, $\gamma \sim \mathcal{U}[0.8, 1.2)$, $\delta \sim \mathcal{U}[0.04, 0.08)$, $\sigma \sim \mathcal{U}[0.5, 10.0)$, and $\nu = 1/6$. Initial populations are sampled as $U_0 \sim \mathcal{U}[5, 100)$ and $V_0 \sim \mathcal{U}[5, 100)$. Additionally, a global scale factor is sampled from $\mathcal{U}[1, 5)$ and applied multiplicatively to the populations. Populations are capped at 500 to prevent numerical divergence.

Trajectories are integrated using the Euler–Maruyama method over the time interval $[-10, 100]$ with 5000 integration steps (step size $\approx 0.022$); the integrated trajectories are then recorded at a temporal resolution of 0.05. The burn-in period $[-10, 0]$ is discarded. Both populations are rescaled by a factor of 0.01 and time values by a factor of 0.1, yielding effective ranges of approximately $[0, 5]$ for populations and $[0, 10]$ for time.

**Tasks and Splits.** Training data are generated on the fly by sampling trajectories from the stochastic model; a pool of 2048 trajectories is maintained and refreshed periodically. The numbers of context and query points are sampled per batch as $|\mathcal{D}_c|, |\mathcal{D}_q| \sim \mathcal{U}[5, 50]$ and shared across all tasks in the batch, with point locations sampled uniformly within each task from the rescaled time interval $[0, 10]$. Validation uses a fixed set of 8,000 deterministic tasks (pinned random seed) drawn from the same simulator, with $|\mathcal{D}_c| \sim \mathcal{U}[5, 50]$ and $|\mathcal{D}_q| = 128$. We report on two test sets, each of 64,000 tasks with a fixed random seed: a *simulated* set that follows the validation procedure ($|\mathcal{D}_c| \sim \mathcal{U}[5, 50]$, $|\mathcal{D}_q| = 128$), and a *real* sim-to-real set built from the classical Hudson Bay hare–lynx dataset (Leigh, 1968), which records annual trapping counts of snowshoe hares and Canada lynx from 1845 to 1935 (91 time points). The real data are rescaled with the same population and time factors as the simulated data; the context count $|\mathcal{D}_c| \sim \mathcal{U}[5, 50]$ is shared across all tasks in a batch, with the remaining points used as queries.

**Training Protocol.** All models are trained for 250 epochs using the AdamW optimizer with learning rate $5 \times 10^{-4}$ and gradient clipping at maximum $\ell_2$-norm 0.5. The learning rate is annealed via a cosine schedule with minimum learning rate $10^{-6}$. Each epoch processes 16,000 training tasks in batches of 32. Data loading used a single worker process for both training and evaluation.

**Evaluation Protocol.** Validation is performed every epoch, and the best checkpoint is selected based on validation log-likelihood. Each selected model is then evaluated on both the simulated and the real (Hudson Bay) test sets described above.

**Qualitative Predictions.** Figure 4 shows example predictions for all nine models on a shared simulated test trajectory and a shared real Hudson Bay hare–lynx trajectory. Within each column every model sees the same context split, so the panels are directly comparable: blue curves show the predictive mean $\pm 2$ standard deviations, black points the context, and red the dense ground-truth trajectory.

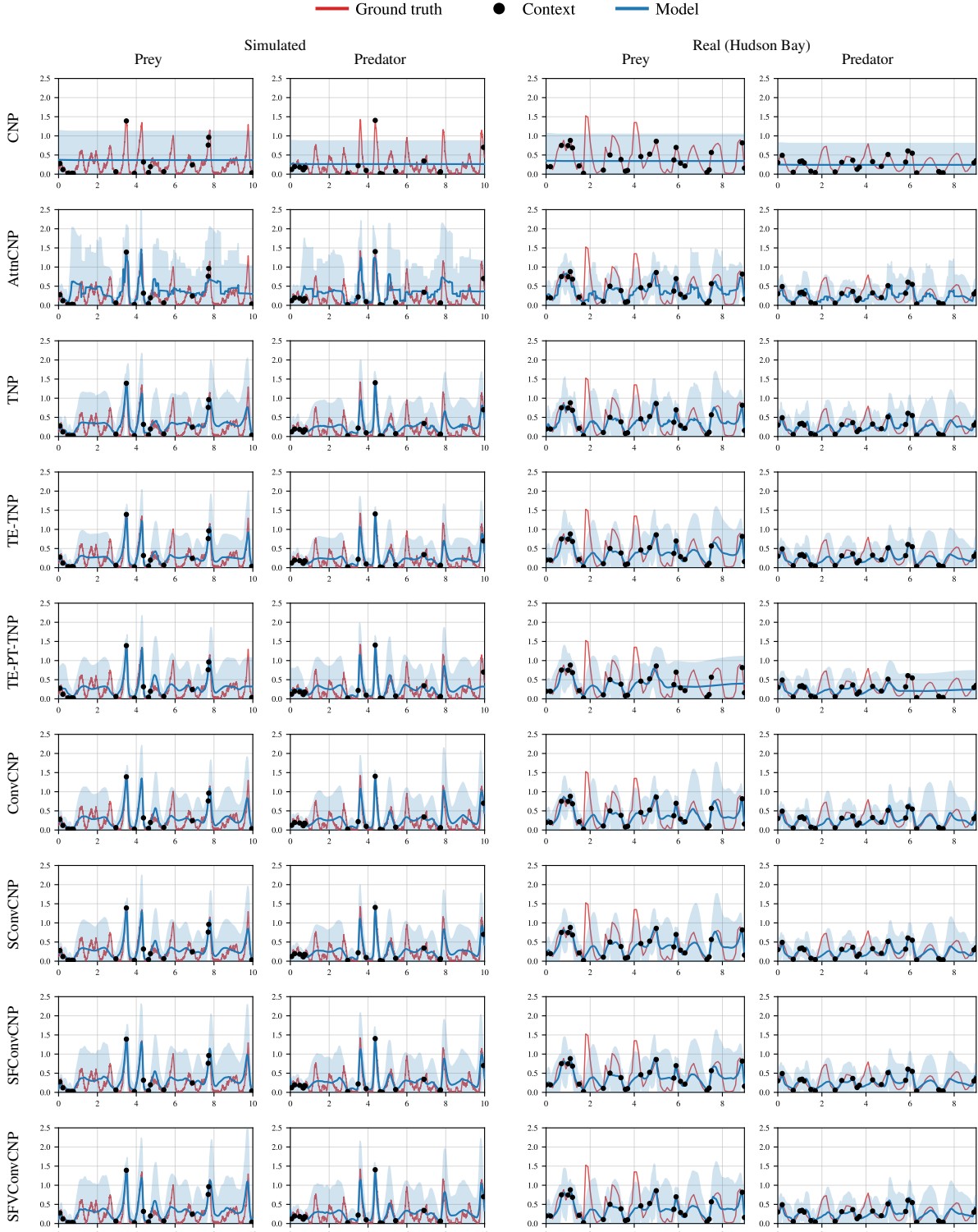

*Figure 4.* Example predictions on the predator–prey benchmark. Each row shows one model; columns 1–2 share a simulated test trajectory (prey and predator populations) and columns 3–4 share a real Hudson Bay hare–lynx trajectory. All models in a column see the same context split. Blue: predictive mean $\pm 2$ std. Black: context points. Red: ground-truth dense trajectory.

## E.2. Two-Dimensional Image Regression

**Model Architectures.** This section describes the architectural adaptations used for the image completion experiments (Section 4.2). Table 13 provides the parameter counts for all models. All models output factorized Gaussian predictive distributions over pixels and RGB channels, parameterized by a mean and a scale. Since pixel intensities are normalized to $[0, 1]$, the predictive mean is passed through a sigmoid activation. The scale is parameterized in pre-softplus form, transformed via a softplus non-linearity, scaled by $0.99$, and offset with a minimum noise floor of $0.01$ to ensure numerical stability.

*Table 13.* Trainable parameter counts for models used in the image completion experiments.

| Model | Parameters |
|---|---|
| CNP | 52.66M |
| AttnCNP | 51.57M |
| TNP | 51.46M |
| TE-PT-TNP | 55.29M |
| Grid-ConvCNP | 55.56M |
| Grid-SConvCNP | 51.18M |
| SFConvCNP | 50.44M |
| SFVConvCNP | 54.51M |

**Conditional Neural Process (CNP).** The CNP architecture follows the design described in Appendix E.1.1, with all embedding dimensions and MLP widths increased to 1664 (retaining the two-hidden-layer input and output encoder MLPs, the six-hidden-layer combiner and decoder MLPs, and mean-pooling context aggregation).

**Attentive Conditional Neural Process (AttnCNP).** The AttnCNP is based on the architecture in Appendix E.1.1, using embedding dimensions and MLP widths of 768. Both self-attention over context representations and cross-attention for query updates employ 12 attention heads, each with head dimension 64 and feedforward width 4096 (retaining the four self-attention layers, the single cross-attention layer, pre-layer normalization with residual connections, and the six-hidden-layer decoder MLP).

**Transformer Neural Process (TNP).** The TNP follows the architecture described in Appendix E.1.1, with the feedforward width increased to 3072 (embedding dimensions and MLP widths remain at 512, 8 attention heads with head dimension 64, six transformer layers with unshared parameters).

**Translation-Equivariant Pseudo-Token Transformer Neural Process (TE-PT-TNP).** The TE-PT-TNP follows the design in Appendix E.1.1, with the number of pseudo-tokens set to 96. The MLPs used within the translation-equivariant attention modules for computing attention logits consist of two hidden layers of width 64. All other MLP widths and embedding dimensions are increased to 512, while the feedforward width within each transformer layer remains at 2048 (the six transformer layers, the 8 attention heads with head dimension 64, and the Induced Set Attention design are inherited unchanged).

**On-the-grid Convolutional Conditional Neural Process (Grid-ConvCNP).** Because the data lie on a regular grid, we adopt the on-the-grid implementation of the ConvCNP (Gordon et al., 2019), which we refer to as Grid-ConvCNP. This formulation avoids explicitly constructing a continuous functional embedding and subsequently discretizing it on a dense grid, which is computationally expensive. Let I denote an incomplete image in which unobserved pixels are filled with dummy values, and let $M_c$ be a binary mask indicating observed (context) pixels. For multi-channel images, $M_c$ is broadcast along the channel dimension. As in Appendix E.1.1, the convolutional deep set module produces two outputs: (i) a density channel encoding the spatial distribution of context pixels, and (ii) a kernel-smoothed representation of the observed values. The kernel is implemented as a 2D convolutional layer with kernel size 11, $d_{\mathcal{Y}}$ input channels, and 256 output channels, without bias. *Nonnegativity* of the kernel is enforced by taking the absolute value of the learned weights during the forward pass, following Gordon et al. (2019). The density channel is obtained by convolving this nonnegative kernel with the mask $M_c$. The kernel-smoothed representation is computed by first multiplying I elementwise with $M_c$, thereby zeroing non-context pixels, and then applying the same convolution. Unlike the ConvCNP described in Appendix E.1.1, we omit

normalization by the density channel. The density channel and the kernel-smoothed feature channels are concatenated and processed pointwise by an MLP with two hidden layers of width 256; as in the off-grid ConvCNP, this pointwise mixing makes the order of concatenation immaterial. The resulting features are passed to a ResNet-style CNN (He et al., 2016) comprising six residual convolutional blocks (kernel size 15, 196 channels), using the implementation of Bruinsma et al. (2023). Finally, embeddings corresponding to query pixels are gathered and fed into a decoder MLP with two hidden layers of width 256.

**On-the-grid Spectral Convolutional Conditional Neural Process (Grid-SConvCNP).**    The Grid-SConvCNP replaces the ResNet backbone of the Grid-ConvCNP with a Fourier Neural Operator. The grid encoder is identical to Grid-ConvCNP, so the density and feature channels are concatenated as there, with the order immaterial for the same reason. The FNO backbone consists of six residual layers with 128 channels and retains modes [16, 16] along the two spatial dimensions.

**Set Fourier Convolutional Conditional Neural Process (SFConvCNP).**    The SFConvCNP is adapted from the architecture in Appendix E.1.1, with all embedding channels and MLP widths set to 384 and feedforward subnetwork widths set to 1536. The frequency grid is two-dimensional, with per-dimension parameters $\xi_{\max} = 4.8$ and $\Delta_{\mathcal{F}} = 0.2$. As in the one-dimensional models, the $2 \times 384 = 768$ density and feature channels are stacked blockwise and partitioned into 128 contiguous groups of 6, each mapped to 3 output channels.

**Set Fourier Volterra Convolutional Conditional Neural Process (SFVConvCNP).**    The SFVConvCNP follows the architecture described in Appendix E.1.1, with embedding channels and MLP widths set to 256 and feedforward subnetwork widths set to 1024. The frequency grid is two-dimensional, with per-dimension parameters $\xi_{\max} = 4.75$ and $\Delta_{\mathcal{F}} = 0.25$. The Volterra rank is set to $R = 2$ and the model employs six SFVConvBlocks (instead of five). Within each Volterra branch, each underlying SFConv stacks its $2 \times 256 = 512$ density and feature channels blockwise and partitions them into 128 contiguous groups of 4, each mapped to 2 output channels, instead of the 4 groups used in Appendix E.1.1.

**Datasets.**    We evaluate image completion on CIFAR-10 (Krizhevsky et al., 2009) and SVHN (Netzer et al., 2011). CIFAR-10 contains 50,000 training images and 10,000 test images, while SVHN provides 73,257 training images and 26,032 test images. Although SVHN includes an additional set of 531,131 extra images, these are not used in this work. All images have a fixed spatial resolution of $32 \times 32$ with three RGB channels. Pixel coordinates are defined by uniformly discretizing the interval $[-1, 1]$ into 32 points along each spatial axis. Pixel intensities are normalized independently per channel to lie in $[0, 1]$.

**Tasks and Splits.**    Each task corresponds to a single image. The number of context pixels is sampled as $|\mathcal{D}_c| \sim \mathcal{U}[5, 512]$ and shared across all tasks in a batch, with the remaining pixels treated as query points ($|\mathcal{D}_q| = 1024 - |\mathcal{D}_c|$); context and query pixels are drawn uniformly without replacement from the image grid for each task. Validation tasks are constructed from a held-out split of 10% of the training images (disjoint from those used for training) using the same sampling procedure with a fixed random seed, and test tasks from the official test splits (10,000 images for CIFAR-10 and 26,032 for SVHN).

**Training Protocol.**    All models are trained for 150 epochs using the AdamW optimizer with learning rate $5 \times 10^{-4}$ and gradient clipping at maximum $\ell_2$-norm 0.5. The learning rate is annealed via a cosine schedule with minimum learning rate $10^{-6}$. Each epoch consists of batches of 32 tasks, where each task corresponds to a single image. Data loading used a single worker process for both training and evaluation.

**Evaluation Protocol.**    Validation is performed every epoch, and the best checkpoint is selected based on validation log-likelihood. Test evaluation is run in batches of 32.

**Qualitative Predictions.**    Figure 5 shows example predictions on a single held-out task per dataset for all eight image-completion models. Within each dataset column, every model is evaluated on the same image and same context split so that panels are directly comparable. The reference row at the top shows the ground-truth image and the context mask (blue pixels denote unobserved locations); subsequent rows show each model's predictive mean and channel-aggregated standard deviation over the full image grid. The mean and standard-deviation panels display the model's prediction at every pixel, including at the observed context positions: the values shown at those positions are model outputs (conditioned on the context), not the original ground-truth pixel values.

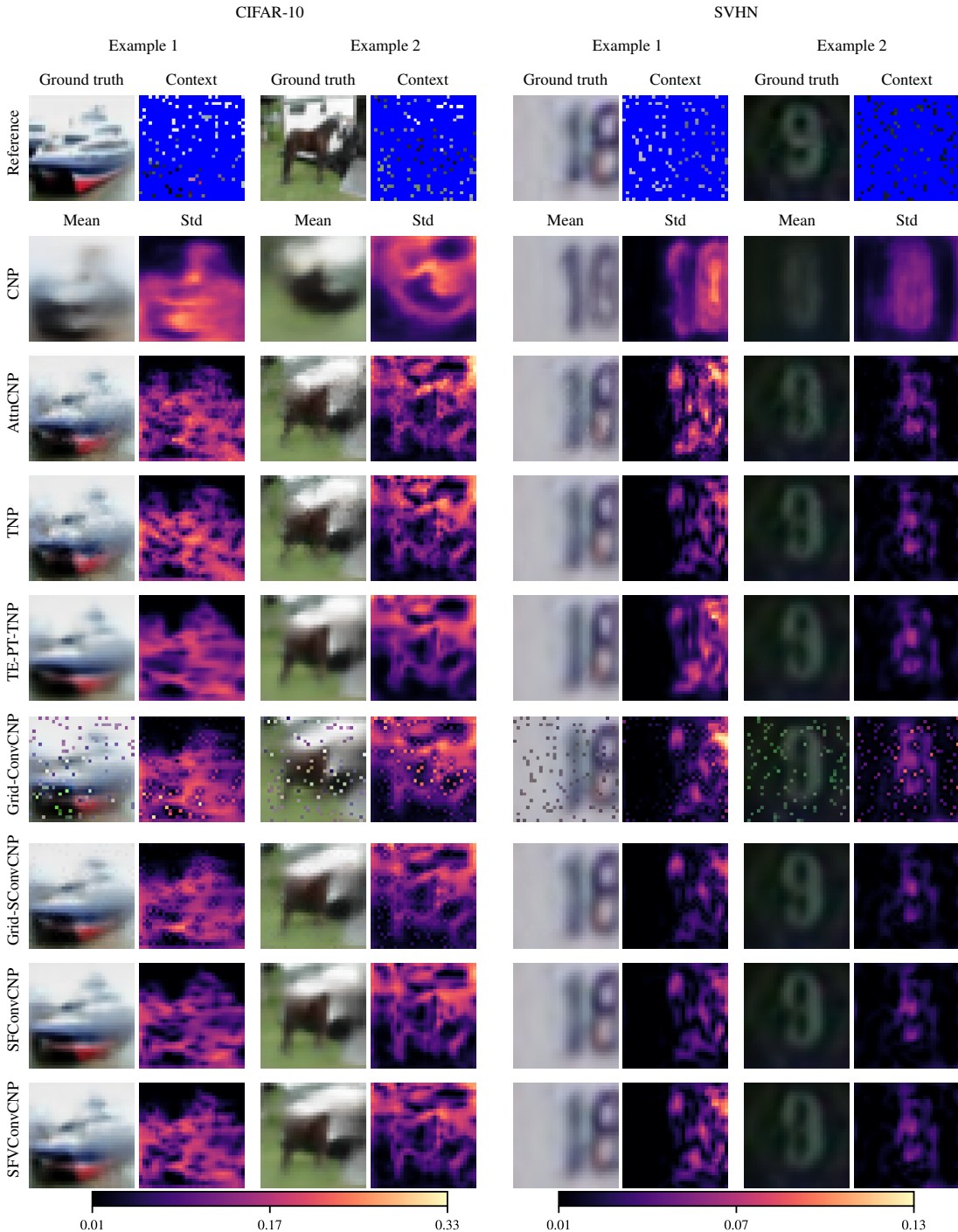

*Figure 5.* Example predictions on the image-completion benchmarks. For each dataset, two held-out test images are shown side-by-side. The top row gives the ground-truth image and the context mask (blue pixels are unobserved); subsequent rows show, for each model, the predictive mean and the channel-aggregated predictive standard deviation. The mean and standard-deviation panels show the model's prediction at every pixel, including at the observed context positions; the values shown at those positions are model outputs (conditioned on the context), not the original observed pixel values. All models on a given example see the same context split. Standard-deviation panels within each dataset share a common color scale (bottom colorbar) so that uncertainty magnitudes are directly comparable across both models and examples.

## E.3. Three-Dimensional Spatiotemporal Regression

### E.3.1. KOLMOGOROV FLOW REGRESSION

**Model Architectures.** This section describes the architectural adaptations used for the three-dimensional regression experiments (Section 4.3.1). Parameter counts for all models are reported in Table 14. All models predict factorized Gaussian distributions over query points and over the components of the two-dimensional velocity vector field at each spatiotemporal location, parameterized by a mean and a scale. The scale is produced in pre-softplus form and transformed using a softplus non-linearity, with an added minimum noise floor of $10^{-6}$.

*Table 14.* Trainable parameter counts for models used in the Kolmogorov flow experiments.

| Model | Parameters |
| --- | --- |
| CNP | 100.92M |
| AttnCNP | 98.29M |
| TNP | 101.56M |
| TE-PT-TNP | 100.29M |
| Grid-ConvCNP | 99.86M |
| Grid-SConvCNP | 98.65M |
| SFConvCNP | 95.82M |
| SFVConvCNP | 96.22M |

**Conditional Neural Process (CNP).** The CNP follows the architecture described in Appendix E.1.1, with all embedding dimensions and MLP widths uniformly increased to 2304 (retaining the two-hidden-layer input and output encoder MLPs, the six-hidden-layer combiner and decoder MLPs, and mean-pooling context aggregation).

**Attentive Conditional Neural Process (AttnCNP).** The AttnCNP is based on the architecture in Appendix E.1.1, using embedding dimensions and MLP widths of 1152. Both self-attention over context representations and cross-attention for query updates employ 18 attention heads, each with head dimension 64 and feedforward width 4608 (retaining the four self-attention layers, the single cross-attention layer, pre-layer normalization with residual connections, and the six-hidden-layer decoder MLP).

**Transformer Neural Process (TNP).** The TNP follows the architecture in Appendix E.1.1, with embedding dimensions and MLP widths set to 768. All attention modules use 12 attention heads, each with head dimension 64 and feedforward width 3840 (retaining the six transformer layers with unshared parameters and the two-hidden-layer token-embedding and decoder MLPs).

**Translation-Equivariant Pseudo-Token Transformer Neural Process (TE-PT-TNP).** The TE-PT-TNP follows the design in Appendix E.1.1, with the number of pseudo-tokens set to 196. The MLPs used within translation-equivariant attention modules to compute attention logits consist of two hidden layers of width 64. All other MLP widths and embedding dimensions are set to 640. All attention modules employ 10 attention heads, each with head dimension 64 and feedforward width 3200 (the six transformer layers and the Induced Set Attention design are inherited unchanged).

**On-the-grid Convolutional Conditional Neural Process (Grid-ConvCNP).** Since the Kolmogorov flow data lie on a regular spatiotemporal grid, we adopt an on-the-grid implementation analogous to Grid-ConvCNP described in Appendix E.2. The grid encoder uses a 3D convolutional layer with kernel size 9 and 256 output channels. Nonnegativity of the kernel weights is enforced by taking absolute values. The CNN backbone is a ResNet-style architecture comprising six residual convolutional blocks (kernel size 9, 141 channels). An MLP with two hidden layers of width 256 processes the grid encoder output before passing it to the CNN.

**On-the-grid Spectral Convolutional Conditional Neural Process (Grid-SConvCNP).** The Grid-SConvCNP replaces the ResNet backbone of Grid-ConvCNP with a Fourier Neural Operator. The grid encoder is identical to Grid-ConvCNP. The FNO backbone consists of six residual layers with 180 channels and retains 5 Fourier modes along each spatiotemporal dimension.

**Set Fourier Convolutional Conditional Neural Process (SFConvCNP).** The SFConvCNP is adapted from the architecture in Appendix E.1.1, with embedding channels and MLP widths set to 176 and feedforward subnetwork widths set to 512. The frequency grid is three-dimensional, with per-dimension parameters $\xi_{\max} = 4.25$ and $\Delta_{\mathcal{F}} = 0.25$. All SFConv operations are grouped at the finest granularity: the $2 \times 176 = 352$ density and feature channels are stacked blockwise and partitioned into 176 contiguous groups of two, each mapped to a single output channel. The model uses six SFConvBlocks with output feature mixing enabled.

**Set Fourier Volterra Convolutional Conditional Neural Process (SFVConvCNP).** The SFVConvCNP follows the architecture described in Appendix E.1.1, with embedding channels and MLP widths set to 52 and feedforward subnetwork widths set to 208. The frequency grid is three-dimensional, with per-dimension parameters $\xi_{\max} = 3.75$ and $\Delta_{\mathcal{F}} = 0.25$. The model employs six SFVConvBlocks (instead of five) with Volterra rank $R = 2$. All SFConv operations are grouped at the finest granularity: within each Volterra branch, the $2 \times 52 = 104$ density and feature channels are stacked blockwise and partitioned into 52 contiguous groups of two, each mapped to a single output channel. Output feature mixing is disabled.

**Data-Generating Process.** We adopt the Kolmogorov flow benchmark of Ashman et al. (2024a), with adaptations to suit our experimental setting. Simulation trajectories are generated following the setup of Rozet & Louppe (2023). The following description of the governing equations and simulation setting closely follows their presentation and is included here for self-containment. The dynamics of an incompressible Newtonian fluid are governed by the Navier–Stokes equations

$$\partial_t \mathbf{u}(\mathbf{x}, t) = -\big(\mathbf{u}(\mathbf{x}, t) \cdot \nabla\big)\mathbf{u}(\mathbf{x}, t)$$
$$+ \frac{1}{Re}\nabla^2 \mathbf{u}(\mathbf{x}, t) - \frac{1}{\rho}\nabla p(\mathbf{x}, t) + \mathbf{f}(\mathbf{x}), \tag{22}$$
$$\nabla \cdot \mathbf{u}(\mathbf{x}, t) = 0,$$

where $\mathbf{u} \colon \Omega \times [0, T] \to \mathbb{R}^2$ denotes the time-dependent velocity field of the fluid, mapping each spatial location $\mathbf{x} \in \Omega$ and time $t$ to a two-dimensional velocity vector. The domain $\Omega \subset \mathbb{R}^2$ represents the spatial region occupied by the fluid (not to be confused with the input space $\mathcal{X} = \mathbb{R}^3$ of the corresponding regression task, which comprises space and time). The incompressibility constraint $\nabla \cdot \mathbf{u} = 0$ enforces local volume conservation.

The right-hand side of (22) consists of four terms with standard physical interpretations. The non-linear advection term $-(\mathbf{u} \cdot \nabla)\mathbf{u}$ describes self-transport of momentum by the flow. The viscous diffusion term $\frac{1}{Re}\nabla^2 \mathbf{u}$ models momentum dissipation due to viscosity, controlled by the Reynolds number $Re$. The pressure gradient term $-\frac{1}{\rho}\nabla p$, where $p \colon \Omega \times [0, T] \to \mathbb{R}$ is the scalar pressure field and $\rho$ is the fluid density, enforces incompressibility by redistributing momentum instantaneously. Finally, $\mathbf{f} \colon \Omega \to \mathbb{R}^2$ denotes an external body force that injects energy into the system.

Following Kochkov et al. (2021), we consider a two-dimensional periodic domain $\Omega = [0, 2\pi]^2$ with periodic boundary conditions, a constant density $\rho = 1$, and a high Reynolds number $Re = 10^3$, corresponding to a turbulent regime. The external forcing $\mathbf{f}$ is chosen to be Kolmogorov forcing with linear damping (Chandler & Kerswell, 2013; Boffetta & Ecke, 2012), which sustains statistically stationary turbulence.

We solve (22) using the `jax-cfd` library (Kochkov et al., 2021) on a $256 \times 256$ spatial grid. All initial conditions are sampled from the statistically stationary regime of the flow. During simulation, snapshots of the velocity field $\mathbf{u}$ are recorded and subsequently coarsened to a $64 \times 64$ resolution. The time interval between consecutive snapshots is $\Delta = 0.2$ (in nondimensionalized time units), corresponding to 82 integration substeps.

**Tasks and Splits.** We generate 1024 independent trajectories, each consisting of 64 consecutive velocity field states. These trajectories are partitioned into training, validation, and test sets with proportions of 0.8, 0.1, and 0.1, respectively (819, 102, and 103 trajectories). All data are normalized using the mean and standard deviation computed from the training set. During training and validation, each task is constructed by randomly cropping a uniform $16 \times 16 \times 16$ space–time grid from a single trajectory, with validation crops fixed by a pinned random seed for reproducibility. The proportion of context points, shared across all tasks in a batch, is sampled uniformly as $|\mathcal{D}_c|/|\mathcal{D}| \sim \mathcal{U}[0.05, 0.25]$, with the remaining points used as query points ($|\mathcal{D}_q| = 4096 - |\mathcal{D}_c|$). At test time, by contrast, we evaluate exhaustively: each test trajectory (64 time steps on a $64 \times 64$ spatial grid) is tiled into non-overlapping $16 \times 16 \times 16$ crops, using a stride equal to the crop size along the temporal, height, and width axes. This yields $4 \times 4 \times 4 = 64$ crops per trajectory and $103 \times 64 = 6{,}592$ test tasks, covering every space–time location exactly once.

**Training Protocol.**     All models are trained for 250 epochs using the AdamW optimizer with gradient clipping at maximum $\ell_2$-norm 0.5. The learning rate is annealed via a cosine schedule with minimum learning rate $10^{-6}$. The base learning rate is $5 \times 10^{-4}$, except for AttnCNP, TNP, and TE-PT-TNP, which use $10^{-4}$ to stabilize training. Each epoch consists of mini-batches of 6 tasks. Data loading used 4 worker processes for both training and evaluation.

**Evaluation Protocol.**     Validation uses randomly cropped tasks fixed by a pinned random seed and the same context/query sampling as training, whereas test evaluation uses the deterministic non-overlapping tiling described above; both are applied to the corresponding dataset splits. The best checkpoint is selected based on validation log-likelihood.

**Qualitative Predictions.**     Figure 6 shows example predictions on two held-out test tasks for the eight Kolmogorov models. Within each example column a single random time step is selected and held fixed across the model rows, so the predictive maps in a row are directly comparable. Because the Kolmogorov velocity field is two-dimensional, $\mathbf{u} = (u, v)$, each model row shows four panels per example: the predictive mean of the $x$-component $u$, its standard deviation, the predictive mean of the $y$-component $v$, and its standard deviation. The reference row at the top shows the ground-truth $u$ and $v$ alongside their context-only views (non-context cells in white). The predictive panels display the model's output at every cell, including at the observed context positions; values shown at those positions are model outputs (conditioned on the context), not the original observations. Within each example, the mean panels for $u$ and $v$ share a single color scale with the ground-truth and context references, and the standard-deviation panels for $u$ and $v$ share a single color scale across both components; the two colorbars at the bottom of each example calibrate these two scales, respectively.

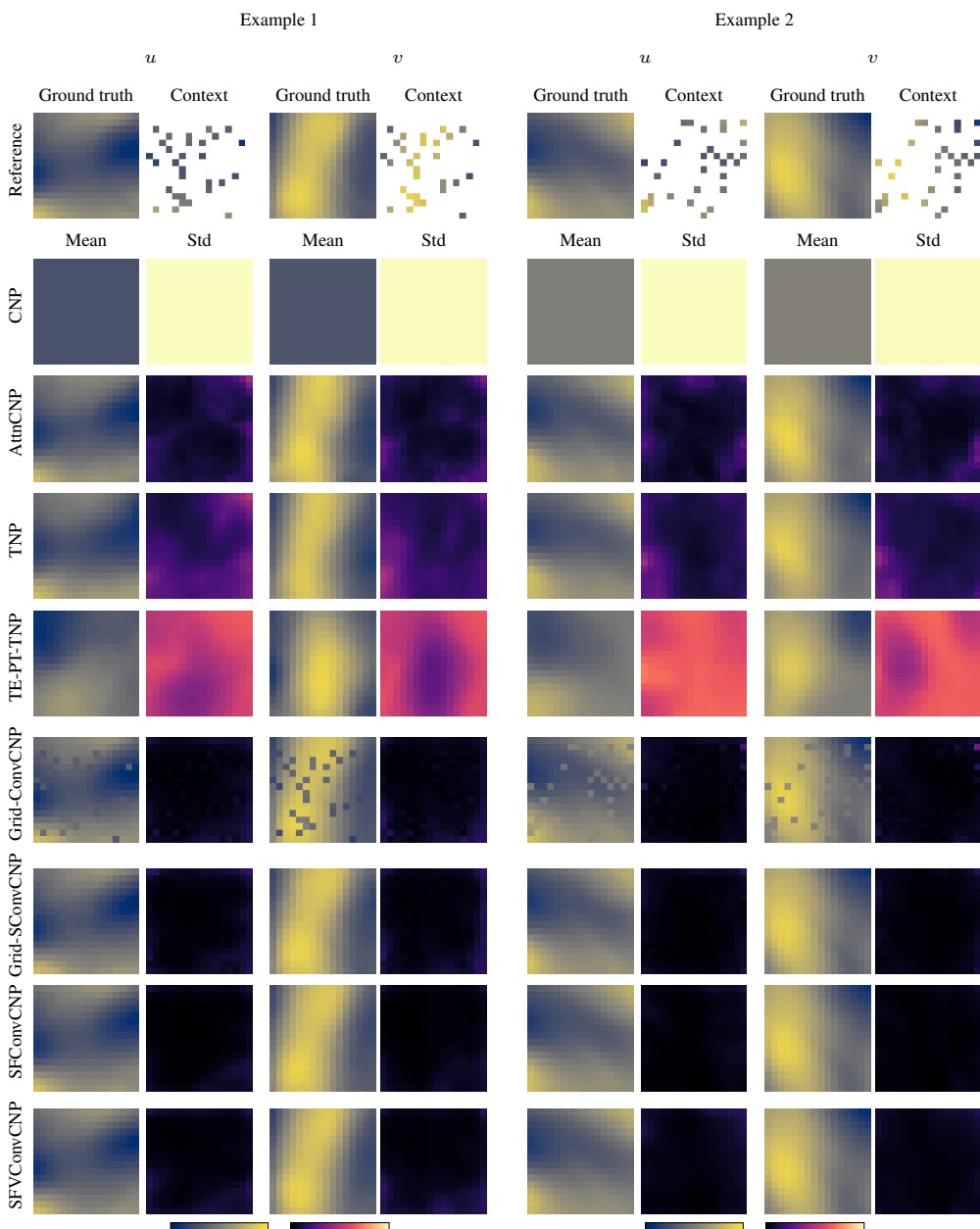

*Figure 6.* Example predictions on the Kolmogorov flow benchmark. Each of the two column groups corresponds to one held-out test task with a single randomly selected time step held fixed across model rows; within each group, the four columns show the $x$-component $u$ (mean and standard deviation) followed by the $y$-component $v$ (mean and standard deviation). The top row gives the ground-truth velocity components $u$ and $v$ together with their context masks (non-context cells in white). Subsequent rows show, for each model, the predictive mean and standard deviation of each velocity component. The predictive panels display the model's output at every cell, including at the observed context positions; values shown at those positions are model outputs (conditioned on the context), not the original observations. Within each example, the mean panels for $u$ and $v$ share a single color scale with the ground-truth and context references (left colorbar), and the standard-deviation panels for $u$ and $v$ share a single color scale (right colorbar).

E.3.2. ERA5 CLIMATE REGRESSION

**Model Architectures.** This section describes the architectures of all models used in the ERA5 climate regression experiments (Section 4.3.2). Parameter counts for each model are reported in Table 15. All models predict factorized Gaussian distributions over query points, parameterized by a mean and a scale. The scale is produced in pre-softplus form and transformed via a softplus non-linearity, with an added minimum noise floor of $10^{-6}$.

*Table 15.* Trainable parameter counts for models used in the ERA5 experiments.

| Model | Parameters |
|---|---|
| CNP | 100.92M |
| AttnCNP | 98.29M |
| TNP | 101.56M |
| TE-PT-TNP | 100.41M |
| SFConvCNP | 95.82M |
| SFVConvCNP | 96.22M |

**Conditional Neural Process (CNP).** The CNP follows the architecture described in Appendix E.1.1, with all embedding dimensions and MLP widths set to 2304 (retaining the two-hidden-layer input and output encoder MLPs, the six-hidden-layer combiner and decoder MLPs, and mean-pooling context aggregation).

**Attentive Conditional Neural Process (AttnCNP).** The AttnCNP is based on the architecture in Appendix E.1.1, using embedding dimensions and MLP widths of 1152. Both self-attention and cross-attention modules employ 18 attention heads, each with head dimension 64 and feedforward width 4608. Context representations are refined using four layers of multi-head self-attention (with the single cross-attention layer, pre-layer normalization with residual connections, and the six-hidden-layer decoder MLP inherited unchanged).

**Transformer Neural Process (TNP).** The TNP follows the architecture described in Appendix E.1.1, with embedding dimensions and MLP widths set to 768. All attention modules use 12 attention heads, each with head dimension 64 and feedforward width 3840. The model uses six transformer layers with unshared parameters (with the two-hidden-layer token-embedding and decoder MLPs inherited unchanged).

**Translation-Equivariant Pseudo-Token Transformer Neural Process (TE-PT-TNP).** The TE-PT-TNP follows the design in Appendix E.1.1, with embedding dimensions and MLP widths set to 640. The model uses 384 pseudo-tokens. All attention modules employ 10 attention heads with head dimension 64 and feedforward width 3200. The MLPs within translation-equivariant attention modules for computing attention logits consist of two hidden layers of width 64 (the six transformer layers and the Induced Set Attention design are inherited unchanged).

**On-the-grid Baselines (Grid-ConvCNP and Grid-SConvCNP).** Since the ERA5 data lie on a regular spatiotemporal grid, we also attempted to train the on-the-grid ConvCNP variants used in the lower-dimensional benchmarks (Appendix E.2). The Grid-ConvCNP configuration we tried uses a 3D convolutional grid encoder (kernel size 5, 256 output channels, with nonnegativity enforced by taking absolute values of the weights) feeding a two-hidden-layer MLP of width 256 followed by a ResNet-style backbone of six residual convolutional blocks (kernel size 5, 384 channels). The Grid-SConvCNP configuration replaces this ResNet backbone with a Fourier Neural Operator (six residual layers, 195 channels, retaining modes $[3, 6, 6]$ along the temporal, latitudinal, and longitudinal dimensions, respectively), keeping the grid encoder unchanged. Under our training protocol, however, neither on-the-grid model converged on the ERA5 task: training was unstable and the validation log-likelihood failed to improve meaningfully over the run. We additionally experimented with smaller configurations (fewer channels and residual blocks), but these performed no better. We therefore omit the on-the-grid baselines from the ERA5 results.

**Set Fourier Convolutional Conditional Neural Process (SFConvCNP).** The SFConvCNP is adapted from the architecture in Appendix E.1.1, with embedding channels set to 176 and feedforward subnetwork widths set to 512. The frequency grid is three-dimensional with per-dimension parameters $\xi_{\max} = 4.25$ and $\Delta_{\mathcal{F}} = 0.25$. All SFConv operations are grouped at the finest granularity: the $2 \times 176 = 352$ density and feature channels are stacked blockwise and partitioned into 176

contiguous groups of two, each mapped to a single output channel. The model uses six SFConvBlocks with output feature mixing enabled.

**Set Fourier Volterra Convolutional Conditional Neural Process (SFVConvCNP).** The SFVConvCNP follows the architecture described in Appendix E.1.1, with embedding channels set to 52 and feedforward subnetwork widths set to 208. The frequency grid is three-dimensional with per-dimension parameters $\xi_{\max} = 3.75$ and $\Delta_{\mathcal{F}} = 0.25$. All SFConv operations are grouped at the finest granularity: within each Volterra branch, the $2 \times 52 = 104$ density and feature channels are stacked blockwise and partitioned into 52 contiguous groups of two, each mapped to a single output channel. The model uses six SFVConvBlocks with Volterra rank $R = 2$. Output feature mixing is disabled.

**Datasets.** We adopt the ERA5 climate benchmark of Ashman et al. (2024a), with adaptations to suit our experimental setting. We use the ERA5 reanalysis dataset (Hersbach et al., 2020), accessed via the Copernicus Climate Data Store. All data correspond to the year 2019. Input coordinates consist of one temporal and two spatial dimensions (time, latitude, longitude), yielding $\mathcal{X} = \mathbb{R}^3$ with $d_{\mathcal{X}} = 3$. The output is a scalar atmospheric variable ($\mathcal{Y} = \mathbb{R}$, $d_{\mathcal{Y}} = 1$). Data are normalized using statistics computed from the training subset.

**Tasks and Splits.** To probe spatial generalization, we adopt a spatial split. The training and validation sets are drawn from the same central European window (latitude $[42°, 53°]$, longitude $[8°, 28°]$, covering central-to-eastern Europe). Each trained model is then evaluated independently on three test regions, defined by the following latitude/longitude windows:

- **Center**: latitude $[42°, 53°]$, longitude $[8°, 28°]$—identical to the training/validation window and therefore an in-distribution reference. The split is purely spatial (all data are from 2019, with no temporal hold-out), so the test crops are drawn from the same region seen during training, only with a different fixed random seed.

- **West**: latitude $[42°, 53°]$, longitude $[-4°, 8°]$—shifted westward in longitude.

- **North**: latitude $[53°, 62°]$, longitude $[8°, 28°]$—shifted northward in latitude.

The West and North windows are geographically disjoint from the training region and thus measure out-of-region generalization, whereas Center quantifies in-region performance. Each task is constructed by randomly cropping a spatiotemporal subgrid of size $6 \times 16 \times 16$ (time $\times$ latitude $\times$ longitude) from the data. The temporal dimension comprises 6 time steps sampled at a temporal stride of 5. The proportion of context points, shared across all tasks in a batch, is sampled as $|\mathcal{D}_c|/|\mathcal{D}| \sim \mathcal{U}[0.05, 0.33]$, with the remaining points used as query. Grid cells with at least 50% missing values are excluded. Validation uses 512 deterministic tasks drawn from the training region; for each of the three test regions (Center, West, North), the test set consists of 16,000 deterministic tasks drawn from that region.

**Training Protocol.** All models are trained for 250 epochs using the AdamW optimizer with learning rate $5 \times 10^{-4}$ and gradient clipping at maximum $\ell_2$-norm 0.5. The learning rate follows a cosine annealing schedule with minimum learning rate $10^{-6}$. Each epoch processes 8,000 training tasks in batches of 8. Data loading used 16 worker processes for both training and evaluation.

**Evaluation Protocol.** Validation is performed every epoch, and the best checkpoint is selected based on validation log-likelihood. Each model is evaluated independently on the Center, West, and North test regions, with metrics reported per region.

**Qualitative Predictions.** Figure 7 shows example predictions on one held-out task per evaluation region for the six ERA5 models. The three columns correspond to the evaluation regions of Europe (Center, West, and North), defined by the latitude/longitude windows in Appendix E.3.2. Within each region column, a single random time step is selected and held fixed across the model rows, so the predictive maps in a row are directly comparable. The reference row at the top shows the ground-truth field (cells with missing data in red) and the context mask (non-context cells in white); subsequent rows show, for each model, the predictive mean (shared color scale with the ground truth) and the predictive standard deviation. The mean and standard-deviation panels display the model's prediction at every cell, including at the observed context positions: the values shown at those positions are model outputs (conditioned on the context), not the original ground-truth observations. Standard-deviation panels within each region share a common color scale (bottom colorbar) so that uncertainty magnitudes are directly comparable across both models and regions.

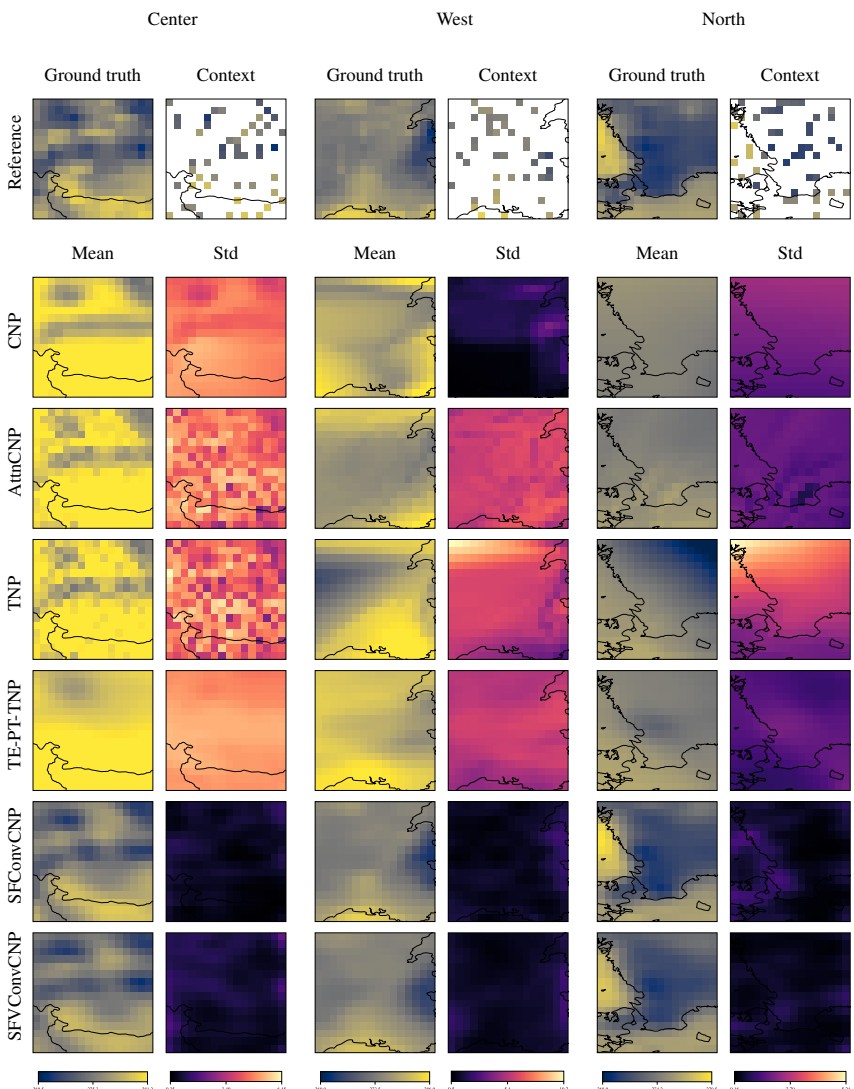

*Figure 7.* Example predictions on the ERA5 climate regression benchmark. The three column groups (Center, West, North) correspond to the evaluation regions of Europe defined by the latitude/longitude windows in Appendix E.3.2; each shows a single held-out test task with one randomly selected time step held fixed across model rows. The top row gives the ground-truth temperature field (cells with missing data appear in red) and the context mask (non-context cells in white). Subsequent rows show, for each model, the predictive mean (shared color scale with the ground truth) and the predictive standard deviation. The predictive panels show the model's output at every cell, including at the observed context positions; values shown at those positions are model outputs (conditioned on the context), not the original observations. Standard-deviation panels within each region share a common color scale (bottom row of colorbars); the leftmost colorbar per region calibrates temperature, the rightmost calibrates uncertainty.

