# OpenReview forum: "Revisiting Neural Processes via Fourier Transform and Volterra Series"
_ICML.cc/2026/Conference — ICML 2026 regular_

### Official Review · Reviewer_iyk8 · 2026-02-14

**Soundness:** 3
**Presentation:** 3
**Significance:** 3
**Originality:** 4
**Overall Recommendation:** 5
**Confidence:** 4

**Summary:**

The paper targets two limitations of Neural Processes (NPs): limited model interpretability and poor scalability in ConvCNPs or attention-based NPs. It uses Volterra series theory to express translation-equivariant operators as sums of higher-order convolutions, yielding a more interpretable operator form. For efficient implementation on irregular inputs, the authors introduce Set Fourier Convolutions (SFConv), which parameterize these convolutions in the frequency domain via a continuous Fourier transform on scattered points. This avoids spatial discretization and achieves  a linear complexity. They instantiate two models, SFConvCNP  and SFVConvCNP, along with strong results on 1D/2D/3D regression benchmarks against competitive baselines.

**Compliance With Llm Reviewing Policy:**

Affirmed.

**Final Justification:**

The paper studies an important problem and has several strengths in motivation and technical development. In particular, I find the core idea of Volterra series interesting and potentially valuable to the community of Neural Process.

After reading the authors’ rebuttal, I find that my main concerns have been largely addressed.

**Key Questions For Authors:**

- Does the SFConv block correspond to the first-order term in the Volterra expansion (Eq. 4)?

- Appendix D.2 fixes $|\Xi|$. How does performance change as $|\Xi|$ increases (capacity scaling)? Is there evidence of underfitting or overfitting?

- The continuous transform relies on a Gaussian kernel $\psi$ with learnable length scales. How sensitive is training to length-scale initialization, especially when the initial bandwidth is mismatched to sampling density (e.g., overly narrow causing noisy spectra or overly wide causing oversmoothing)?

**Limitations:**

Yes

**Strengths And Weaknesses:**

## Strengths

- strong motivation: The paper clearly motivates the work by identifing two practical limitations: limited interpretability of NP models and poor scalability of grid- or attention-based constructions. Accordingly, the authors clearly explain how the proposed solution is designed to address both limitations.

- Design of SFConv: The proposed Set Fourier Convolution cleverly utilizes the continuous Fourier transform to handle irregular data without forcing it onto a dense spatial grid, effectively solving the "grid mismatch" problem. The continuous-transform derivation (Eq. 12) and the spectral weighting scheme are presented clearly and with solid mathematical grounding.

- Well-structured experiments: The design of numerical experiments is well-strutrued, specifically the ablation on the frequency grid $\Xi$. By fixing the total grid size, By fixing the total grid size, the study separates spectral coverage from resolution, clarifying sensitivity to function structure and the role of $\Xi$ as a key hyperparameter.

## Weakness

- Limited discussion of linear-attention baselines: While standard attention is quadratic, efficient transformer variants (e.g., Linformer [Wang et al., (2020)], Performer [Choromanski et al., (2020)]) offer near-linear scaling. Positioning against these families would strengthen the scalability narrative.

- Missing overview and qualitative figures: A schematic of the architecture/data flow and qualitative visualizations (e.g., 1D regression functions,spectral effects) would offer more insights, and improve clarity and readbility.

### References

- Wang, Sinong, et al. "Linformer: Self-attention with linear complexity." arXiv preprint arXiv:2006.04768 (2020).

- Choromanski, Krzysztof, et al. "Rethinking attention with performers." arXiv preprint arXiv:2009.14794 (2020).

---

> ### Author Rebuttal · Authors · 2026-03-31
>
> We thank the reviewer for the constructive feedback.
>
> ---
>
> **Q1 — Does SFConv correspond to the first-order Volterra term?**
>
> > "Does the SFConv block correspond to the first-order term in the Volterra expansion (Eq. 4)?"
>
> Yes, with a nuance. The first-order Volterra term $\mathcal{C}_1[h]$ (Eq. 5) is a standard linear convolution for any valid $h$. SFConv corresponds to this term when $h$ takes the specific form $h := \rho_e[\mathcal{D}_c]$ (Eq. 2) — a sum of shifted kernels over context observations, akin to kernel regression. This structure admits a **closed-form Fourier transform** (Eq. 14), enabling exact frequency-domain evaluation without spatial discretisation. The combination of first-order Volterra structure with closed-form spectral computation via this functional embedding is the core technical contribution of SFConv.
>
> ---
>
> **Q2 — Capacity scaling: underfitting or overfitting?**
>
> > "How does performance change as capacity scales?"
>
> The existing ablation (Appendix D.2) fixes $|\Xi|$ while varying $\xi_{\max}$ and $\Delta_\mathcal{F}$ jointly, conflating coverage and resolution effects, so it does not directly answer the capacity question. We ran a new ablation holding $\Delta_\mathcal{F} = 0.1$ fixed while increasing $\xi_{\max}$, so that $|\Xi|$ and parameter count grow monotonically (avg. target log-likelihood, 8,000 test samples; fixed: embed_channels=128, volterra_rank=4, num_blocks=5):
>
> | $\xi_{\max}$ | $\|\Xi\|$ | Params | Matérn-5/2 | Sawtooth |
> |---|---|---|---|---|
> | 1.0 | 10 | 8.8M | −0.002 | −0.584 |
> | 2.5 | 25 | 19.8M | +0.025 | +0.526 |
> | 5.0 | 50 | 38.3M | +0.029 | +1.173 |
> | 7.5 | 75 | 56.7M | +0.028 | +1.207 |
> | 10.0 | 100 | 75.1M | +0.028 | +1.218 |
>
> No evidence of overfitting: sawtooth performance improves monotonically (primary failure mode at low capacity is underfitting); Matérn-5/2 saturates at $|\Xi| = 25$ but does not degrade as capacity grows. We will extend the ablation to higher $\xi_{\max}$ and add this interpretation to Appendix D.2.
>
> ---
>
> **Q3 — Sensitivity to lengthscale initialisation**
>
> > "How sensitive is training to length-scale initialisation?"
>
> No sensitivity was observed in practice. The learnable lengthscale $\varrho_d$ is initialised to $0.1$ and jointly optimised, following ConvCNP. Across all benchmarks, models learned appropriate lengthscales consistently, as evidenced by low cross-seed variance in Tables 1–3.
>
> ---
>
> **Efficient transformer designs and linear-attention baselines**
>
> > "Positioning against Linformer, Performer … would strengthen the scalability narrative."
>
> We agree this is important context. In the NP literature, the dominant efficient-attention paradigm is the _pseudo-token_ (inducing-point) design — Set Transformer (Lee et al., 2019) and Perceiver (Jaegle et al., 2021) — which reduces $O(N^2)$ attention to $O(Nm)$ by attending over $m \\ll N$ learned points, naturally handling unordered, irregularly spaced inputs. By contrast, most efficient transformer variants such as Cosformer are tailored to 1-D sequences and do not generalise readily to point clouds. Within the benchmarks commonly used to evaluate NPs, scalability is not a binding constraint for non-equivariant TNPs (FlashAttention mitigates this substantially). The bottleneck arises when imposing _translation equivariance_: full TE-TNP requires embedding the concatenation of pairwise positional encodings with the dot product of feature embeddings through an MLP, which is incompatible with FlashAttention. TE-PT-TNP (Ashman et al., 2024) addresses this by replacing full attention with equivariant inducing-point cross-attention, and is the only tractable equivariant attention baseline in 2-D/3-D settings — albeit at the cost of permitting degeneracy (Ashman et al., 2024, §3.1). It is already included in all experiments. We will acknowledge Linformer, Performer, Cosformer, Set Transformer, and Perceiver in the related-work section.
>
> ---
>
> **Architecture schematic**
>
> We will add a schematic illustrating the SFConvCNP data flow: context set → functional embedding (Eq. 2) → closed-form FT (Eq. 14) → frequency-domain multiplication (Eq. 16) → inverse FT → predictions, annotating where the second-order Volterra path branches off in SFVConvCNP.
>
> ---
>
> **References**
>
> - Ashman et al. (2024). "Translation Equivariant Transformer Neural Processes."
> - Jaegle et al. (2021). "Perceiver: General Perception with Iterative Attention."
> - Lee et al. (2019). "Set Transformer: A Framework for Attention-Based Permutation-Invariant Neural Networks."
> - Qin et al. (2022). "cosformer: Rethinking Softmax in Attention."
> - Feng et al. (2022) "Latent bottlenecked attentive neural processes."

---

> > ### Author Rebuttal · Reviewer_iyk8 · 2026-04-01
> >
> > I appreciate the authors’ response, which has addressed my concerns, and I have adjusted my assessment accordingly.

---

> > > ### Author Response · Authors · 2026-04-08
> > >
> > > Dear Reviewer iyk8,
> > >
> > > Thank you very much for taking the time to carefully read our rebuttal and for adjusting your assessment. We truly appreciate your constructive engagement throughout the review process.
> > >
> > > Best regards,
> > >
> > > The Authors

---

### Official Review · Reviewer_ojnn · 2026-02-24

**Soundness:** 4
**Presentation:** 4
**Significance:** 3
**Originality:** 4
**Overall Recommendation:** 5
**Confidence:** 2

**Summary:**

This paper revisits translation-equivariant Neural Processes and introduces two new concepts: (1) characterizing continuous translation-equivariant operators through their Volterra expansions, representing them as sums of higher-order convolutions, and (2) introducing set Fourier convolutions (SFConvs), a frequency-domain parameterization that operates directly on irregularly sampled sets. Although this manuscript is not within my core expertise, I did enjoy reading it and found it relatively easy to follow.

**Compliance With Llm Reviewing Policy:**

Affirmed.

**Final Justification:**

After my initial review and acknowledgment of the authors' rebuttal, my final recommendation is to accept the paper. I have increased my score in response to the authors' rebuttal.

**Key Questions For Authors:**

In what scenario would I prefer the proposed methodology over an easier approach like a GP?

**Limitations:**

I would be curious to read more about weaknesses compared to more baseline approaches like a (stationary and non-sttaionary) GP.

**Strengths And Weaknesses:**

Soundness:
Is the submission technically sound?
Yes, as far as I can assess.

Are claims well supported?
Yes, as far as I can assess.

Are the methods used appropriately?
Yes, as far as I can assess, with one exception: coverage probability should be added to experiments (or CRPS as an alternative)


If the paper includes theoretical results, are the proofs correct and based on reasonable assumptions?
Yes, as far as I can assess.


Are the experiments well-designed?
Yes, but other scores should be added, in particular, probability coverage and CRPS.

Are the authors careful and honest about evaluating both the strengths and weaknesses of their work?
As far as I can tell, weaknesses are not explicitly discussed. Comparative tests seem to focus on very specific methods. I might suggest adding more baseline approaches, like a pure GP.

Presentation:
Is the submission clearly written and well-structured?
Yes

Is the overall narrative easy to follow?
Yes

Does the work properly position itself in the context of prior/concurrent literature and clearly discuss how it differs?
Yes, even though I am not an expert in this area.

Significance:
Does the paper address an important or relevant problem?
Yes, distributional regression is very relevant in many ML applications.

Does it advance understanding, capabilities, or practice in machine learning?
Yes

Could it influence future research or applications (e.g., other researchers or practitioners are likely to use the ideas or build on them)?
Yes

Is the scope of impact broad or specialized, and is that appropriate for the contribution?
The scope is potentially broad. But I find the comparison to other methods and a discussion of strengths and weaknesses a little unsatisfying.

Even if the improvements are modest or domain-specific, could they unlock new directions or provide practical utility?
Yes


Originality:
Does the work provide new insights, deepen understanding, or highlight important properties of existing methods?
Yes

Does the work introduce new tasks, methods, theory, data, or perspectives that advance the field in some dimensions?
Yes

Does this work offer a novel combination of existing techniques, and is the reasoning behind this combination well-articulated?
Yes

Are the contributions clearly distinguished from closely related literature, and is the novelty well justified?
Yes

---

> ### Author Rebuttal · Authors · 2026-03-31
>
> We thank the reviewer for the positive assessment.
>
> ---
>
> **Q1 — When to prefer the proposed methodology over a GP?**
>
> > "In what scenario would one prefer the proposed methodology over an easier approach like a GP?"
>
> Three concrete scenarios favour SFConvCNP over a GP.
>
> 1. **Scalability.** Exact GP inference is $\mathcal{O}(N^3)$; SFConvCNP is $\mathcal{O}(|\Xi|(|\mathcal{D}_c|+|\mathcal{D}_q|))$, i.e. linear. On the Kolmogorov benchmark (~4,096 context points per task), exact GP inference is intractable without sparse approximations.
>
> 2. **Non-Gaussian processes.** Standard GPs cannot model non-Gaussian processes without elaborate transformations (e.g. warped or deep GPs). Our sawtooth and square-wave benchmarks (Table 1) are precisely such cases, and SFConvCNP handles them naturally.
>
> 3. **Amortised kernel learning.** A GP requires a manually specified or optimised kernel — a separate step for each new dataset. SFConvCNP meta-learns a spectral kernel representation across tasks, enabling fast amortised inference at test time.
>
> We will add a discussion of these trade-offs to the Conclusion, including when a GP *is* preferable: small $N$, a known stationary kernel, or when an exact posterior is required.
>
> ---
>
> **Coverage probability and CRPS**
>
> > "Coverage probability (or CRPS as an alternative) should be added to the experiments."
>
> We agree and provide preliminary CRPS results below (subset of 1-D synthetic benchmarks, target points, single seed; lower is better). Full results across all benchmarks will appear in the revised version.
>
> **Table R4 — CRPS on 1-D synthetic benchmarks**
>
> | Model | Matérn-5/2 | Sawtooth | Square |
> |---|---|---|---|
> | CNP | 0.196 | 0.338 | 0.580 |
> | AttnCNP | 0.196 | 0.338 | 0.500 |
> | TNP | 0.172 | 0.338 | 0.572 |
> | TE-TNP | 0.171 | 0.338 | 0.403 |
> | TE-PT-TNP | 0.172 | 0.338 | 0.580 |
> | ConvCNP | 0.169 | 0.143 | 0.347 |
> | **SFConvCNP (ours)** | **0.169** | **0.064** | **0.296** |
> | **SFVConvCNP (ours)** | **0.169** | **0.065** | **0.255** |
>
> These metrics were not included in the submission because we followed standard practice in the NP literature, which uniformly reports log-likelihood (Gordon et al., 2020; Nguyen & Grover, 2022; Lee et al., 2023) to maintain direct comparability with prior work. The CRPS results confirm our models are competitive.
>
> ---
>
> **Weaknesses relative to GP baselines and direct GP comparison**
>
> > "Weaknesses are not explicitly discussed. Comparative tests seem to focus on very specific methods. Additional baseline approaches (e.g., a pure GP) would be beneficial."
>
> We address this on two levels: why a direct GP comparison is non-trivial, and what the genuine weaknesses of our method are.
>
> **Why a direct GP comparison is not straightforward.** NPs and GPs operate under fundamentally different training regimes, making head-to-head comparison potentially misleading. NPs meta-learn a shared model across *many realisations* of a stochastic process; GPs are fitted to observations from a *single realisation* via marginal-likelihood optimisation. As noted in Kim et al. (2019), this difference in inductive setup means the two classes are not directly comparable, and such comparisons have been avoided in prior NP literature. Additionally, kernel selection is non-trivial even in low dimensions, and this difficulty compounds with input dimensionality — NPs sidestep this entirely by meta-learning a flexible prior directly from data.
>
> **Genuine weaknesses of our method.** We will add an explicit weaknesses paragraph to the Conclusion covering:
>
> - **Mean-field predictive.** CNPs (including ours) produce marginal predictives and cannot model inter-query correlations. This is a known limitation of the CNP class; variants such as LNPs or GNPs address it at additional cost.
> - **Bayesian consistency.** Exact GPs satisfy Bayes' update rule; NPs do not in general — a class-wide limitation, not specific to our method.
> - **When a GP is preferable.** In data-limited meta-learning regimes, or when a known kernel and exact posterior are available, a GP remains the more principled tool.
>
> Notwithstanding these caveats, we will include a GP oracle comparison on the 1-D synthetic benchmarks (using the ground-truth kernel) as a reference performance ceiling, providing a meaningful GP reference point without conflating the two training regimes above.
>
> ---
>
> **References**
>
> - Gordon et al. (2020). "Convolutional Conditional Neural Processes." *ICLR 2020.*
> - Kim et al. (2019). "Attentive Neural Processes." *ICLR 2019.*
> - Lee et al. (2023). "Martingale Posterior Neural Processes." *ICLR 2023.*
> - Nguyen & Grover (2022). "Transformer Neural Processes." *arXiv:2207.04179.*

---

> > ### Author Rebuttal · Reviewer_ojnn · 2026-03-31
> >
> > My concerns are all addressed. I will consider increasing my score.

---

> > > ### Author Response · Authors · 2026-04-08
> > >
> > > Dear Reviewer ojnn,
> > >
> > > Thank you very much for taking the time to carefully read our rebuttal and for raising your score. We truly appreciate your constructive engagement throughout the review process.
> > >
> > > Best regards,
> > > The Authors

---

### Official Review · Reviewer_TVDt · 2026-03-12

**Soundness:** 3
**Presentation:** 4
**Significance:** 3
**Originality:** 3
**Overall Recommendation:** 5
**Confidence:** 4

**Summary:**

This paper constructs a new pair of families of translation-equivariant neural processes (NPs) using Fourier decompositions and Volterra series. The authors show that continuous translation-equivariant operators can be characterized through Volterra expansions, expressing them as sums of higher-order convolutions. Building on this construction, they introduce set Fourier convolutions (SFConvs), a frequency-domain parameterisation that operates directly on irregularly sampled sets, enabling global receptive fields without spatial discretisation and scaling linearly with the number of observations. Using these components, the paper proposes two new conditional neural process architectures: SFConvCNP, which stacks SFConv blocks with nonlinearities, and SFVConvCNP, which uses the Volterra formulation. Experiments on synthetic and real-world datasets demonstrate competitive or improved performance relative to existing neural process baselines while offering an arguably more interpretable and natural model for translation-equivariant problems.

**Compliance With Llm Reviewing Policy:**

Affirmed.

**Final Justification:**

I like the paper, recommend acceptance, and retain my score.

**Key Questions For Authors:**

My main comment / question is that I expected a little more discussion about the relationship to Spectral Convolutional Conditional Neural Process from Mohseni & Duffield, 2025. Although this paper was cited, it was only mentioned in passing. Importantly, to my mind, it wasn’t mentioned in the FNO discussion on page 5 left hand column nor does the related work section mention this work. The FNO construction in (Mohseni & Duffield, 2025) is arguably the closest in spirit to the current work (albeit different).

In this light, I think table should also include this model class too and probably the experiments. In general, outside of NPs, FNOs perform really well in simple band limited / smooth problems, but suffer on problems with more complex spectral structure. See here  https://arxiv.org/abs/2209.15616 I would expect to similar things for SFConvCNPs and SFVConvCNPs. It would be interesting to comment on this — I might have expected this to show up in the CIFAR-10 experiments, but it doesn’t. Perhaps CIFAR-10 is too simple?

On line 66 the paper says "However, this comes at the cost of quadratic complexity in the number of data points, introducing a different scalability bottleneck.” I agree that this is true for the vanilla TNPs. But it’s probably fair here to acknowledge that there are variants which sidestep this problem to some degree e.g.

Latent Bottlenecked Attentive Neural Processes, Feng et al, ICRL 2023
Gridded Transformer Neural Processes for Large Unstructured Spatio-Temporal Data, Ashman et al., ICML 2025

It was interesting that the TNPs struggled with the non-GP 1d tasks. I was slightly surprised that they weren’t competitive with the convCNP here as they tend to be comparable on the GP tasks e.g. see the TE-TNP paper figure 2 (right), and the convCNP can struggle to capture the discontinuities in these signals. I might have expected the TNP to be better.

**Limitations:**

I like the paper and vote for acceptance.

As noted above, I think the experiments are comparatively weak (what is there is good, but the tasks are quite simple and there isn't a huge range).

I would also like to see more of a comparison to FNO-based NPs as these are close cousins of the new approaches discussed here.

**Strengths And Weaknesses:**

Strengths

The paper was very clearly written and presented. It was solid throughout -- clear technical development, excellent appendices with e.g. full experimental details -- sensible experiments.

I like the central construction which leverages the Fourier transform of the convolutional deep set to sidestep the use of grids. I think this was  a neat contribution. The set Fourier convolutions are composed to build models by interleaving these layers with point-wise non-linearities. This architectural motif resembles transformers (which interleave spatial attention and MLPs, GNNs which interleave spatial message passing with MLPs, FNOs which interleave spatial-frequency processing with MLPs).

Weaknesses

I thought arguably the most natural comparison point to this model family is the FNO based NPs (Spectral NPs). This line of work is mentioned in the paper, but I would have expected it to be discussed in more detail and featured more centrally, including in the experiments (unless there is good reason not to) (see below).

The experiments were slightly light in terms of volume and complexity. I think that there is enough here for me to vote for acceptance, but the range and complexity of the experiments is not the strongest part of the paper.

---

> ### Author Rebuttal · Authors · 2026-03-31
>
> We thank the reviewer for the constructive feedback.
>
> ---
>
> **Q1 — SpectralCNP comparison (Mohseni & Duffield, NeurIPS 2025)**
>
> > "What is the relationship between SFConvCNP and SpectralCNP, and how does it compare experimentally?"
>
> We agree SpectralCNP is the closest prior work in spirit and will give it a more central treatment in the revision (Section 3.4 + Table 1).
>
> **Architectural distinction.** SpectralCNP obtains spectral coefficients by discretising the functional embedding on a regular spatial grid and applying the DFT. This reintroduces grid dependency: DFT bins are grid-specific, so learned weights correspond to different physical frequencies at different resolutions (Appendix C of our paper). SFConvCNP instead derives ĥ(ξ) analytically from the deep-set structure (Eq. 14), yielding a spectral parameterisation that is invariant to observation locations.
>
> **Experimental comparison.** We trained SpectralCNP on the new **PredPrey** benchmark (predator–prey dynamics, simulated and real sim-to-real splits). Results on all other benchmarks will appear in the revised Table 1.
>
> Table R1 — PredPrey (test log-likelihood, higher is better):
>
> | Model         | Simulated          | Real               |
> |---------------|--------------------|--------------------|
> | CNP           | −0.392 ± 0.000     | −0.211 ± 0.002     |
> | AttnCNP       | −0.085 ± 0.087     | −0.078 ± 0.041     |
> | TNP           |  0.385 ± 0.041     |  0.003 ± 0.015     |
> | TE-TNP        |  0.370 ± 0.081     |  0.021 ± 0.016     |
> | TE-PT-TNP     |  0.164 ± 0.102     | −0.041 ± 0.025     |
> | ConvCNP       |  0.324 ± 0.054     |  0.027 ± 0.014     |
> | SConvCNP      |  0.452 ± 0.001     |  0.026 ± 0.006     |
> | SFConvCNP     |  0.430 ± 0.002     |  0.032 ± 0.005     |
> | SFVConvCNP    |  0.394 ± 0.001     |  0.046 ± 0.002     |
>
> To further broaden evaluation we also added ERA5 experiments (following Ashman et al., ICML 2024), a large-scale spatio-temporal benchmark with rich spectral content, where our models remain competitive.
>
> ---
>
> **Q2 — Spectral complexity and FNO limitations**
>
> > "How does the method handle spectral complexity, and what are the limitations relative to FNO-style approaches?"
>
> We agree this is an important point. We conjecture that deriving spectral coefficients analytically (rather than via a grid DFT) provides a smoother frequency decomposition with better inductive biases for complex spectral structure. The ERA5 results offer some empirical support, but a targeted study of spectral complexity is a valuable direction for future work that we will note explicitly.
>
> ---
>
> **Q3 — Quadratic complexity — efficient transformer variants**
>
> > "The quadratic complexity of attention is a concern. Have efficient transformer variants been considered?"
>
> Both Feng et al. (ICLR 2023) and Ashman et al. (ICML 2025) are already cited; we will strengthen the discussion at line 66. Two points: (1) for non-equivariant TNPs our FlashAttention implementation is already efficient enough that pseudo-token variants were unnecessary; (2) for equivariant variants, Ashman et al. (2024, §3.1) note that pseudo-token approaches can degenerate — we discuss this in Section 3.4 and include TE-PT-TNP as a baseline throughout, making it the most directly comparable efficient equivariant method.
>
> ---
>
> **Q4 — TNP struggling on non-GP 1-D tasks**
>
> > "Why does TNP struggle on non-GP 1-D tasks such as sawtooth benchmarks?"
>
> This failure mode is independently reported by Mohseni & Duffield (NeurIPS 2025, §4.1), who also could not train TNP/TE-TNP on sawtooth benchmarks and attribute it to spectral bias (Rahaman et al., 2019; Vasudeva et al., 2025). We will add a cross-reference in the revised paper.
>
> ---
>
> **References**
>
> - Mohseni & Duffield (2025). "Spectral Convolutional Conditional Neural Process."
> - Feng et al. (2023). "Latent Bottlenecked Attentive Neural Processes."
> - Ashman et al. (2024). "Translation Equivariant Transformer Neural Processes."
> - Ashman et al. (2025). "Gridded Transformer Neural Processes for Large Unstructured Spatio-Temporal Data."
> - Bruinsma et al. (2023). "Autoregressive Conditional Neural Processes."
> - Vasudeva et al. (2024). "Transformers learn low sensitivity functions: Investigations and implications."

---

> > ### Author Rebuttal · Reviewer_TVDt · 2026-04-01
> >
> > Thank you for the responses. I'm very happy to maintain my "accept" recommendation and score.

---

> > > ### Author Response · Authors · 2026-04-08
> > >
> > > Dear Reviewer TVDt,
> > >
> > > Thank you for your time and for the constructive engagement throughout the review process. We are glad our responses were satisfactory.
> > >
> > > Best regards,
> > > The Authors

---

### Official Review · Reviewer_LfFw · 2026-03-15

**Soundness:** 3
**Presentation:** 2
**Significance:** 3
**Originality:** 2
**Overall Recommendation:** 5
**Confidence:** 2

**Summary:**

The paper targets continuous translation-equivariant operators for irregular data. It studies them within neural processes. The theory is developed through Volterra expansions and proposes set Fourier convolutions. The methods are claimed to achieve global context and scale linearly. The resulting models perform well on 1d/2d/3d experiments.

**Compliance With Llm Reviewing Policy:**

Affirmed.

**Final Justification:**

The rebuttal addressed my main concerns.

**Key Questions For Authors:**

1. Can you support the claim of approximately global receptive fields more directly, both theoretically and empirically?

2. Can you clarify the role of Sections 3.1 and 3.2 more explicitly, separating what is already established in the literature from the main methodological moves of the paper?

3. Can you make the assumptions and purpose of Equations 12, 13, and 14 more explicit, possibly by elevating them to lemma-style statements and briefly explaining what each one buys?

**Limitations:**

The paper should more explicitly discuss the current gap between the claims and demonstrated results, such as the translation-deterioration claim (does AttnCNP/TNP actually deteriorate) and the approximately global receptive field claim, and clarify the scope under which these statements should be understood.

**Strengths And Weaknesses:**

**Strengths**
The paper is technically coherent and empirically well supported. Evaluations across 1d/2d/3d tasks, strong performance against competitive baselines, together with ablations, seem reasonable to me. The design choices also seem reasonable. Although the parts of the work are not all novel in isolation, I believe they still make a meaningful contribution as a whole. In that sense, the significance is meaningful even if the originality is more in the combination.

**Weaknesses**
The main weakness is presentation: The abstract is too long. Sections 3.1 and 3.2 currently read as if they are further background; the setup parts blur the line between what is already in the literature and what is the methodological move the paper makes. Section 3.3 introduces the main architectures the paper contributes, but they are underspecified in the main text, with no reference to Appendix E. More generally, the paper would benefit from rebalancing the main text to highlight its actual contributions more clearly.

Also some claims would benefit from sharper support. The claim of approximately global receptive fields seem plausible, but given that it is given as one of the main properties of the proposed methods, it is not convincingly demonstrated, especially since theoretical receptive field need not match effective receptive field in practice (as is the case for CNNs). Also, Eqns 12,13, and 14 are central derivations, but they are currently presented in a technical narration. They should be highlighted to make their role and assumptions immediately clear.

--

Overall, despite the presentation issues, I think the paper has a good motivation, sound technical tools, and strong empirical results. Since presentation seems fixable, I am leaning toward weak accept. **Note:** This is not my main area, so I may be missing some context from the literature.

---

> ### Author Rebuttal · Authors · 2026-03-31
>
> We thank the reviewer for the constructive feedback.
>
> ---
>
> **Q1 — Approximately global receptive fields**
>
> > "The claim of approximately global receptive fields is not convincingly demonstrated."
>
> We provide (1) a sharpened theoretical argument and (2) a gradient-based effective receptive field (ERF) analysis (Luo et al., 2016).
>
> **Theory.** Each frequency component $\hat{h}(\xi) = \mathcal{F} [\psi_e] (\xi)\sum_{(x_c,y_c)\in\mathcal{D}_c} \varphi(y_c)\,e^{-i2\pi\langle x_c,\xi\rangle}$ aggregates over *all* context points simultaneously, so every query output is influenced by the entire context set at every layer by construction.
>
> **Empirical ERF.** We compute $\mathrm{ERF}(x_t,x_c)=\mathbb{E}_{y_c}[(\partial\mu(x_t)/\partial y_c)^2]$ for trained and randomly initialised SFConvCNP and ConvCNP under a $2\times 2$ design (random vs. data-driven inputs $\times$ trained vs. random-init weights) on three 1-D tasks.
>
> - At random initialisation, SFConvCNP's ERF is near-uniform across all distance bands (110–138% of near-field), confirming that global reach is an architectural property independent of training.
> - After training, far-field influence is 23–31% for GP-Periodic and 12–23% for shorter-range tasks, but *never drops to zero*.
> - ConvCNP falls to $\leq 3\%$ beyond $|x_t - x_c| \geq 0.5$ in all settings.
>
> Full ERF tables and kernel visualisations will appear in the revised version.
>
> ---
>
> **Q2 — Sections 3.1 and 3.2: background vs. contribution**
>
> > "Sections 3.1–3.2 blur the boundary between prior work and our contributions."
>
> **Section 3.1 (Contribution 1).** While the Volterra series is classical, prior deep learning adoptions (Roheda et al., JMLR 2024; Li et al., JMLR 2022) treat it as an engineering device with no connection to equivariance or operator theory. Our use is fundamentally different: approaching translation-equivariant maps over finite datasets from an operator-theoretic perspective, we adopt the Volterra series because it yields an *explicit, constructive* form for any such operator — enabling transparent inductive-bias analysis, in contrast to the notoriously opaque analysis that accompanies stacked nonlinear convolutions or softmax attention.
>
> **Section 3.2 (Contribution 2).** DFT/FNO methods operate on fixed grids; as shown in Appendix C, retained modes correspond to different physical frequencies at different resolutions. The key step absent from prior work is that the functional embedding (Eq. 3) admits a closed-form continuous FT (Eq. 14), enabling $\hat{h}(\xi)$ to be evaluated at any physical frequency without discretisation. The result is a spatial grid-free, resolution-invariant $\mathcal{O}(|\Xi|(|\mathcal{D}_c|+|\mathcal{D}_q|))$ convolution operator.
>
> *Roadmap: Section 3.1 is Contribution 1 — an operator-theoretic treatment of translation-equivariant maps via the Volterra series, yielding an explicit form for transparent inductive-bias analysis. Section 3.2 is Contribution 2 — the closed-form FT of the functional embedding (Eq. 14) gives a grid-free, resolution-invariant, linear-complexity spectral parameterisation distinct from DFT/FNO approaches.*
>
> ---
>
> **Q3 — Equations 12–14: purpose and assumptions**
>
> > "Eqs. 12–14 lack explicit labelling of their role and assumptions."
>
> Eq. 12 characterises all continuous TE operators (continuity + shift-commutativity); Eq. 13 introduces the cascade decomposition (finite-rank quadratic kernel); Eq. 14 derives the closed-form FT of the functional embedding ($\psi_e \in L^1 \cap L^2$, satisfied by the Gaussian) — the step that enables grid-free linear-complexity SFConv. We will add explicit labels in the revised version.
>
> ---
>
> **On the limitations note**
>
> > "The gap between claims and demonstrated results needs explicit discussion."
>
> **Translation-deterioration claim.** Models trained on $[-3,3]$ are evaluated at shifts $\tau \in \{0,3,6,9,12,15\}$, measuring target log-likelihood over 8,000 tasks. Non-equivariant models degrade catastrophically: EQTNP drops from $0.22$ to $-258$ on GP-RBF and to $-5\times10^8$ on GP-Periodic; CNP diverges to $-274$ on GP-Matérn-5/2 at $\tau=6$; ACNP degrades more gradually but consistently. SFConvCNP and SFVConvCNP maintain *identical* performance at all shifts, confirming the mathematical guarantee: translation equivariance is structural, since SFConv (Eq. 16) commutes with translation by construction (Gordon et al., 2019). The full results table will be included in the revised appendix.
>
> **Approximately global receptive field.** Addressed in Q1.
>
> ---
>
> **References**
>
> - Boyd & Chua (1984). "Fading Memory and the Problem of Approximating Nonlinear Operators with Volterra Series."
> - Gordon et al (2020). "Convolutional Conditional Neural Processes."
> - Li et al (2022). "Toward Understanding Convolutional Neural Networks from Volterra Convolution Perspective."
> - Roheda et al (2024). "Volterra Neural Network"
> - Luo et al (2016). "Understanding the Effective Receptive Field in Deep Convolutional Neural Networks."

---

> > ### Author Rebuttal · Reviewer_LfFw · 2026-04-04
> >
> > I thank the authors for their response which addressed all of my concerns. I increased my score accordingly.

---

> > > ### Author Response · Authors · 2026-04-08
> > >
> > > Dear Reviewer LfFw,
> > >
> > > Thank you very much for taking the time to carefully read our rebuttal and for raising your score. We truly appreciate your constructive engagement throughout the review process.
> > >
> > > Best regards,
> > > The Authors

---

### Decision · Program_Chairs · 2026-04-30

**Decision:**

Accept (regular)

**Comment:**

Neural processes (NPs) offer a powerful framework for probabilistic functional inference. In many domains, however, these functions exhibit symmetries that can be exploited to improve sample efficiency and generalization. Existing translation-equivariant NPs are limited as they are constructed by stacking generic components with nonlinearities, convolutional designs are based on localized receptive fields and require dense discretization. Attention-based methods avoid these issues, but scale quadratically with the number of observations. This paper address these challenges by caraterizing continuous translation-equivariant operators via their Volterra expansions. Moreover, it introduces set Fourier convolutions, a frequency-domain parameterization that operates directly on irregularly sampled sets. Using these ideas, the authors propose two families of conditional NPs (CNPs). Experiments on synthetic and real-world datasets demonstrate the efficacy of their methods. All the reviewers agree that this paper is a nice submission with clear strengths such as  technically coherent and empirically well supported, with  excellent appendices, and  sensible experiments. While the range and complexity of the experiments is not the strongest part of the paper, it is acceptable. The paper studies an important problem is well motivated and technically developed. All the issues raised by the reviewers have been properly addressed during the rebuttal.